# Polyclonal selection of immune checkpoint mutations in thyroid autoimmunity

Pantelis A. Nicola[1,15], Andrew R. J. Lawson[1,15 ✉], Alexandra Tidd[1], Juliette Imbert[1], Yoshihiro Ishida[1], Luke A. Wylie[1], Paul A. Scott[1], Kenny Roberts[2], Luke M. R. Harvey[1], Stefanie V. Lensing[1,3], Wei Cheng[1,3], Federico Abascal[1], Daniel Leongamornlert[1], Yvette Hooks[1], Matthew Mayho[3], Nicole Müller-Sienerth[3], Sara Widaa[3], Laura Mincarelli[1], James Illing[4], Flavia Peci[4], Bee Ling Ng[4], Georgeina L. Jarman[1], Andrew J. C. Russell[5,6], Krishnaa T. A. Mahbubani[7,8], Kourosh Saeb-Parsy[7,8], Anna L. Paterson[9], Krishna Chatterjee[10], Raheleh Rahbari[1], Omer Ali Bayraktar[11], Michael R. Stratton[1], Peter J. Campbell[1,12], John A. Tadross[9,10,13], Nadia Schoenmakers[10,14] & Iñigo Martincorena[1 ✉]

Our immune system contains multiple checkpoints to prevent the activation of self-reactive lymphocytes. How some lymphocytes escape these constraints to cause autoimmune disease remains poorly understood. A long-standing hypothesis posits that somatic mutations in immune regulatory genes may enable self-reactive lymphocytes to bypass tolerance checkpoints[1–3], but testing this has been challenging owing to technical limitations. Here we used whole-exome and targeted NanoSeq[4,5], an accurate single-molecule DNA sequencing protocol, to comprehensively search for driver mutations in autoimmune thyroid disease. This showed many B cell clones convergently acquiring loss-of-function mutations in the key immune checkpoint genes *TNFRSF14* (also known as *HVEM*) and *CD274* (which encodes PD-L1), as well as less frequent mutations in other immune genes. In highly inflamed biopsies, we detected tens to hundreds of independent immune checkpoint mutant clones. Laser microdissection, methylation sequencing, spatial transcriptomics, immunostaining, single-nucleus DNA sequencing and antibody synthesis localized these mutations to B cells, confirmed some to be self-reactive and identified clones carrying multiple hits. We found widespread *TNFRSF14* biallelic loss, and clones with as many as 4–6 driver mutations. While each clone accounts for a small fraction of cells (typically less than 1%), the myriad mutant clones in each donor amounted to a substantial fraction of B cells harbouring driver mutations. Our results support the hypothesis that somatic mutations in autoimmune lymphocytes may allow them to escape tolerance constraints through a polyclonal cascade of somatic evolution, providing insights into the molecular basis of autoimmune disease.

Autoimmune diseases affect 5–10% of the global population[6], but their molecular bases remain incompletely understood. During B and T cell development, somatic recombination generates highly diverse repertoires of antigen receptors. The stochasticity of this process results in the production of lymphocytes capable of recognizing self-antigens. To prevent autoimmune attack, many of these self-reactive lymphocytes are eliminated by central tolerance mechanisms in the thymus and bone marrow. However, large numbers of potentially self-reactive lymphocytes escape into circulation, where they are restrained by peripheral tolerance mechanisms, including anergy from lack of co-stimulatory signals, clonal deletion, neglect, and suppression by regulatory T cells[7]. The precise molecular events by which these safeguards fail, allowing self-reactive lymphocytes to become activated and drive autoimmune pathology, remain unclear.

A long-standing hypothesis, first proposed by Burnet in the 1950s[1] and expanded in the 1970s[2], is that somatic mutations could allow self-reactive lymphocytes to evade suppression mechanisms (Supplementary Note 1). This idea has been modernized by others, notably

[1]Somatic Genomics Programme, Wellcome Sanger Institute, Hinxton, UK. [2]Cellular Operations, Wellcome Sanger Institute, Hinxton, UK. [3]Sequencing Operations, Wellcome Sanger Institute, Hinxton, UK. [4]Cytometry Core Facility, Wellcome Sanger Institute, Hinxton, UK. [5]Broad Institute of MIT and Harvard, Cambridge, MA, USA. [6]Department of Stem Cell and Regenerative Biology, Harvard University, Cambridge, MA, USA. [7]Department of Surgery, University of Cambridge, Cambridge, UK. [8]NIHR Cambridge Biomedical Research Centre, Cambridge Biomedical Campus, Cambridge, UK. [9]Department of Histopathology, Cambridge University Hospitals NHS Foundation Trust, Cambridge, UK. [10]Medical Research Council Metabolic Diseases Unit, Institute of Metabolic Science-Metabolic Research Laboratories, University of Cambridge, Cambridge, UK. [11]Cellular Genomics Programme, Wellcome Sanger Institute, Hinxton, UK. [12]Quotient Therapeutics Limited, Saffron Walden, UK. [13]Cambridge Genomics Laboratory, Cambridge University Hospitals NHS Foundation Trust, Cambridge, UK. [14]Department of Metabolism and Systems Science, College of Medicine and Health, University of Birmingham, Birmingham, UK. [15]These authors contributed equally: Pantelis A. Nicola, Andrew R. J. Lawson. ✉e-mail: al28@sanger.ac.uk; im3@sanger.ac.uk

Goodnow, who in 2007 proposed a polyclonal and multi-stage model of autoimmunity in which increasingly aggressive self-reactive clones could emerge through a stepwise acquisition of somatic mutations in multiple tolerance checkpoints[3]. Several studies have found lymphoma driver mutations in some autoimmune patients[8–15], including in large self-reactive clones, but strong evidence for a causal role of somatic mutations in classical polyclonal autoimmunity has remained elusive owing to the difficulty of detecting somatic mutations in polyclonal lymphocytes.

In the last few years, advances in DNA sequencing have started to transform our understanding of the somatic mutation landscape of normal tissues. Deep sequencing studies of small areas of tissue have shown that many epithelia are mosaics composed of multitudinous microscopic clones of cells carrying driver mutations[16–18]. Yet detecting driver mutations in highly polyclonal and dispersed cell types, such as lymphocytes, has remained challenging. This limitation has been overcome by duplex sequencing protocols[4,19], including our recent development of whole-exome NanoSeq[5], which enables accurate detection of somatic mutations in single molecules of DNA across all genes, permitting comprehensive driver discovery in polyclonal cell populations.

Through a combination of whole-exome NanoSeq, deep targeted NanoSeq and single-cell DNA sequencing, here we reveal an extraordinary landscape of polyclonal and multi-stage somatic evolution of lymphocytes in autoimmune thyroid disease (AITD), with some patients harbouring hundreds of B cell clones with convergent driver mutations in key immune checkpoint genes.

## Highly recurrent checkpoint mutations

AITD is one of the most common autoimmune disorders[20]. AITD encompasses distinct clinical manifestations, including Hashimoto thyroiditis, which can progress to hypothyroidism due to lymphocyte-mediated thyroid destruction, and Graves' disease, marked by thyroid-stimulating autoantibodies driving hyperthyroidism. Anti-thyroid peroxidase (TPO) and anti-thyroglobulin (TG) autoantibodies are often detectable across AITD, with anti-TPO being particularly prevalent in Hashimoto thyroiditis. Patients with AITD have an increased risk of developing other autoimmune conditions and Hashimoto thyroiditis in particular is linked to increased risk of mucosa-associated lymphoid tissue (MALT) lymphoma.

To search for somatic driver mutations in AITD, we initially obtained thyroid biopsies from three patients with a clinical diagnosis of Hashimoto thyroiditis (H1–H3) and with extensive lymphocytic infiltration, characteristic of advanced AITD (Fig. 1a). In addition to histopathological evaluation, we used targeted methylation sequencing to quantify the cell type composition of the samples[21] (Methods). This confirmed high levels of B cell (median = 21%, range = 15–34%) and T cell (median = 47%, range = 43–50%) infiltration, with a low proportion of follicular epithelial cells (median = 9%, range = 1–13%) (Fig. 1b). This was in stark contrast to healthy thyroid samples that were dominated by follicular epithelium (median = 76%, range = 59–76%).

We then carried out deep whole-exome NanoSeq on these inflamed biopsies, to an aggregate duplex coverage (dx) of 3,125 dx (770, 1,254 and 1,101 dx per patient). It is important to note that bulk NanoSeq can detect mutations in any cell type in the sample, proportionally to their frequency of in the sample. This yielded a total of 28,855 coding mutations across all cells sequenced from these three donors. To look for evidence of driver mutations among them, we used dNdScv, a $dN/dS$ method optimized for the detection of genes under positive selection from somatic data, which controls for known sources of mutation rate heterogeneity across genes[22] ($dN/dS$ refers to the normalized ratio of non-synonymous to synonymous mutation rates) (Methods). Specifically, we used a modified version of dNdScv robust to the confounding effect of off-target somatic hypermutation (SHM) in activated B cells

(dNdSshm) (Methods). This revealed strong positive selection in four genes: *TNFRSF14* (also known as *HVEM*) ($Q < 1 × 10^{-15}$), *CD274* (which encodes PD-L1) ($Q < 1 × 10^{-15}$), *TET2* ($Q = 1.2 × 10^{-6}$) and *TNFAIP3* (which encodes A20) ($Q = 9.1 × 10^{-6}$) (Fig. 1c). All four genes showed a very high enrichment of truncating mutations (nonsense mutations, essential splice site mutations and frameshifting indels), particularly *TNFRSF14* and *CD274*, revealing that somatic loss of these genes confers a strong selective advantage on the mutant cells.

Notably, we observed large numbers of clones per individual carrying different, independently acquired, driver mutations in these genes. In just these three donors (H1–H3) we found 102 non-synonymous mutations in *TNFRSF14* (72, 16 and 14 per donor), 40 in *CD274* (17, 20 and 3, respectively), 18 in *TET2* and 11 in *TNFAIP3*. Most clones were very small, evidenced by low variant allele fractions (VAFs) in *TNFRSF14* (median VAF = 0.002, maximum VAF = 0.024) and *CD274* (median VAF = 0.0006, maximum VAF = 0.019). Almost all individual clones accounted for less than 1% of cells in the biopsy, explaining why these mutations have been undetectable by conventional DNA sequencing methods. Although each mutation is only present in a small proportion of cells, cumulatively across many clones the estimated fraction of mutant cells in these biopsies reaches considerable levels, albeit with significant variation across patients. For donor H1, at least 32.7% and 5.0% of all cells bear *TNFRSF14* and *CD274* mutations, respectively (Methods). The other donors had lower, but still substantial, *TNFRSF14* (H2 = 3.0%, H3 = 5.5%) and *CD274* (H2 = 4.4%, H3 = 0.3%) mutant cell fractions. Of note, these are lower bound estimates from biopsies composed of several different cell types. We demonstrate later in the manuscript that the *TNFRSF14* and *CD274* driver mutations occur specifically in B cells, which implies that the fraction of *TNFRSF14*- or *CD274*-mutant B cells is considerably higher than the estimates above.

The polyclonality observed in these three samples is exceptionally high, with most driver mutations detected in a single DNA molecule in the bulk sample. This suggests that deeper sequencing would identify many more clones in these donors. To explore that, we performed deep targeted NanoSeq to a total duplex depth of 29,584 dx, using a 2.3-megabase bait capture panel composed of 725 genes selected for their role in immune function and lymphomagenesis or as common drivers in cancer and normal tissues (Methods). This revealed many more clones with driver mutations, yielding a total of 229 *TNFRSF14*, 125 *CD274*, 84 *TET2* and 48 *TNFAIP3* non-synonymous mutations across the exome and targeted data. Remarkably, we identified 135 different *TNFRSF14* and 59 *CD274* mutations in donor H1 alone (Fig. 1d). As 34% of *TNFRSF14* mutations were still detected in a single DNA molecule, deeper sequencing is expected to uncover additional clones.

Notably, *TNFRSF14* and *CD274* encode canonical immune checkpoints with known causal links to autoimmunity and thyroid pathology. TNFRSF14 (also known as HVEM) is a TNF family receptor expressed on B cells, T cells and other immune cells, which acts as a key regulator of immune activation and tolerance. When expressed on the surface of B cells, binding to BTLA in B or T helper cells sends repressive signals bidirectionally to suppress B and T cell activation[23]. Loss of the TNFRSF14–BTLA axis has been implicated in the breakdown of peripheral tolerance, leading to heightened B cell activation, germinal centre expansion and the development of autoimmune and inflammatory diseases such as systemic lupus erythematosus, rheumatoid arthritis and inflammatory bowel disease[23–25]. Germline mutations in *TNFRSF14* are also known to increase the risk of AITD and several other autoimmune diseases from genome-wide association studies[26].

PD-L1, encoded by *CD274*, is a critical immune checkpoint with a crucial role in maintaining peripheral tolerance and suppressing autoimmune responses. PD-L1 binds to PD-1 on T cells, delivering inhibitory signals that reduce T cell activation, cytokine production and proliferation. The loss or reduced expression of PD-L1 in B cells in the context of AITD is expected to lead to aberrant B and T cell activation against self-antigens. This is further supported by the association of germline

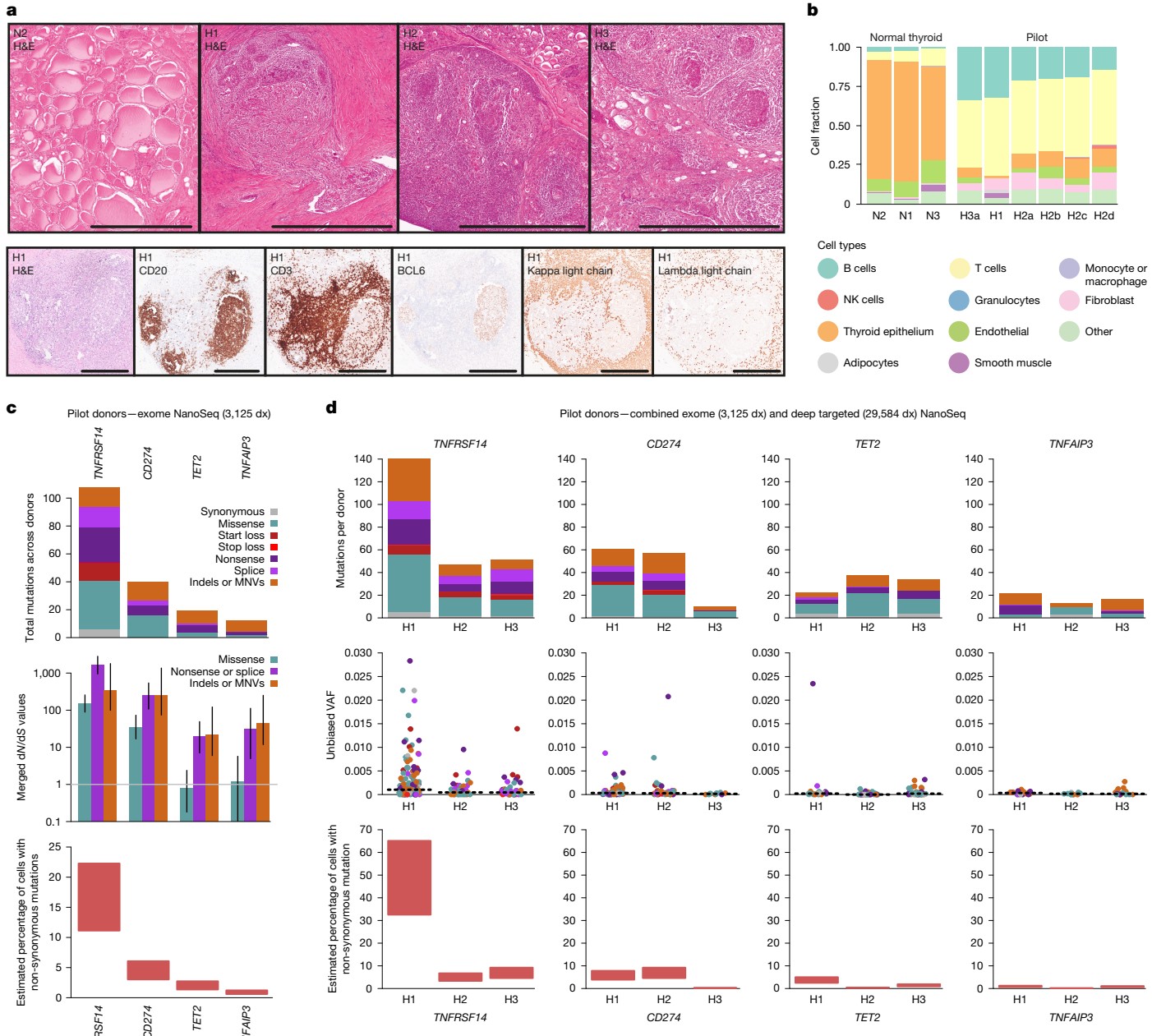

**Fig. 1 | Exome-wide driver discovery in pilot AITD cohort. a**, Representative haematoxylin and eosin (H&E)-stained sections for normal thyroid (N2) and AITD samples in the pilot cohort (H1–H3) (top row), as well as IHC (anti-CD20, anti-CD3 and anti-BCL6) and in situ hybridization (ISH) (kappa and lambda light chains) on tissue sections from donor H1 (bottom row). Kappa–lambda light chain staining of cytoplasmic messenger RNA outside of germinal centres is expected to reflect plasma cells. **b**, Cell type composition of bulk biopsies from normal thyroid and the pilot AITD cohort estimated by targeted methylation sequencing. Within each group, samples are sorted by decreasing B cell fraction. NK, natural killer. **c**, For the four genes found to be under significant positive selection ($Q < 0.01$) in exome NanoSeq data from bulk biopsies of the pilot AITD cohort, panels show: top, mutation counts per mutation consequence category; middle, d$N$/d$S$ ratios with 95% confidence intervals per mutation consequence category (horizontal line indicates neutral d$N$/d$S$ = 1); and bottom, estimated mutant cell percentages (upper and lower bounds). **d**, For the combined targeted and exome NanoSeq data from bulk biopsies of the pilot AITD cohort, mutation counts per mutation consequence category (top), unbiased VAFs for each mutation with the median indicated by the dashed black line (middle) and estimated mutant cell percentages (bottom) are shown per donor for the four genes reaching exome-wide significance ($Q < 0.01$). Scale bars, 1 mm (**a**, upper panels), 500 μm (**a**, lower panels). MNV, multi-nucleotide variant.

*CD274* mutations with AITD[27] and the fact that thyroid autoimmunity is a common adverse effect of immune checkpoint inhibitors, particularly those targeting the PD-1–PD-L1 axis[28].

Together, the known functions of *TNFRSF14* and *CD274*, the available evidence from mouse models[24,25], the genetic associations between *TNFRSF14* or *CD274* germline mutations and AITD[26,27], and the frequent development of thyroiditis following PD-L1 inhibition[28] all suggest that the highly recurrent inactivating somatic mutations observed here in both genes play a causal role in the pathogenesis of AITD.

Notably, all four driver genes are also the most frequently mutated drivers of thyroid MALT lymphomas[29], with the combination of *TNFRSF14* and *CD274* mutations being particularly common in MALT lymphomas emerging from a background of AITD[29]. Our discovery of large numbers of clones with these mutations in AITD provides a

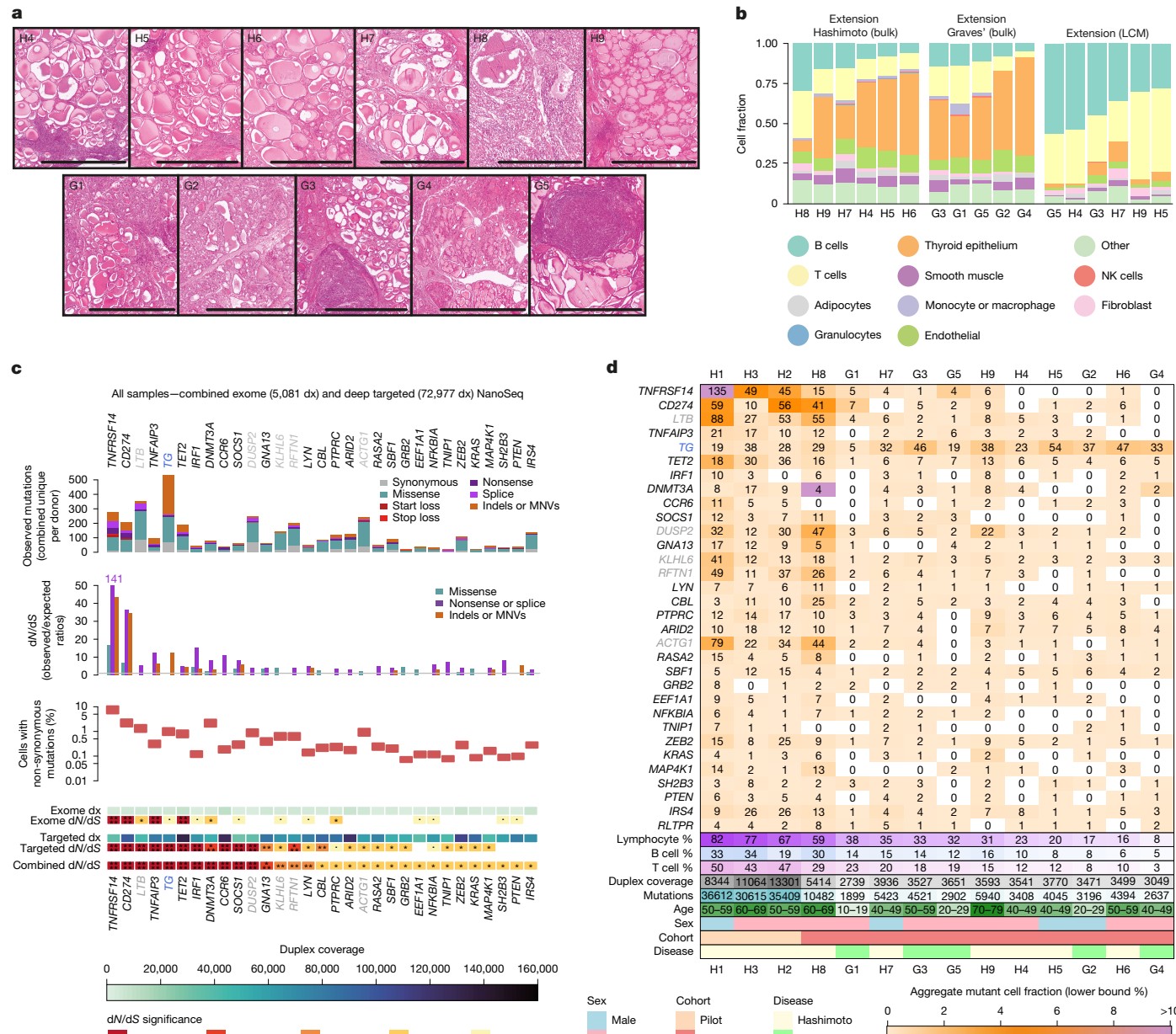

**Fig. 2 | Selection landscape across pilot and extension AITD cohorts.**
**a**, Representative H&E sections for donors in the extension cohort (*n* = 11) with clinical diagnoses of Hashimoto thyroiditis (H4–H9) and Graves' disease (G1–G5). **b**, Cell type composition of bulk and LCM biopsies from the extension AITD cohort estimated by targeted methylation sequencing. Within each group, samples are sorted by decreasing B cell fraction. **c**, For 31 genes found to be under significant positive selection (*Q* < 0.1) in the combined pilot and extension cohorts, panels show: top, mutation counts per mutation consequence category; second, d*N*/d*S* ratios per mutation consequence category (horizontal line indicates neutral d*N*/d*S* = 1; only categories with significant d*N*/d*S* ratios are shown for each gene); third, estimated mutant cell percentages (upper and lower bounds); and bottom, d*N*/d*S* significance (*Q* values were calculated with

the Benjamini–Hochberg method) and duplex coverage in the exome and targeted NanoSeq datasets separately and combined. Gene names highlighted in grey denote those reaching significance solely owing to the signal from exons affected by SHM, whereas those in blue indicate lineage defining genes with increased indel mutagenesis. **d**, Heatmap showing the number of non-synonymous mutations as text (unique counts per donor) and the lower bound mutant cell fraction (background colour, capped at 10%) per donor from combined targeted and exome NanoSeq data from bulk biopsies for 32 genes (31 significant by gene-level d*N*/d*S* and *RLTPR*, which is significant by site-level d*N*/d*S*). Columns are sorted by decreasing proportion of cells that are lymphocytes (estimated from targeted methylation data). Scale bars, 1 mm.

mechanistic explanation for the increased risk of MALT lymphoma development in these patients. This also suggests that these driver mutations occur in B cells, as we demonstrate later.

## Deep driver landscape in a larger cohort

To explore whether this phenomenon is found in a broader spectrum of patients with AITD, particularly in less advanced cases with lower

lymphocytic infiltration, we obtained snap-frozen thyroid tissue from six additional patients with clinical diagnoses of Hashimoto thyroiditis (H4–H9), and five patients with clinical diagnoses of Graves' disease (G1–G5) (Fig. 2a,b). All five donors with Graves' disease were selected based on their high levels of serum anti-TPO autoantibodies and/or the presence of substantial intra-thyroidal lymphocytic infiltration. Targeted methylation sequencing (Fig. 2b) and Xenium spatial transcriptomics (Extended Data Fig. 1a,b) provide further evidence of the

lower levels of lymphocytic infiltration and higher thyroid preservation in most samples in the extension cohort.

For six donors, we used laser capture microdissection (LCM) to isolate areas of lymphocytic infiltration and performed further whole-exome NanoSeq to an aggregate duplex coverage of 1,956 dx (Extended Data Fig. 1c), achieving a total of 5,081 dx with the pilot exome discovery dataset. We also performed targeted NanoSeq on bulk biopsies for the 11 donors in the extension cohort (43,393 dx), yielding an aggregate targeted coverage across all 14 donors of 72,977 dx. Using dNdSshm on this extended dataset identified 15 genes under positive selection ($Q < 0.01$): *TNFRSF14*, *CD274*, *LTB*, *TNFAIP3*, *TG*, *TET2*, *IRF1*, *DNMT3A*, *CCR6*, *SOCS1*, *DUSP2*, *GNA13*, *KLHL6*, *RFTN1* and *LYN* (Fig. 2c). Using a less stringent cutoff ($Q < 0.1$), a further 16 genes were identified, many of which were deemed likely to be genuine drivers given their mutation distributions and relevant immune regulatory functions: *CBL*, *PTPRC*, *ARID2*, *ACTG1*, *RASA2*, *SBF1*, *GRB2*, *EEF1A1*, *NFKBIA*, *TNIP1*, *ZEB2*, *KRAS*, *MAP4K1*, *SH2B3*, *PTEN* and *IRS4*. Of note, the signal of positive selection detected in *TNFRSF14* and *CD274* remained much stronger than in any other gene, with d$N$/d$S$ ratios for truncating mutations of 141 and 37, respectively (Fig. 2c).

Clones with mutations in immune checkpoint genes were observed across biopsies from all donors with sufficiently high lymphocyte abundance and depth of sequencing (Fig. 2d). *TNFRSF14* and *CD274* mutations were detected in 10 and 11 of the 14 donors, respectively. The few donors without detected mutations in them had low levels of lymphocytic infiltration, as confirmed by histology and targeted methylation data (Fig. 2a,b), which limits the sensitivity of the bulk sequencing approach. However, considering the extended list of genes under positive selection, we detected immune driver mutations in all donors in the cohort. Notably, *TNFRSF14*, *CD274* and *TNFAIP3* were also significant in the extension cohort alone, validating them in an independent cohort ($Q < 1 \times 10^{-15}$, $<1 \times 10^{-15}$ and $6.6 \times 10^{-4}$, respectively).

The considerable number of driver mutations detected in these genes provide information on the type of selection acting on them (Fig. 3, Extended Data Figs. 2 and 3 and Supplementary Note 2). *TNFRSF14* and *CD274* exhibited strong selection on truncating mutations, including nonsense mutations, essential splice site mutations and frameshifting indels. Both genes also showed recurrent start codon mutations (Fig. 3a), which may lead to disrupted translation of the entire protein or to N-terminal truncation, depending on whether a suitable alternative in-frame start codon occurs downstream (Supplementary Note 2). We also found strong selection of missense mutations in the cysteines involved in disulphide bridges of TNFRSF14 and CD274 (also known as PD-L1) (Fig. 3b and Extended Data Fig. 3h), expected to disrupt their extracellular folding, hindering receptor–ligand interactions. Separate selection analyses of missense mutations in the three cysteine-rich domains (CRDs) of TNFRSF14 suggest that selection primarily favours mutations disrupting the inhibitory BTLA interaction (via CRD1)[30], rather than affecting stimulation of TNFRSF14 by LIGHT (which binds CRD3)[31] (Fig. 3b and Supplementary Note 2).

Beyond *TNFRSF14* and *CD274*, other driver genes identified by the deep targeted sequencing data are noteworthy given their important immune regulatory functions and patterns of selection. A detailed discussion is available in Supplementary Note 2, but we summarize a few key genes here.

TNFAIP3 (TNF induced protein 3, also known as A20), TNIP1 (TNFAIP3-interacting protein 1) and IκBα (encoded by *NFKBIA*) are negative regulators of NF-κB signalling, suppressing inflammation and immune responses[32]. All three have been identified as genome-wide association study hits for multiple autoimmune diseases[33,34] and in our data exhibit truncating mutations throughout the gene consistent with loss of function (Extended Data Fig. 3c). CCR6 is a chemokine receptor expressed on T helper 17 cells, B cells and dendritic cells, guiding their

migration to sites of inflammation and mucosal tissues[35]. We observed clustered nonsense mutations in the final exon of *CCR6* (Fig. 3a) that cause C-terminal truncation, expected to result in reduced desensitization and internalization of CCR6 (mediated by β-arrestin) and upregulation of CCL20-mediated NF-κB signalling[36].

CBL is an E3 ubiquitin ligase involved in the regulation of several tyrosine kinase signalling pathways governing germinal centre B cell dynamics[37] and is also a prominent driver in myeloid leukaemias[38]. Mutations in *CBL* cluster in the RING finger and linker domains lead to a loss of ubiquitin ligase activity, while preserving its adaptor role in proliferative signalling cascades[39] (Fig. 3a and Extended Data Fig. 3e,h). GNA13 also regulates germinal centre B cell dynamics and exhibited a pattern of mutations similar to that observed in lymphomas[40].

*KLHL6*, *LTB*, *DUSP2*, *RFTN1* and *ACTG1* have relevant immune regulatory functions but are also targets of B cell SHM[41] (Methods and Extended Data Figs. 2c,d and 3b,d), and so the observed excess of non-synonymous mutations needs to be treated with greater caution. Of these, *KLHL6* is especially likely to be a genuine driver as the observed missense mutations within the BTB and Kelch domains have been shown to abrogate its ubiquitin ligase activity, resulting in increased NF-κB activity via reduced degradation of Roquin2, a negative regulator of TNFAIP3 (ref. 42) (Supplementary Note 2).

TG is a protein produced by thyrocytes that serves as a precursor for the synthesis of the thyroid hormones tetra-iodothyronine (T4, thyroxine) and tri-iodothyronine (T3). The mutation pattern in this gene, with a large excess of indels of length 2–5 base pairs (bp) in both exons and introns, and lack of selection on missense and truncating substitutions (d$N$/d$S \approx 1$), is characteristic of a process of transcription-associated indel hypermutation known to occur in highly transcribed lineage-defining genes[43], rather than positive selection. As expected for mutations in thyrocytes, the aggregate *TG* mutant cell fraction was high in biopsies with low lymphocytic infiltration (Fig. 2d).

To determine whether the signal of positive selection on these driver genes is specific to AITD, and not present in control non-autoimmune lymphocytes and thyroid samples, we generated deep targeted NanoSeq data from six different sets of control samples. These included: (1) sorted B memory, CD4⁺ and CD8⁺ T memory cells from peripheral blood from non-autoimmune donors over 50 years old ($n = 20$ donors; aggregate depths 19,975 dx, 23,902 dx and 17,515 dx, respectively) (Methods); (2) lymph nodes ($n = 4$; 10,740 dx); (3) spleen ($n = 30$; 21,748 dx); (4) 183 microdissections of lymphoid aggregates from colon, ileum and healthy tonsil ($n = 5$; 3,656 dx); (5) chronic tonsillitis biopsies ($n = 5$; 19,803 dx); and (6) thyroid biopsies with nodular goitres but without AITD ($n = 5$; 7,024 dx) (Extended Data Fig. 4). These data confirmed that mutations in *TNFRSF14* and *CD274* are very rare and not under significant selection in any of the control datasets above ($Q > 0.1$) (Fig. 3c,d and Extended Data Fig. 4). Interestingly, however, some of the less frequent driver genes found in AITD are under selection in normal ageing B memory cells, including the clonal haematopoiesis drivers *TET2* and *DNMT3A*, as expected, but also *TNFAIP3*, *LTB*, *DUSP2*, *KLHL6*, *SOCS1*, *GRB2*, *NFKBIA*, *TNIP1*, *MAP4K1* and *SBF1*. This suggests that these genes drive clonal expansions of B cells during normal ageing and that their presence in AITD may not be disease-specific, although they could still play a pathogenic role in the context of an autoimmune response (see Supplementary Note 2 for more details). By contrast, the strong signal of convergent selection in *TNFRSF14* and *CD274* is highly specific to AITD, despite B cells only representing a minority of cells in most AITD samples (Fig. 2d). Dividing the *TNFRSF14* and *CD274* mutant cell fractions by the proportion of B cells in AITD samples estimated by targeted methylation sequencing yields an approximate fraction of mutant B cells in AITD greater than 2% for several samples (*TNFRSF14*: over 50% in donor H1, 18% H2, 14% H3, 10% G5, 4% H8, 3.5% G1 and 2.7% H9; *CD274*: 24% H2, 13% H8, 12% H1, 9% G1, 2.7% H6 and 2.3% H9). By contrast, mutations

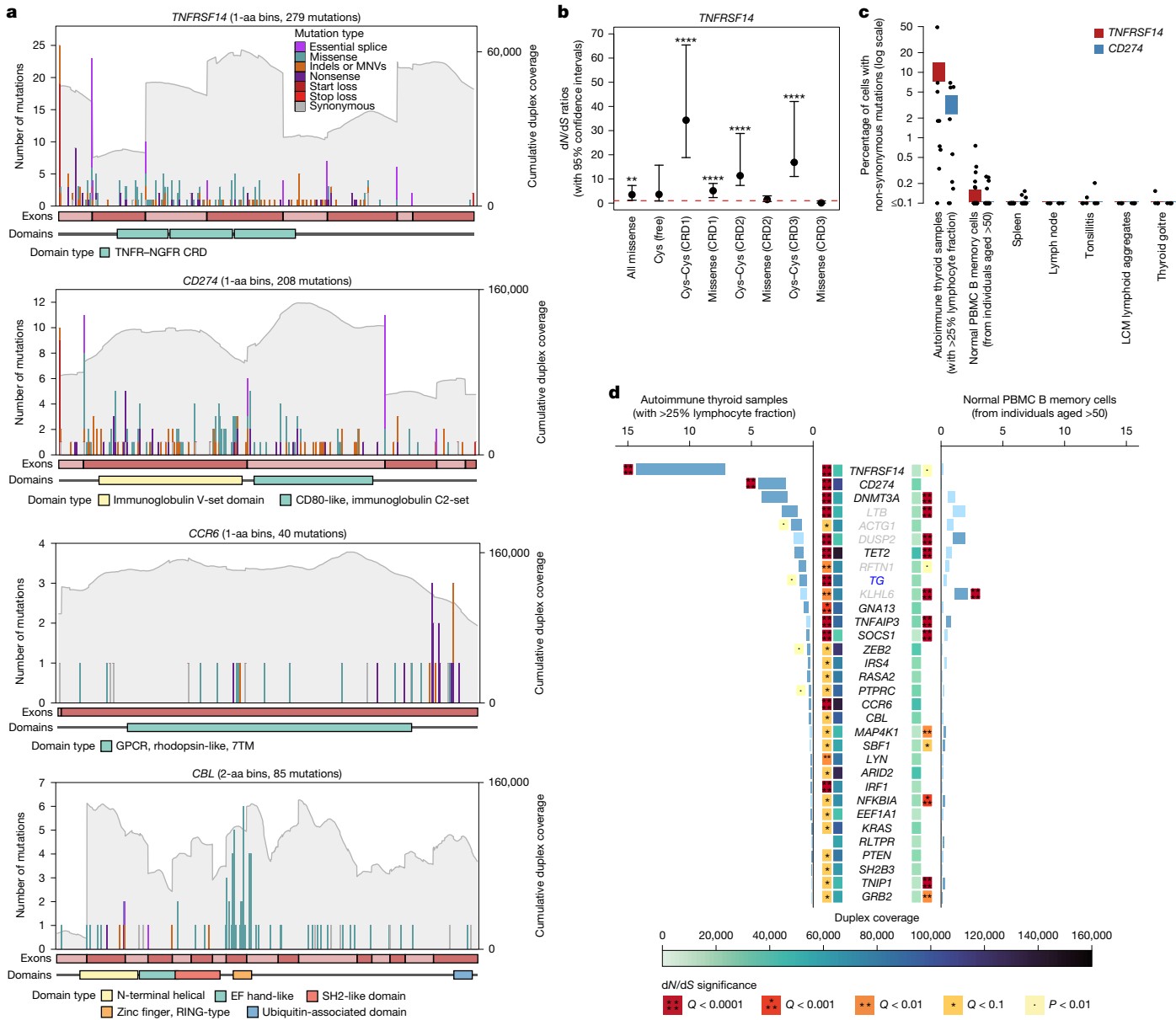

**Fig. 3 | Patterns of selection within thyroid autoimmunity driver genes.**
**a**, Mutation barplots for *TNFRSF14*, *CD274*, *CCR6* and *CBL*. The *x* axis represents coordinates along the coding sequence. Exons and protein domains are indicated along the *x* axis. The *y* axis represents the number of mutations (coloured according to mutation consequence category). Grey shading indicates cumulative duplex coverage across AITD samples. **b**, d*N*/d*S* ratios and 95% confidence intervals estimated by withingenednds for classes of missense mutations in *TNFRSF14*. Cys–Cys and Cys (free) denote cysteine residues involved in, and not involved in, disulphide bonds, respectively. The horizontal red dashed line indicates neutral d*N*/d*S* = 1. ****$Q < 10^{-4}$; ***$Q < 10^{-3}$; **$Q < 0.01$; *$Q < 0.1$; •$P < 0.01$. *Q* values were calculated with the Benjamini–Hochberg method. **c**, Mutant cell percentage estimates for *TNFRSF14* and *CD274* in AITD samples with more than 25% lymphocyte fraction and six control datasets. Upper and lower bound estimates are shown for each group (aggregated across

donors weighted by duplex coverage) and midpoint estimates for each donor (points). **d**, Comparison of mutant cell fractions and strength of selection between lymphocyte-rich AITD samples and sorted B memory cells derived from peripheral blood mononuclear cell (PBMC) samples from healthy donors aged over 50 (*n* = 20). Gene names highlighted in grey denote genes that reach significance solely owing to the signal from exons affected by SHM, whereas those in blue indicate lineage defining genes with increased indel mutagenesis. Boxes either side of gene names show duplex coverage and d*N*/d*S* significance within each sample cohort. *P* values were calculated using dndsshm and pairwisednds and corrected using the Benjamini–Hochberg method. Blue bars denote non-synonymous mutant cell percentages (upper and lower bounds), with genes showing significantly increased selection in one sample set compared with the other sample set (by pairwisednds) denoted by significance boxes. aa, amino acid.

in *TNFRSF14* are seen in less than 0.1% of most B memory cells from non-autoimmune donors and do not show signals of significant selection (*Q* > 0.1) (Fig. 3c,d).

Altogether, these data unveil an unexpectedly rich landscape of somatic evolution in autoimmune lymphocytes. In biopsies with high levels of lymphocytic infiltration, we found tens to hundreds of different mutant clones carrying inactivating mutations in *TNFRSF14* or

*CD274*, as well as a wide diversity of less frequent driver mutations in key immune regulatory genes.

## Spatial distribution of mutant clones

Given the extensive polyclonality uncovered, we next sought to study the spatial distribution of clones within biopsies. To do so, we

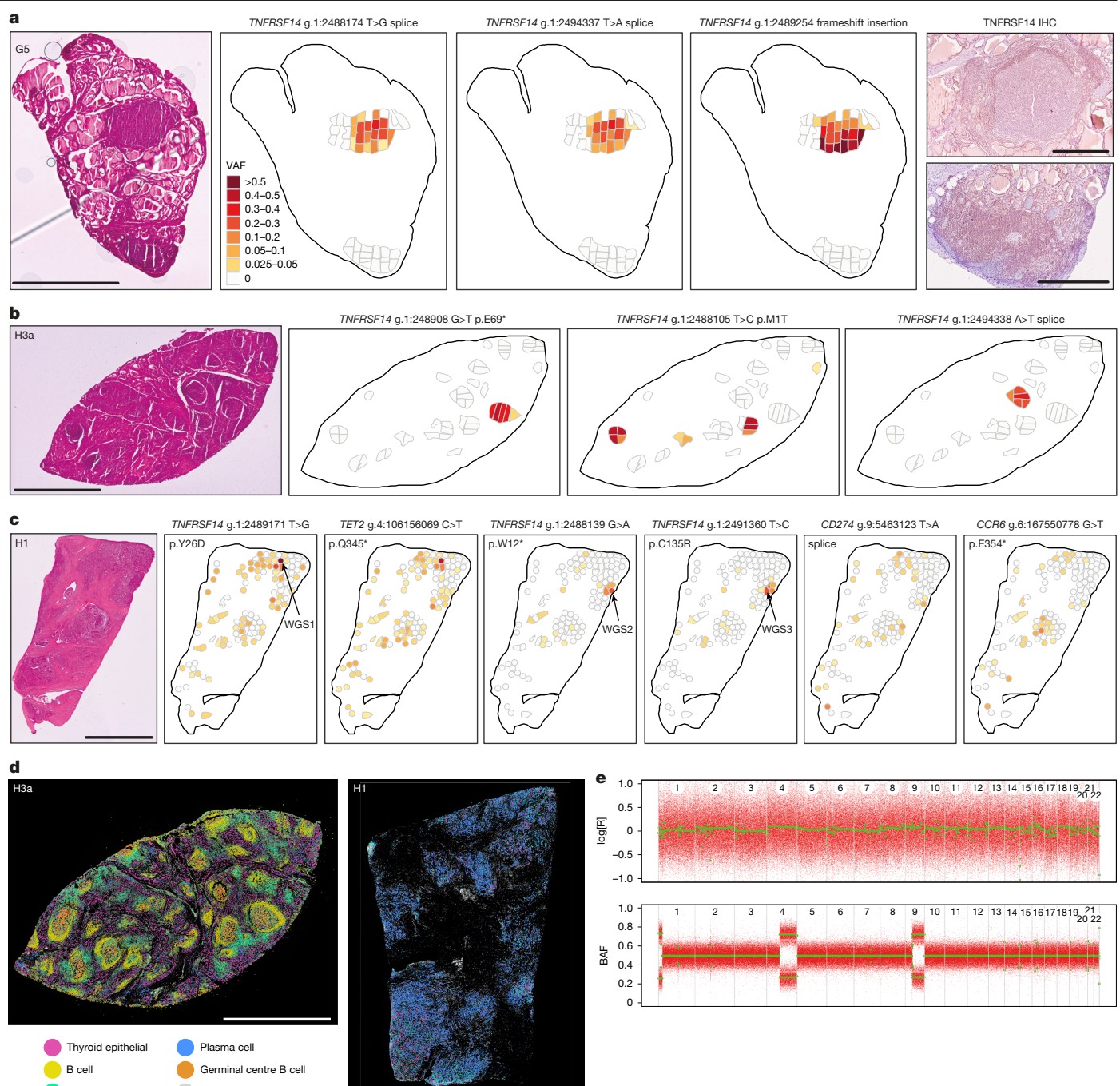

**Fig. 4 | Spatial distribution of mutant clones identified by LCM. a–c**, For three donors, G5 (**a**), H3 (**b**) and H1 (**c**), reference H&E sections are shown, followed by spatial maps of VAFs for selected mutations detected in LCMs by low-input exome sequencing. For donor G5, IHC staining for TNFRSF14 is also shown for two lymphocyte aggregates. For donor H1, three sufficiently clonal microdissections that underwent whole-genome sequencing are indicated (WGS1–3). **d**, Cell type composition inferred from spatial transcriptomics using the 380-plex Xenium Immune-Oncology Panel for donors H3 and H1. Further Xenium 5K data for seven donors (including H3 and H1) are shown in Extended Data Fig. 1, with single-marker images showing T cells, B cells and plasma cells in donor H1. **e**, log[R] and B-allele fraction (BAF) plot for a clonal microdissection (WGS1) from donor H1. Scale bars, 2 mm (**a**, left; **b**, **c**, **d**), 500 μm (**a**, right).

undertook LCM of 221 microscopic areas of lymphoid aggregates from three donors (H1, H3 and G5), followed by standard low-input whole-exome sequencing of each microdissection (54× median exome coverage; Methods). For each donor, we then genotyped all mutations detected in the NanoSeq data across all microdissections, obtaining detailed spatial maps of the distribution of driver clones across these tissue sections (Fig. 4a–c). This revealed different patterns of clonal organization in the three donors.

In G5, a donor with a clinical diagnosis of Graves', we performed 46 microdissections distributed across two large lymphoid aggregates (Fig. 4a). Three mutations from the NanoSeq data reached high VAFs in most microdissections from one aggregate, suggesting that these mutations occurred in lymphocytes. The high VAFs of these mutations and their co-occurrence across microdissections revealed the existence of one clone carrying two independent essential splice site mutations in *TNFRSF14* and a separate adjacent clone carrying a

frameshift indel reaching VAFs greater than 50% in several microdissections. Copy number analyses confirmed that the *TNFRSF14* locus (on chromosome 1p) was affected by copy-neutral loss of heterozygosity (CN-LOH) (Extended Data Fig. 5a). Together, these data suggest that this large lymphoid aggregate in G5 was composed of two large mutant clones, each with biallelic loss of *TNFRSF14*. Immunohistochemistry (IHC) confirmed loss of TNFRSF14 expression in this large lymphoid aggregate (Fig. 4a). Contrasting this, a second large lymphoid aggregate harboured no detectable driver event and normal levels of TNFRSF14 expression.

In H3, a donor with multiple intra-thyroidal lymphoid aggregates resembling germinal centres (Extended Data Fig. 1a,b), the spatial distribution of driver mutations revealed that individual germinal centres were frequently dominated by a single clone (or a few clones) (Fig. 4b). Independent start loss mutations were observed across multiple aggregates in this donor. Biallelic loss of *TNFRSF14* by CN-LOH or compound heterozygosity was observed in several germinal centres (Extended Data Fig. 5b). Multiplexed immunofluorescence and IHC for CD3, CD19 and TNFRSF14 in this donor suggest that these mutations occur specifically in B cells (Extended Data Fig. 5c–e), which is more definitively demonstrated by single-nucleus sequencing below. Xenium spatial transcriptomics confirmed that these clonal lymphoid aggregates are dominated by germinal centre B cells (Fig. 4d).

Finally, a different pattern was observed in H1, a donor with florid thyroiditis and the highest frequency of *TNFRSF14* mutations (Fig. 4c). Histopathological assessment showed diffuse lymphocytic infiltration and fibrosis across the tissue section (Fig. 1a and Extended Data Fig. 1a,b). Unlike the other two donors, most clones were dispersed throughout the tissue section. Xenium data from this donor suggest that the most common B-lineage cell type in this sample is plasma cells (Fig. 4d). Despite the widespread clonal mixing, three microdissections in this donor were almost clonal and they were subject to whole-genome sequencing to study the extent of copy number and structural changes in mutant B cell clones, as these events are not detected by NanoSeq (WGS1–3; Fig. 4c). Notably, one clone (WGS1) had acquired at least four driver mutations, with biallelic losses of *TNFRSF14* and *TET2*, as well as CN-LOH of chromosome 9q of uncertain significance (Fig. 4e).

Altogether, these data demonstrate significant inter-patient differences in clonal spreading and mixing, highlight the existence of individual clones of B cells carrying multiple driver mutations and reveal that ectopic germinal centres can be composed of a single mutant clone. To further investigate the frequency of multiple hits in individual lymphocytes, and to better assign driver mutations to B or T cells, we then performed single-nucleus DNA sequencing.

## Single-nucleus whole-genome sequencing

Recent advances in single-cell DNA sequencing, notably primary template-directed amplification (PTA), now enable the study of somatic mutations in individual cells with higher accuracy and sensitivity[44]. We applied PTA to 112 single nuclei from the most florid AITD donor (H1), using the ResolveOME DNA + RNA protocol (Methods). Deep targeted sequencing on 725 genes was performed on all 112 nuclei (314× median coverage) and whole-genome sequencing was performed on 86 nuclei (18× median coverage; Fig. 5a–e and Extended Data Figs. 6 and 7). Quality control analyses revealed good coverage across the genome, with 80% of sites having more than 10× coverage and with low allelic dropout (approximately 8.4%). Of the 112 nuclei sequenced, 66 were identified as B cells and 38 as T cells based on analyses of their B cell receptor (BCR) and T cell receptor sequences (Methods and Extended Data Figs. 6 and 7).

Using the deep catalogue of driver mutations detected in this donor with NanoSeq, we genotyped all known mutant sites across all nuclei. Driver mutations were found only in the 66 B cells (none in T cells or other cell types), including 31 non-synonymous *TNFRSF14* mutations

in 41 cells (implying that approximately 62% of B cells are *TNFRSF14*-mutant in donor H1), 7 *CD274* mutations in 8 cells and a *TET2* Q345* mutation in 6 cells (Fig. 5a). All of the mutant B cells showed evidence of SHM, confirming them as activated B cells that have undergone a germinal centre reaction (Extended Data Figs. 6 and 7). Class switch recombination was detected in 90% (37 of 41) of *TNFRSF14*-mutant B cells, suggesting that these are plasma or B memory cells (Extended Data Figs. 6 and 7). Reconstruction of the BCR heavy and light chain nucleotide sequences for these driver-mutated B cells revealed a highly polyclonal BCR repertoire (Extended Data Fig. 7 and Methods). Phylogenetic analyses using whole-genome sequencing data then confirmed that most mutant B cell lineages share no more than a few mutations, consistent with them being entirely independent lineages since early embryogenesis (Fig. 5c). To further demonstrate that *TNFRSF14* and *CD274* mutations occur in B cells beyond donor H1, we generated targeted PTA data from 180 cells from donors H2 and H8 (for a total 292 cells from 3 donors). These data support the original conclusions from donor H1, identifying 13 further single cells with *TNFRSF14* and/or *CD274* mutations, all of which show BCR recombination and class switch recombination (plasma or B memory cells) (Extended Data Fig. 8).

The single-nucleus whole-genome sequencing in donor H1 also allowed us to explore the extent of copy number changes. This revealed widespread biallelic loss of *TNFRSF14*. Of the 41 B cells carrying *TNFRSF14* point mutations or indels, 30 had lost the second allele through CN-LOH and 8 had two independent point mutations or indels consistent with compound heterozygosity. The remaining three nuclei (12, 23 and 41) all had high mutant VAFs and suggestive evidence of focal loss of heterozygosity (LOH) events (Extended Data Fig. 9a). Genome-wide analysis of the regions of CN-LOH affecting *TNFRSF14* revealed a wide diversity of breakpoints across mutant cells affecting the terminal 1p region of chromosome 1 (1pter) (Fig. 5b), consistent with them being independent events, as implied by the genome-wide phylogenies. Beyond the frequent CN-LOH events affecting *TNFRSF14*, we also found structural variants affecting other genes, most notably *CD274* in six B cells (Fig. 5a and Extended Data Figs. 9 and 10). Finally, we also performed mutational signature analyses on the whole-genome PTA data from donor H1. Among other findings (Supplementary Note 3), this revealed that B cells with driver mutations had significantly more mutations in their B cell receptors as well as higher rates of genome-wide signatures of SHM (Fig. 5e and Extended Data Fig. 11a–d). This increase was more prominent in cells with *TNFRSF14* mutations. This suggests that driver-mutant B cells may undergo more rounds of SHM within germinal centres.

Notably, these data also revealed multiple examples of single B cells with several driver mutations (Fig. 5a). Of the 41 B cells with *TNFRSF14* mutations, 8 also had point mutations, indels or structural variants affecting *CD274*. Other *TNFRSF14*-mutant B cells had further mutations in *TET2*, *BRAF*, *PIK3CD*, *PLCG2*, *KLHL6* and *CXCR3*. Of particular interest was a clade of six cells. Their pattern of phylogenetic diversification provides a remarkable example of multistep evolution of B cells in this patient with AITD (Fig. 5d). A *TET2* Q345* mutation was acquired first, before B cell differentiation and V(D)J rearrangement, probably in a bone marrow stem or progenitor cell and probably in the first decade of life (in a branch with 10–174 mutations since conception[45,46]). This led to at least two different B cell lineages with different V(D)J recombination events and independent biallelic losses of *TNFRSF14*: (1) *TNFRSF14* Y83N followed by two separate 1pter CN-LOH events in nuclei 5 and 6; and (2) *TNFRSF14* Y26D followed by 1pter CN-LOH and 9q CN-LOH in nuclei 1 to 4. Each of nuclei 1 to 4 subsequently acquired further driver mutations and/or copy number changes, including independent biallelic losses of *CD274* in nuclei 2 and 4 (Fig. 5d and Extended Data Figs. 9 and 10). One of the clonal microdissections identified in the spatial map of donor H1 belonged to this subclade (Fig. 4d) and exhibited another independent driver event: CN-LOH of chromosome 4q (resulting in biallelic loss of *TET2*). It is interesting to note that some clones seem

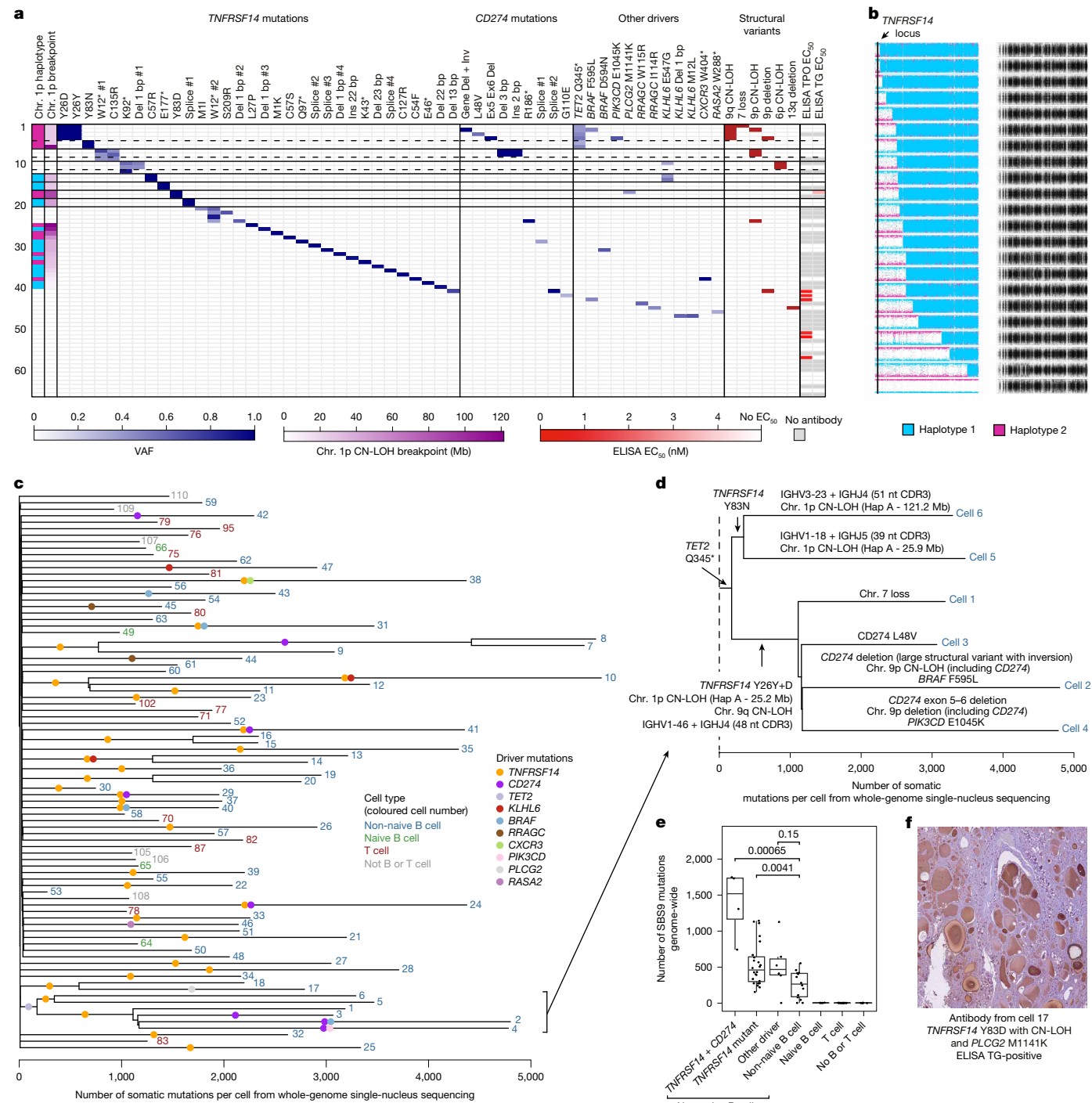

**Fig. 5 | Single-nucleus sequencing from donor H1. a**, Heatmap of driver mutations in 66 B cells from donor H1 characterized by single-nucleus sequencing. From left to right, sections denote: the haplotype and breakpoint position for 1p CN-LOH events; VAFs for mutations in *TNFRSF14*, *CD274* and selected other genes; and half-maximum effective concentration ($EC_{50}$) values for TG and TPO ELISA using recombinant antibodies generated from reconstructed BCR sequences. Horizontal lines demarcate clades (or sub-clades for dashed lines) of related cells. The full heatmap for all 112 nuclei from donor H1 (including V(D)J recombination events) is shown in Extended Data Fig. 7. Heatmaps for single-nucleus sequencing data from donors H2 and H8 are shown in Extended Data Fig. 8. Chr., chromosome; Del, deletion; Inv, inversion. **b**, B-allele fraction plots showing the retained haplotype and breakpoint position for 22 independent CN-LOH events on chromosome 1p. Single nucleotide polymorphisms on 1p were phased into haplotypes using a CN-LOH event that spanned the entire chromosomal arm. The vertical line denotes the location of *TNFRSF14*. **c**, Dendrogram of mutations called from whole-genome sequencing of 86 single nuclei from donor H1. Branch length is scaled by mutation burden. **d**, Dendrogram showing independent acquisition of driver mutations and V(D)J recombination events in a clade of related nuclei (1–6). **e**, Number of mutations attributed to SBS9, which is associated with SHM, from single-nucleus whole-genome sequencing across cell types. *P* values from two-sided Wilcoxon tests comparing mutant non-naive B cells with wild-type non-naive B cells. Only one cell per clade was selected. **f**, IHC of normal thyroid (N1) using the recombinant antibody generated from the reconstructed BCR sequence of cell 17 (found to bind TG by ELISA) on a rabbit IgG backbone as a primary antibody. Mb, megabase; nt, nucleotide.

to have acquired their first driver mutations many years before the diagnosis of Hashimoto thyroiditis, consistent with a possible role of these mutations in disease development. The PTA data from patients H2 and H8 also revealed that most cells with *TNFRSF14* or *CD274* mutations in these patients carried more than one driver mutation, including cells with biallelic losses of both *CD274* and *TNFRSF14*, and other driver combinations (Extended Data Fig. 8).

The single-cell data also allowed us to test whether some of the clones were self-reactive. Previous evidence suggests that a substantial fraction of germinal centre B cells in AITD recognize TPO and TG[47]. From donor H1, we were able to reconstruct complete heavy and light chain BCR sequences from 32 B cells. We then generated recombinant antibodies from these sequences and each of the antibodies was tested for binding to two classical autoantigens, TPO and TG by enzyme-linked immunosorbent assays (ELISAs). This identified at least seven B cell clones as self-reactive against either TPO ($n = 6$) or TG ($n = 1$). Of these, four carried driver mutations: three in *TNFRSF14* and/or *CD274*, and one in *BRAF* (Fig. 5a and Extended Data Fig. 11e), suggesting that a considerable fraction of driver-mutant clones are self-reactive against TPO or TG. To explore whether some clones may be recognizing other self-antigens, we also ordered eight of the highest-confidence BCR sequences in a rabbit IgG backbone, to test whether we could detect binding anywhere in thyroid tissue sections using them as primary antibodies for IHC (Methods). Of these eight antibodies, five showed clear binding in a control thyroid sample, including four antibodies that were positive by ELISA against TPO or TG, and an extra antibody that showed weak TPO binding by ELISA and had been considered ELISA-negative (Fig. 5f and Extended Data Fig. 11e,f). Considering that these tests are not sensitive to all possible autoantigens, and that some BCR sequences may not have been accurately reconstructed, these data show that a considerable fraction of mutant (and wild-type) B cell clones in the sample are self-reactive, strengthening the argument for a possible pathogenic role in AITD.

In summary, the single-cell data confirmed that the highly convergent immune checkpoint mutations detected in bulk by NanoSeq were acquired by many independent B cell clones, each with different BCRs (Fig. 5a and Extended Data Fig. 7). The data also provide multiple examples of B cells acquiring several driver mutations, including widespread biallelic loss of *TNFRSF14* and/or *CD274*, and some cells carrying as many as five or six driver mutations. The phylogenetic trees and the order of events relative to V(D)J recombination and SHM events further confirm that these mutations were accrued sequentially, some over long periods of time. The diversity of drivers across clones also suggests that there is plasticity in the ordering of driver acquisition within clones. Altogether, these data provide evidence of strong selection driving large numbers of clonally independent B cells to escape peripheral tolerance constraints during the pathogenesis of AITD. These data provide support for a polyclonal cascade model of autoimmune pathogenesis, in which possibly one or a few initial self-reactive clones with tolerance-escape driver mutations initiate a weak localized autoimmune response, which progressively exacerbates through the incorporation of new clones and the acquisition of further driver mutations within established mutant clones.

## Discussion

The hypothesis that somatic mutations in lymphocytes may drive or contribute to autoimmune disease has been speculated about for decades, but it has remained difficult to study owing to the challenge of detecting somatic mutations in polyclonal cell types. Using single-molecule and single-cell sequencing methods, we have uncovered a remarkable landscape of somatic evolution in autoimmune lymphocytes. In the most lymphocyte-infiltrated samples, we found tens to hundreds of B cell clones carrying convergent immune checkpoint mutations, widespread biallelic loss of *TNFRSF14* and *CD274*, and some B cells carrying as many as four to six driver mutations.

Studies in other diseases and in larger cohorts, as well as functional and mechanistic studies in animal models, will be needed to fully understand the extent and impact of this phenomenon. However, the landscape unveiled here seems consistent with Burnet's and Goodnow's somatic evolution hypotheses, in which multiple somatic driver mutations in a single clone or in cooperating clones may enable self-reactive clones to bypass sequential tolerance checkpoints and cause overt autoimmune disease[2,3]. Unlike tumours, which are predominantly monoclonal in origin, the patterns observed in this study suggest a highly polyclonal contribution of somatic mutations to autoimmune disease, in which many mutant clones cooperate to mount an autoimmune response. The extreme convergence observed here, with tens to hundreds of clones per donor, is unlikely to emerge at once; instead, our observations seem more consistent with a polyclonal cascade in which the emergence of one or a few initial self-reactive mutant clones facilitates the progressive incorporation of new mutant clones and the acquisition of further driver mutations in established clones over time, presumably leading to the consolidation and exacerbation of an autoimmune response. This is further supported by multiple instances of a stepwise acquisition of driver mutations in our data. Such a somatic evolution model of autoimmunity is also compatible with the accepted pathogenic roles of environmental triggers and germline mutations in autoimmune disease. For example, in some instances an infection could cause the emergence of a lymphocyte clone showing cross-reactivity with a self-antigen (molecular mimicry). Whereas this clone should be suppressed once the infection is cleared, cells with somatic mutations in tolerance checkpoints rendering them less sensitive to peripheral tolerance suppression may escape, potentially initiating a slow polyclonal cascade. This process would also be more likely in individuals with germline predisposing mutations, including HLA alleles facilitating self-antigen presentation or germline mutations weakening central or peripheral tolerance.

If correct, a polyclonal cascade model of autoimmunity driven by somatic mutations could explain several clinical phenomena. First, the observation of large numbers of clones with thyroid MALT lymphoma driver mutations may explain its increased risk in patients with AITD[29]. Second, a polyclonal cascade model is consistent with a long subclinical phase of autoimmune disease development, which may explain the frequent observation of low-titre autoantibodies in apparently healthy individuals, some of which are risk factors for future autoimmune disease[15,48]. Third, a polyclonal cascade may explain the phenomenon of epitope spreading. Finally, the observation of frequent driver mutations in activated B cells, if it extends to other autoimmune diseases, may help explain the success of deep B cell depletion therapies against some autoimmune diseases[49]. In this context, we note that AITD is often considered a T cell-mediated disorder[50], partly driven by dysregulation of T helper cells and recruitment of cytotoxic T cells. The *TNFRSF14* and *CD274* mutations observed in B cells could explain or significantly contribute to T helper cell dysregulation, derepressing B cells to act as unregulated antigen-presenting cells and chronically stimulating TPO- and TG-specific T cells, as well as generating autoantibody-producing plasma cells. This exemplifies how driver mutations in B cells may contribute to classic polyclonal autoimmune diseases, beyond the previous observation of drivers in large B cell clones producing pathological autoantibodies in some extreme individuals[14,15].

Important questions remain unanswered. In-depth studies will be needed to determine the driver landscape of other autoimmune diseases and at different stages of disease development. Studies with better cell type resolution will be needed to explore the extent and role of driver mutations in other immune cell types, including T cells and macrophages. Most importantly, although our results provide a plausible mechanism for how somatic mutations could contribute to the initiation of an autoimmune disease, they are not definitive evidence of causality. It remains unclear whether this mechanism is necessary or even sufficient for initiation in most patients, or whether it mainly

acts within an already established disease and may contribute only to its consolidation or exacerbation.

Altogether, our results reveal a fascinating landscape of somatic evolution in thyroid autoimmunity, with tens to hundreds of clones carrying immune checkpoint mutations in some donors and multiple driver mutations per cell in some clones, providing suggestive evidence for a long-hypothesized role of somatic mutations in autoimmune disease. These results emphasize how little we know about the extent of somatic evolution in autoimmunity and in other diseases, and they provide an example of a polyclonal contribution of somatic mutations to a complex chronic disease. Fuelled by advances in single-molecule and single-cell DNA sequencing, the next few years promise to transform our understanding of the role of somatic mutations in autoimmunity and beyond.

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

## Methods

### Ethics statement and sample collection

Snap-frozen thyroid tissue from the thyroidectomies of three patients (H1–H3) with histopathological features of Hashimoto thyroiditis were obtained from Cambridge University Hospitals (CUH) Tissue Bank (Cambridge, UK). One donor (H1) underwent further investigations through the regional Haematopathology & Oncology Diagnostic Service (HODS), ruling out a diagnosis of lymphoma. Further snap-frozen thyroid biopsies were obtained from five donors with nodular goitres, without a history of thyroid autoimmunity, as extra controls. All patients provided informed consent for their tissue to be used for research purposes. Demographic details were provided from available clinical records. The use of these samples was approved by the South West – Central Bristol Research Ethics Committee (REC 19/SW/0031).

Snap-frozen post-mortem thyroid biopsies from six further donors with a clinical diagnosis of Hashimoto thyroiditis (H4–H9) and from three healthy controls (N1–N3) were purchased from a commercial supplier (AMS Biotechnology Europe, AMSBIO). Biopsies of healthy lymph node (4 donors), spleen (3 donors), sigmoid colon (3 donors), ileum (1 donor) and tonsil (1 donor) from post-mortem controls, without any history of autoimmune disease, were acquired from the same source, as were chronically inflamed tonsil samples obtained from the tonsillectomies of five donors and PBMCs from individuals with no reported history of autoimmune disease (AMS Biotechnology Europe, AMSBIO). All patients provided informed consent for their tissue to be used for commercial and research purposes. The pseudonymized clinical metadata were limited to the cause of death and co-morbidities. The use of these samples was approved by the London – Surrey Research Ethics Committee (REC 17/LO/1801).

Snap-frozen thyroid tissue from the thyroidectomies of five donors with clinical diagnoses of Graves' disease (G1–G5) and anti-TPO antibodies and/or histopathological evidence of lymphocytic thyroiditis was obtained from Addenbrooke's Hospital (Cambridge, UK). One donor (G3) had lymphoma ruled out through the regional HODS. All patients provided informed consent for their tissue to be used for research purposes. The use of these samples was approved by the South Central – Oxford B Research Ethics Committee (REC 21/SC/0158).

Spleen biopsies were obtained from 27 organ transplant donors following circulatory or brainstem death and immediately fixed in PAXgene (PreAnalytiX). Informed consent for participation in research and publication of data was provided by the donor's family, as part of the Cambridge Biorepository for Translational Medicine programme. The use of these samples was approved by the East of England – Cambridge South Research Ethics Committee (REC 15/EE/0152).

All snap-frozen tissue was stored at −70 °C. PAXgene-fixed tissue (PreAnalytiX) was stored at −20 °C. Further metadata for donors can be found in Supplementary Table 1. No statistics were used to predetermine sample size, nor did we use any blinding or randomization.

### Tissue processing and sectioning

Fresh and snap-frozen tissue underwent formalin-free fixation with PAXgene (PreAnalytiX), tissue processing and paraffin embedding as described previously[51]. Three types of sectioning were performed on PAXgene-fixed paraffin-embedded (PFPE) tissue using an Accu-Cut SRM 200 microtome (Sakura Finetek).

First, each sample had 5 µm sections mounted onto Superfrost Plus glass microscope slides (VWR International) for histopathology assessment. Second, 30–60 sequential 5 µm sections ('ribbons') were collected and deparaffinized with sequential xylene and ethanol 100% washes as previously described[52]. Third, for samples undergoing LCM, sections with a thickness of 16 µm were mounted onto polyethylene naphtalate (PEN) membrane glass slides (Leica Microsystems) as previously described[51]. Slides were stained with Gill's haematoxylin and eosin (Leica Microsystems) according to a published protocol[51].

Control datasets generated for spleen, lymph node, tonsillitis and thyroid nodular goitre samples were generated using the same tissue processing steps described above with sequential sectioning of bulk biopsies. Laser microdissected lymphoid aggregates from colon, tonsil and ileum were mounted on PEN membrane glass slides and microdissected. Cryopreserved PBMCs were sorted into B memory ($CD3^-CD19^+CD20^+CD27^+$), $CD4^+$ memory T ($CD3^+CD4^+CD8^-CD45RA^-$) and $CD8^+$ memory T cells ($CD3^+CD4^-CD8^+CD45RA^-$) using a Bigfoot spectral cell sorter (Invitrogen, Thermo Fisher Scientific) (Lawson et al., in preparation).

Haematoxylin and eosin (H&E) staining on formalin-fixed paraffin-embedded (FFPE) diagnostic tissue was performed by a United Kingdom Accreditation Service (UKAS)-accredited National Health Service (NHS) histopathology lab at CUH using the Epredi Gemin AS Automated Slide Stainer (Thermo Fisher Scientific). The H&E staining protocol proceeded as follows: two xylene washes (1 min each), three washes with industrial denatured alcohol (1 min each), one tap water wash (1 min), staining with acidified Harris' haematoxylin (1 min, RBA-4205-00A, CellPath), one tap water wash (1 min), one acid alcohol 0.15% wash (25 s), one tap water wash (1 min), Scott's tap water substitute (1 min), one tap water wash (1 min), staining with aqueous eosin Y (RBC-0100-00A, CellPath), one tap water wash (15 s), three washes with industrial denatured alcohol (15 s each), two xylene washes (15 s each). Slides were coverslipped with HistoCore SPECTRA CV X1, a xylene-based mounting solution (3801733, Leica Biosystems). Images were acquired using a NanoZoomer slide scanner (Hamamatsu Photonics).

### IHC and immunofluorescence

IHC and ISH on FFPE diagnostic tissue were performed by a UKAS-accredited NHS histopathology lab at CUH using the BOND-III fully automated IHC and ISH staining system (Leica Biosystems). Tissue-mounted microscope slides first underwent dewaxing, washing, epitope retrieval and peroxide blocking. For IHC, the following primary antibodies were applied to sequential sections: anti-CD20, anti-CD3 and anti-BCL6 (PA0359, PA0122 and PA0204, Leica Biosystems). For ISH, fluorescein-conjugated oligonucleotide probes targeting kappa and lambda light chain mRNA (PB0645 and PB0669, Leica Biosystems) were used with anti-fluorescein antibodies (AR0222, Leica Biosystems). Both IHC and ISH used the Bond Polymer Refine Detection System (DS9800, Leica Biosystems), containing the 3,3′-diaminobenzidine (DAB) chromogen, with a DAB enhancer (AR9432, Leica Biosystems) for detection and counterstaining with haematoxylin. Slides were coverslipped with HistoCore SPECTRA CV X1, a xylene-based mounting solution (3801733, Leica Biosystems). Images were acquired using a NanoZoomer slide scanner (Hamamatsu Photonics).

IHC was also performed manually on 5 µm PFPE thyroid sections. Following deparaffinization and rehydration, antigen retrieval was performed by heating sections in Antigen Retrieval Buffer (ab93678, Abcam) for 10 min. Endogenous peroxidase activity was quenched with a peroxidase/methanol solution, and non-specific binding was blocked using 3% normal goat serum. The tissue was then incubated overnight at 4 °C with anti-CD19 primary antibody (ab270715, Abcam) diluted at 1:200, anti-CD3 primary antibody (ab11089, Abcam) or anti-TNFRSF14 primary antibody (ab314494, Abcam) diluted at 1:1,000. VECTASTAIN Elite ABC-HRP Kit (PK-6101, Vector Laboratories) and ImmPACT DAB Substrate Kit (SK-4105, Vector Laboratories) were used for detection. Sections were counterstained with haematoxylin QS (H-3404-100, Vector Laboratories), dehydrated and coverslipped for histopathological review. Images were acquired using a NanoZoomer slide scanner (Hamamatsu Photonics).

IHC with the eight recombinant antibodies on a rabbit IgG1 backbone was performed as described above. The recombinant antibodies were supplied by Biointron Biological at 0.5 mg ml$^{-1}$ and diluted at 1:250. Further positive control antibodies, anti-TG (ab156008, Abcam) and anti-TPO (ab109383, Abcam), were diluted at 1:500.

Immunofluorescence staining was performed following a previously described protocol[53] with several modifications. PFPE thyroid tissue was sectioned to a thickness of 5 μm. Following deparaffinization and rehydration, antigen retrieval was performed by incubating the slides in Antigen Retrieval Buffer (ab93678, Abcam) for 10 min. The sections were then permeabilized with PBS solution containing gelatin and 0.25% Triton X-100 (Sigma-Aldrich). The tissue was blocked using 5% bovine serum albumin. Subsequently, the tissue sections were incubated overnight at 4 °C with primary antibodies against CD19 (1:1,000 dilution; 14-0194-82, Invitrogen, Thermo Fisher Scientific) and TNFRSF14 (1:1,000 dilution; ab314494, Abcam). The following day, sections were incubated with the corresponding secondary antibodies: Anti-Rat IgG H&L (Alexa Fluor 568, ab175476, Abcam) and Anti-Rabbit IgG H&L (Alexa Fluor 647, ab150083, Abcam). The slides were coverslipped using ProLong Gold Antifade Mountant with DAPI (P36931, Thermo Fisher Scientific). Images were acquired with a Leica SP8 multi-photon digital light sheet Lightning confocal microscope, running the Leica LASX software package for image capture and analysis (Leica Microsystems).

### LCM and DNA extraction
Sections mounted onto PEN membrane slides underwent LCM with the Leica LMD7 laser capture microscope (Leica Microsystems). Micro-dissections undergoing low-input whole-exome sequencing were cut into semi-skirted Eppendorf twin.tec LoBind 96-well PCR plates (0030129504, Eppendorf). Cell lysis was completed with the Arcturus PicoPure DNA kit (KIT0103, Thermo Fisher Scientific) as previously published[51].

Deparaffinized tissue ribbons and microdissections undergoing targeted and exome NanoSeq underwent DNA extraction with the QIAamp DNA Micro kit (56304, QIAGEN) as previously described[52]. An aliquot of this extracted DNA (10–50 ng) was submitted for targeted methylation sequencing, as previously described[5].

### Single-nucleus isolation and sorting
Nuclei dissociation was performed using a modified Slide-tags protocol, a single-nucleus barcoding technique developed for multimodal spatial genomics[54]. The main deviation from this protocol was the absence of mounting tissue sections on the proprietary pucks necessary for spatial mapping. Single nuclei were sorted using a Bigfoot spectral cell sorter (Invitrogen, Thermo Fisher Scientific) based on propidium iodide positivity (with 561 nm laser excitation and propidium iodide fluorescence emission collected using a 625/15 bandpass filter) and size distribution (Extended Data Fig. 6a) into each well of a skirted Eppendorf twin.tec LoBind 96-well PCR plate (0030129512, Eppendorf) filled with 3 μl of cell buffer (BioSkryb Genomics) or Eppendorf twin.tec 384-well PCR plates (0030113578, Eppendorf). Plates were frozen at −70 °C before library preparation.

### Hybridization panel design
Hybridization panels were manufactured by Twist Bioscience. In addition to an exome panel used with NanoSeq and low-input DNA sequencing libraries, a 725-gene panel was designed incorporating driver genes implicated in epithelial cancers, lymphomas and normal tissues, as well as regions of the genome relevant to immune function such as V(D)J loci (Supplementary Table 2). For targeted methylation sequencing, a hybridization panel was constructed based on the discriminatory methylation marks identified across 40 cell types in a published methylation atlas[21] (Supplementary Table 3).

### Sequencing library preparation
Purified DNA underwent NanoSeq library preparation as previously described[5]. Briefly, this consists of DNA fragmentation through sonication, blunting by mung bean exonuclease, A-tailing in the presence of ddBTPs and adaptor ligation. Targeted and exome hybridization was performed as previously described[5]. Low-input whole-genome and whole-exome sequencing libraries for microdissections were prepared according to established protocols[51]. Targeted methylation sequencing was performed as previously described[5] using an updated panel (see 'Methylation sequencing' section).

Single nuclei underwent PTA using the ResolveOME or ResolveDNA kit (BioSkryb Genomics) according to the manufacturer's protocol and using the proprietary reagents provided. Briefly, this protocol includes reverse transcription, nuclear lysis, whole-genome amplification, bead-based DNA and RNA separation, and sequencing library preparation. All nuclei underwent bait capture with the Twist Bioscience Target Enrichment Standard Hybridization Protocol (102033, Twist Bioscience) and the same 725-gene panel used with the NanoSeq libraries. Whole-genome sequencing was performed for selected nuclei. The ResolveOME complementary DNA libraries from nuclei were found to have insufficient complexity for reliable expression analysis and so ResolveOME RNA sequencing data were used only for BCR reconstruction.

### DNA sequencing
DNA libraries underwent 150-bp paired-end DNA sequencing with the Illumina NovaSeq 6000 and X platforms. Sequencing reads were aligned to the GRCh37 (hs37d5 build) reference genome by the Burrows–Wheeler algorithm (BWA-MEM)[55]. Duplicate reads were marked using biobambam[56]. Library complexity and coverage statistics were calculated using Picard (https://broadinstitute.github.io/picard/).

### Methylation sequencing
To estimate the relative proportions of different cell types, we generated low-input enzymatic methylation libraries for 28 samples and then undertook targeted capture with a panel of informative CpG segments, using the NEBNext Enzymatic Methyl-seq Kit (E7120L, New England Biolabs), as previously described[5]. We used a custom Twist Bioscience hybridization panel targeting 977 CpG segments[21], corresponding to those listed in a published reference atlas (https://github.com/nloyfer/UXM_deconv/blob/main/supplemental/Atlas.U25.l4.hg19.tsv). This atlas contains the top 25 differentially methylated regions in 40 different cell types (Supplementary Table 3). We first sequenced sorted immune cell populations to assess the performance of this atlas in known cell types. This suggested that B and T cells in the atlas provided a poor fit to B memory and plasma cells, probably because the atlas was trained on circulating blood cells. Hence, we recalculated this atlas, substituting the original B and T cell types with six new cell types: B memory, B naive, CD4+ memory, CD4+ naive, CD8+ memory and CD8+ naive T cells. For each new cell type, we averaged across sorted immune cell populations from eight donors to recalculate their methylation levels. The modified atlas is available as Supplementary Table 3.

Methylation sequencing yielded a median coverage of 249× (range = 55–313×) across samples (Supplementary Table 4). Reads were mapped to the human genome (GRCh37, hs37d5 build) using bismark[57] through nf-core/methylseq[58] (https://github.com/nf-core/methylseq), with default options. The UXM_deconv pipeline[21] (https://github.com/nloyfer/UXM_deconv/) was used to obtain methylation levels for each CpG segment in the atlas. These methylation levels were then deconvoluted with the EpiDish R package (v.2.24.0), using the Robust Partial Correlations method[59] and our modified atlas (Supplementary Table 3). Finally, to estimate the fractions of B and T cells we summed the fractions of subtypes of B and T cells, respectively.

### NanoSeq mutation calling
Targeted and exome NanoSeq mutation calling was completed as previously described[5]. Briefly, this involved first excluding sequencing reads marked as optical duplicates, or having low base quality or poor mapping quality. Duplex consensus calling was performed in DNA molecules in which there were at least two copies of each strand (2 + 2 calling), with a consensus base quality greater than 60. A deduplicated BAM

was used as the matched normal. Mutations at VAF greater than 0.1 in the matched normal were initially excluded, and later reviewed and rescued if assessed to be a true somatic mutation. Duplex and unbiased VAFs were calculated as previously described[5]. For each gene, the mutant cell fraction was calculated by summing the non-synonymous duplex VAFs. As previously described[17], the sum of duplex VAFs represents the lower bound mutant cell fraction for that gene, in which each mutation is homozygous. For mutations in diploid regions of the genome, doubling this lower bound estimate provides the upper bound estimate, in which each mutation is heterozygous. Aggregated mutant cell fractions across multiple samples per donor or across multiple donors were weighted by the duplex coverage per gene to obtain weighted averages. NanoSeq mutation calls are available in Supplementary Table 5 and the obtained duplex coverage per gene in each sample is shown in Supplementary Table 6.

## Mutation calling from LCM samples

For each LCM, variants called by NanoSeq in biopsies from the same donor were genotyped using the bam2r function in the deepSNV[60] R package (v.1.43.6). This function piles up the nucleotide counts at each position of the specified gene in an aligned BAM file. VAFs were calculated by dividing the mutant bases at a specific site by the total site depth. Somatic substitutions and germline variants were called de novo by running CaVEMan (v.1.15.2) with a matched normal[61]. The matched normal was generated from a large microdissection of mixed non-lymphocytic histologies including thyroid follicles, fibrosis and adjacent connective tissue that underwent whole-genome sequencing. For increased sensitivity, CaVEMan settings included a mutant copy number of 10, wild-type copy number of 2 and normal contamination fraction of 0.1. Further filtering of CaVEMan calls was carried out as previously described[62]. Each CaVEMan run generated a list of germline single nucleotide polymorphisms (SNPs) specific for that donor. Indel calling was performed using Pindel[63] (v.3.10), with post hoc filtering as previously described[62]. Copy number analysis was undertaken with ascatNgs[63] (v.4.5), an implementation of the ASCAT algorithm[64]. The default parameters were used with a segmentation penalty of 100.

## Adaptive immune receptor sequence reconstruction in single nuclei

For each sequenced nucleus from donors H1, H2 and H8, adaptive immune receptor nucleotide sequences were reconstructed from deduplicated BAM files using the exome-seq preset of the MiXCR software[65] (v.4.7.0). MiXCR was run on both deep targeted and whole-genome single-nucleus sequencing data. For the latter, BAM files were pre-filtered using SAMtools[66] (v.1.13) to retain only those reads mapping to the human immunoglobulin and T cell receptor genes. Contigs assembled by MiXCR were aligned against the IMGT human reference *IG* and *TR* gene segments using IgBLAST[67] (default parameters) to determine V(D)J gene segment usage and CDR3 nucleotide and amino acid sequences. Immune receptor sequences that were deemed non-productive are not shown. For nuclei with evidence of a productive rearrangement at both the kappa and lambda light chain loci, coverage at the *IGKC* gene was assessed. Nuclei were assigned lambda light chain identity if they exhibited biallelic *IGKC* deletion.

## Selection and expression of recombinant monoclonal antibodies

For each sequenced nucleus from donor H1, heavy and light chain nucleotide sequences were also reconstructed using an orthogonal software tool, TRUST4 (v.1.1.8-r610)[68]. TRUST4 was run on deduplicated BAM files from deep targeted and whole-genome sequencing data, as well as deduplicated and non-deduplicated BAM files from mRNA transcriptome sequencing (ResolveOME protocol). Contigs were aligned against the IMGT human reference gene segments as described above[67]. Using the MiXCR and TRUST4 results, a full-length

consensus nucleotide sequence was obtained for the variable heavy and light chains of 32 B cells. Positive control nucleotide sequences were obtained from the literature, by searching GenBank for paired heavy and light chain sequences known to bind human TPO or TG: AJ238327.1 and AJ238330.1 (clone ICA5, anti-human TPO); AJ399834.1 and AJ399876.1 (clone T8, anti-human TPO); AY365327.1 and AY365334.1 (clone #6, anti-human TG); AY365330.1 and AY365338.1 (clone #26, anti-human TG).

Subsequent steps for recombinant expression were conducted by Biointron Biological. Variable heavy and light chain sequences for all 32 nuclei and the 4 positive controls were codon-optimized, synthesized and subcloned into expression vectors (pcDNA3.4 backbone) encoding a signal peptide sequence and the human IgG1 heavy chain constant region or kappa or lambda light chain constant regions. For eight nuclei with higher confidence BCR sequence reconstructions, the variable heavy chain sequence was additionally cloned into an expression vector encoding the rabbit IgG1 heavy chain constant region. Plasmids for paired heavy and light chain sequences were transiently transfected into CHO-K1 cells, and monoclonal antibodies were affinity purified before undergoing quality control (SDS–PAGE, size exclusion high-perfomance liquid chromatography and endotoxin quantification).

## ELISA

ELISAs were conducted by Biointron Biological. First, 96-well EIA/RIA microplates (3590, Corning) were coated for 2 h at 25 °C with 2 µg ml$^{-1}$ of one antigen: recombinant human TPO (NM_000547; TP310659, OriGene) or recombinant human TG (NM_003235; TP316216, OriGene). The plates were washed once with 0.05% PBST (PBS + 0.05% Tween20, used for all subsequent washes) and blocked with 5% BSA (Sangon Biotech, A500023-0100) for 1 h at 25 °C. This was followed by a further three washes. For each 96-well ELISA plate, 9–10 recombinant monoclonal antibodies and 2–3 isotype control antibodies (B117901, B646501 and B730001, Biointron Biological) were included. The antibodies were diluted in 1× PBS from 100 nM in a fourfold serial dilution across each plate, followed by incubation for 1 h at 25 °C. The plates were washed three times, and the appropriate horseradish peroxidase-conjugated secondary antibody was added to each well: goat anti-human IgG (Fc-specific) (1:10,000 dilution, 30 min of incubation at 25 °C; A0170, Sigma-Aldrich, Merck) or goat anti-rabbit IgG antibody (1:10,000 dilution, 1 h of incubation at 25 °C, ARG65351, Arigo Biolaboratories). Following a further six washes, 3,3′,5,5′-tetramethylbenzidine reagent (1001, Makewonderbio) was added to each well for an incubation in the dark at 25 °C. The reactions were stopped with 2 M sulphuric acid (Beyotime Biotechnology, P0215). The optical density of the solution at 450 nm was read on a Multiskan FC microplate photometer (Thermo Fisher Scientific). Dose–response curves were fitted and EC$_{50}$ values calculated using the drc R package[69].

## LOH analysis in single nuclei

To assess LOH in the single nuclei of H1, heterozygous germline SNPs were first identified from a high-coverage whole-genome (68×) microdissection using CaVEMan[61]. These calls were filtered to those at common SNP sites, with a depth >50 and a mutant VAF between 0.3 and 0.7. A total of 2,087,702 heterozygous germline SNPs were identified. These SNPs were genotyped in each nucleus using the bam2r function of deepSNV[60]. Allele fraction plots were generated to assess LOH, with phasing of SNPs into haplotypes inferred from the nucleus with the longest stretch of CN-LOH affecting chromosome 1p (nucleus 25). Breakpoint positions for 1p CN-LOH were determined by calculating the median separation in B-allele fractions across a sliding window of 201 heterozygous SNPs. BAMs were subsampled to 2 million reads and genome-wide copy number profiles were estimated based on binned read coverage and uniformity using Ginkgo[70]. Further manual review of sequencing data was undertaken with Integrative Genomics Viewer (IGV)[71].

## Mutation calling in single nuclei

For each sequenced nucleus, somatic variants called by NanoSeq in bulk biopsies and LCMs from the same donor were genotyped using the bam2r function in the deepSNV[60] R package and the following conditions: base quality ≥30, minimum mapping quality ≥30, VAF >0.2, mutant depth ≥2. Genotyped NanoSeq mutations are available in Supplementary Table 7 and extra annotations per nucleus are included in Supplementary Table 8. De novo somatic variant calling in single nuclei was completed with the bj-somatic-variantcalling pipeline and the bioskryb129 DNAscope prediction model (BioSkryb Genomics). Variants at SNP sites in the CaVEMan data were removed to reduce possible germline contamination. Phylogenetic tree reconstruction was performed using Sequoia[72]. Further manual review of sequencing data was undertaken with IGV[71].

## Mutational signature analysis in single nuclei

Mutational signature extraction was undertaken with hdp (v.0.1.5, https://github.com/nicolaroberts/hdp) and SigProfiler[73] (https://cancer.sanger.ac.uk/signatures/tools/). De novo signatures were deconvoluted to reference signatures in the COSMIC v.3.5 database[74] (https://cancer.sanger.ac.uk) and other published signatures[46,75]. Signature fitting was completed with sigfit (v.2.2, https://github.com/kgori/sigfit). Further details can be found in Supplementary Note 3.

## Selection analyses

To identify genes under selection, we used dNdScv (v.0.1.0, https://github.com/im3sanger/dndscv), a maximum-likelihood implementation of d$N$/d$S$ that corrects for sequence composition, trinucleotide mutation rates and variable rates across genes through the use of epigenetic covariates[22]. As previously described[5], dNdScv can be used for driver discovery from exome and targeted NanoSeq data by incorporating the duplex coverage per gene into the model. However, driver discovery in B cells requires a more bespoke approach, as a subset of genes are affected by off-target activation-induced cytidine deaminase-mediated SHM[76], which violates the assumption of a shared substitution model across genes in the standard dNdScv model.

To account for off-target SHM, we utilized deep exome NanoSeq data generated from flow-sorted memory B cells (CD3⁻CD19⁺CD20⁺CD27⁺) obtained from PBMCs of nine donors with no history of autoimmunity to identify exons affected by off-target SHM (Lawson et al., in preparation). Exons were classified into two groups (those affected or unaffected by off-target SHM) based on the density of mutations at synonymous sites and flanking intronic sequences in each exon. We then applied dNdSloc (which uses a separate mutation rate estimate, $t$, for each gene) to the hypermutated exons and standard dNdScv (which utilizes the joint likelihood of a gamma–Poisson compound distribution, modelled as a negative binomial distribution, in which the observed numbers of synonymous mutations per gene are modelled as a Poisson process whose mean is drawn from a gamma distribution reflecting the variation of mutation rate across genes) to unaffected exons[22]. For genes containing both exon types, one-sided $P$ values from dNdSloc and dNdScv were combined using Fisher's method[77] to have a single combined $P$ value per gene, and multiple testing correction was carried out across all genes using the Benjamini–Hochberg method[78].

Selection on individual codons and hotspots was performed by using the codondnds and sitednds functions of dNdScv[5]. The assessment of selection on specific structural aspects and functional domains of individual genes was performed using the withingenednds function of dNdScv[5]. Pairwise comparison of selection on genes between the thyroid autoimmunity cohort and B and T cell populations obtained from PBMCs of aged non-autoimmune donors (Lawson et al., in preparation), as well as the control spleen, lymph node, microdissected lymphoid aggregates, tonsillitis and thyroid goitre samples, was undertaken

using pairwisednds as previously described[62], with further duplex depth correction[5].

## Spatial transcriptomics

Spatial transcriptomics was performed using the Xenium platform (10x Genomics) using a previously described protocol for sample preparation[79]. PFPE tissue was sectioned to 5 μm thickness and mounted onto Xenium slides (10x Genomics) with an imageable area equal to 12 × 24 mm². Pepsin digestion was performed for 1.5 min or 2 min depending on whether a 380-plex or 5,000-plex RNA panel was subsequently used. Using a 380-plex RNA panel specific to immunoregulatory genes (Xenium Immune-Oncology Panel, 10x Genomics) or a 5,000-plex pan-tissue RNA panel (Xenium Prime 5K Human Pan Tissue & Pathways Panel, 10x Genomics), the workflow proceeded as follows: probe hybridization to target mRNAs; probe ligation to form circular templates; rolling-circle amplification to amplify signals. Signals were detected via iterative rounds of fluorescent probe hybridization over approximately 2 d, and imaged using the Xenium Analyzer (10x Genomics). An optical barcode system was used to map each transcript signal to its gene target.

Cell segmentation and transcript assignment were performed using multimodal approaches embedded in the Xenium instrument (10x Genomics). Xenium output data were imported into an R v.4.4.1 environment and processed with the Seurat R package[80,81] (v.5.3.0). Briefly, cells were filtered, the transcript counts were normalized, cells were clustered and dimensionality reduction was performed via uniform manifold approximation and projection (10x Genomics). Cell clusters were identified and annotated using a dual approach: scoring cells for canonical marker panels via the AddModuleScore function and subsequent manual refinement based on cluster-specific expression profiles. Spatial distributions were visualized using Seurat and Xenium Explorer 4.1.1 software (10x Genomics).

Data from the 380-plex Xenium Immune-Oncology Panel are shown in Fig. 4d, as these Xenium sections are proximal to the microdissected regions shown in Fig. 4b,c. Data from the Xenium Prime 5K Human Pan Tissue & Pathways Panel are shown in Extended Data Fig. 1a,b from sections that are further away from those used for microdissection.

## Reporting summary

Further information on research design is available in the Nature Portfolio Reporting Summary linked to this article.

## Data availability

Donor metadata are available in Supplementary Table 1. A list of genes captured in the targeted panel is in Supplementary Table 2. A modified atlas of differentially methylated regions in 44 different cell types is in Supplementary Table 3. Cell type estimates by targeted methylation sequencing are in Supplementary Table 4. NanoSeq mutation calls are in Supplementary Table 5. Per sample duplex coverage is in Supplementary Table 6. Genotyped mutations in single-nucleus sequencing data are in Supplementary Table 7. Additional annotations per nucleus are in Supplementary Table 8. Sequencing data have been deposited in the European Genome-Phenome Archive (EGA) under accession numbers: EGAD00001016058 (Exome NanoSeq), EGAD00001016059 (Targeted NanoSeq), EGAD00001016060 (Targeted EMSeq), EGAD00001016061 (PTA WGS), EGAD00001016062 (PTA DNAHyb), EGAD00001016063 (PTA RNA), EGAD00001016064 (LCMB WGS) and EGAD00001016065 (LCMB WES). Xenium data can be found on the EMBL-EBI BioImage Archive (accession number S-BIAD3103).

## Code availability

Analyses were performed with R[82] (v.4.5) and relevant code can be found as Supplementary Code. R packages used include: tidyverse[83] (v.2),

GenomicRanges[84] (v.1.62.1), dndscv[22] (v.0.0.1.0), scales[85] (v.1.4.0), patchwork[86] (v.1.3.2), viridis[87] (v.0.6.5), RColorBrewer[88] (v.1.1-3), lattice[89] (v.0.22-6), latticeExtra[90] (v.0.6-31), vcfR[91] (v.1.15.0), MASS[92] (v.7.3-65), jsonlite[93] (v.2.0.0), ggforce[94] (v.0.5.0), stringi[95] (v.1.8.7), gtools[96] (v.3.9.5), drc[69] (v.3.0-1), pander[97] (v.0.6.6), ape[98] (v.5.8-1), ggtree[99] (v.4.0.4), ggh4x[100] (v.0.3.1) and ggpubr[101] (v.0.6.2). Full details for running the analyses documented in the Supplementary Code and additional code for analysing the single-nucleus sequencing data can be found at https://doi.org/10.5281/zenodo.19366068.

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

**Acknowledgements** We thank the patients who consented for their tissue samples to be used for clinical research, the CUH Histopathology department, E. Cromwell and CUH Human Research Tissue Bank. We are grateful to the deceased transplant donors and their families for the gift of tissue donation facilitated by the Cambridge Biorepository for Translational Medicine. We thank the Cytometry Core Facility at Wellcome Sanger Institute for their support. We thank CASM Support (L. O'Neill, C. Hardy and R. Thatcher), CASM Research Managers (I. Tang, H. Robertson and M. Venables) and CASM IT (A. Butler, A. Menzies, A. Burdett and V. Offord) at the Wellcome Sanger Institute for facilitating this work. We thank CASM Support Lab (K. Roberts, T. Baxter, K. Smith, E. Ferla, M. Khan and L. Allen) for their assistance with sample shipments. We thank H. Savin, E. Fineberg and E. Tuck for their histopathology and laboratory assistance. We also thank the Sequencing Operations Team and the Sanger Translation Office (G. Dillon) for their support. We thank D. Goulding for microscopy advice. We thank T. Coorens for support in implementing Sequoia. We thank C. Goodnow and L. Donlin for invaluable comments on the manuscript. This research was funded in whole, or in part, by the Wellcome Trust grant no. 220540/Z/20/A. For the purpose of Open Access, the authors have applied a CC BY public copyright licence to any Author Accepted Manuscript version arising from this submission. P.A.N. was funded by a Cambridge NIHR BRC Capacity Building award (RHZB/307; grant no. G122835). CUH Human Research Tissue Bank is supported by the NIHR Cambridge Biomedical Research Centre (grant no. NIHR203312). N.S. is supported by the Wellcome Trust (grant no. 219296/Z/19/Z) and the NIHR Cambridge Biomedical Research Centre. I.M. was funded by Cancer Research UK (grant no. C57387/A21777), the Milky Way Research Foundation, the Dr Josef Steiner Cancer Research Foundation and the Wellcome Trust.

**Author contributions** P.A.N., A.R.J.L. and I.M. conceptualized the project with support from K.C., N.S., J.A.T., M.R.S. and P.J.C. P.A.N., A.R.J.L., A.T., J. Imbert, Y.I., L.A.W., F.A. and I.M. led data analysis with support from L.M.R.H. and D.L. P.A.N., A.R.J.L., S.V.L. and W.C. led the experimental work with support from Y.H., P.A.S., K.R., Y.I., M.M., N.M.-S., S.W., L.M., J. Illing, F.P., B.L.N., G.L.J. and A.J.C.R. A.L.P. and J.A.T. provided histopathology expertise. K.S.-P., K.T.A.M., K.C., J.A.T. and N.S. led patient recruitment, sample collection and clinical annotation. R.R., O.A.B., M.R.S., P.J.C., J.A.T., N.S. and I.M. provided supervision. P.A.N., A.R.J.L. and I.M. wrote the manuscript, and all authors contributed to reviewing and editing it.

**Competing interests** I.M., M.R.S. and P.J.C. are co-founders, shareholders and consultants for Quotient Therapeutics Ltd. P.A.N., A.R.J.L., L.M.R.H., J.A.T., N.S. and I.M. declare a provisional USA patent application filed with Wellcome Sanger Institute (No. 63/806,614, 15 May 2025). The other authors declare no competing interests.

**Additional information**
**Correspondence and requests for materials** should be addressed to Andrew R. J. Lawson or Iñigo Martincorena.

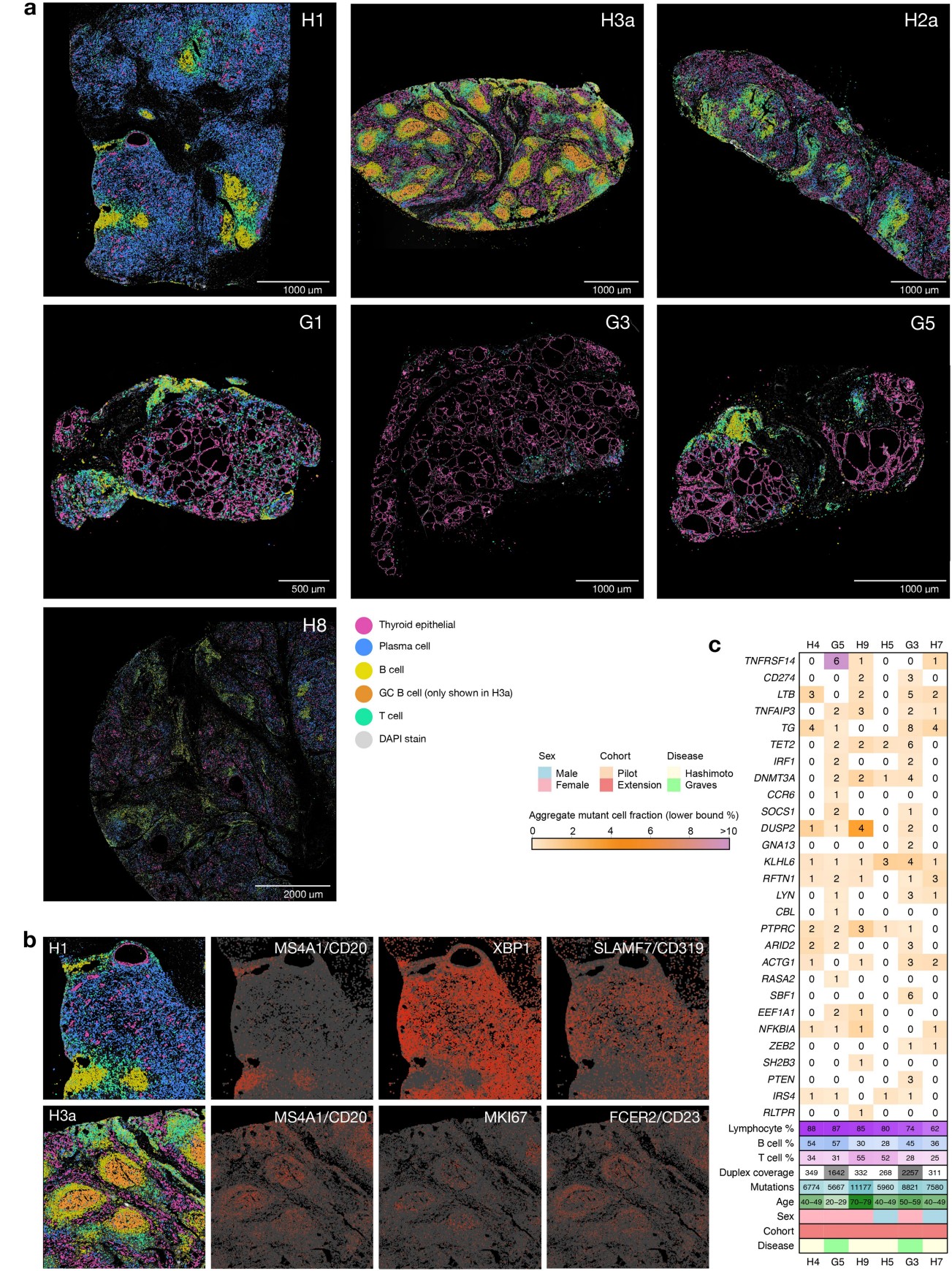

**Extended Data Fig. 1** | See next page for caption.

**Extended Data Fig. 1 | Cell type composition inferred by spatial transcriptomics and mutation heatmap for microdissected regions. a**, Cell type composition inferred from spatial transcriptomics using the Xenium Prime 5K Human Pan Tissue & Pathways Panel for 7 donors. **b**, Selected regions from donors H1 and H3 shown at higher magnification alongside expression data for canonical B cell, germinal centre B cell and plasma cell markers.

**c**, Heatmap showing the number of non-synonymous mutations as text (unique counts per donor) and the lower bound mutant cell fraction (background colour capped at 10%) per donor across laser capture microdissected samples for 32 genes (31 significant by gene-level d$N$/d$S$ and *RLTPR*, which is significant by site-level d$N$/d$S$). Columns are sorted by decreasing proportion of cells that are lymphocytes (estimated from targeted methylation data).

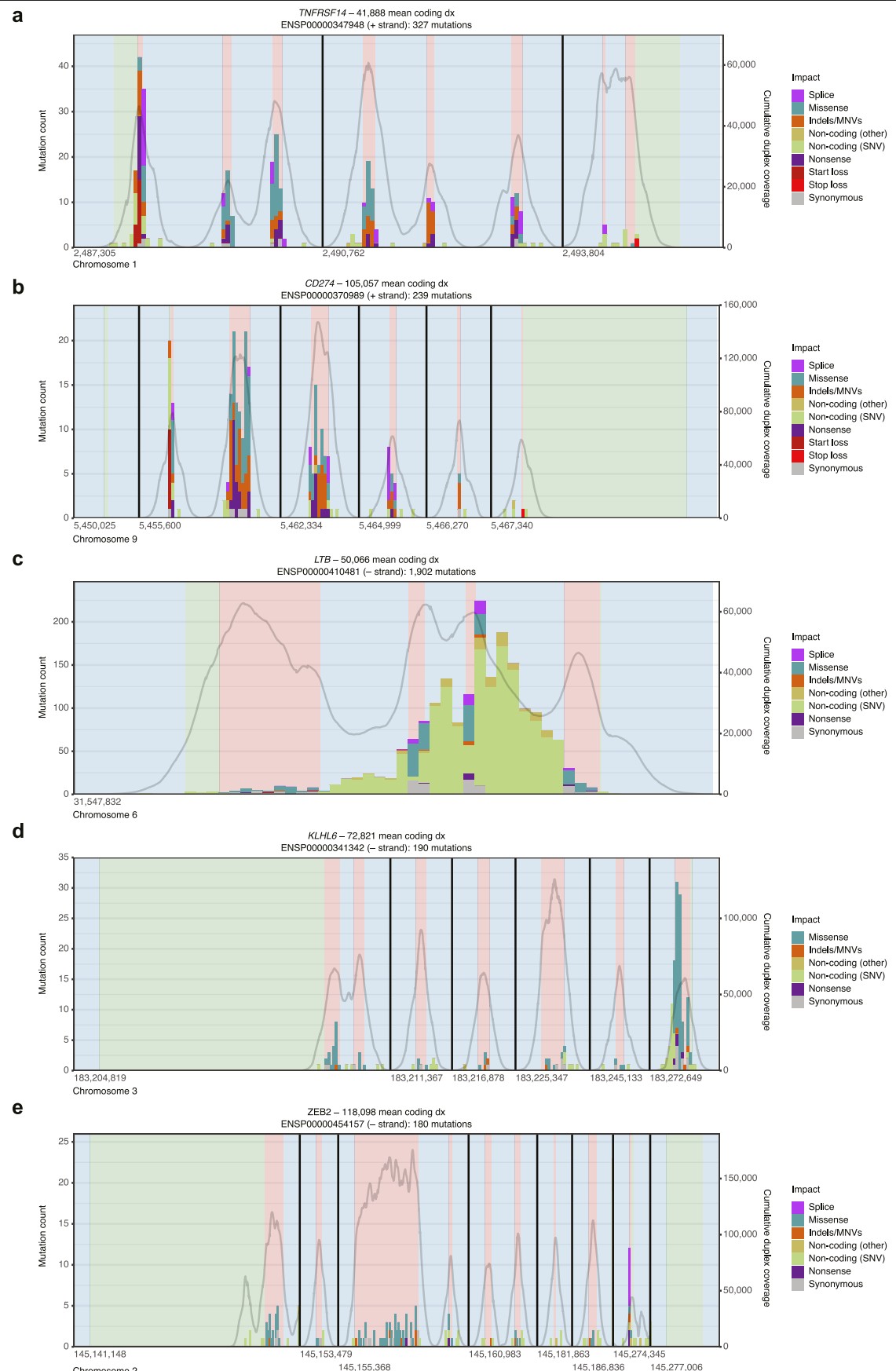

**Extended Data Fig. 2 |** See next page for caption.

**Extended Data Fig. 2 | Mutation distribution across the gene body for selected AITD drivers. a**-**e**, Mutation barplots for five selected driver genes: *TNFRSF14*, *CD274*, *LTB*, *KLHL6* and *ZEB2*. The *x*-axis represents genomic coordinates along the gene body, with coding exons (red), UTRs (green) and intronic / intergenic regions (blue) indicated by the shading within each histogram. The grey line denotes cumulative duplex coverage across AITD samples. Coding mutation counts are coloured according to mutation consequence. Genes are shown in chromosomal orientation with the coding strand indicated above each plot.

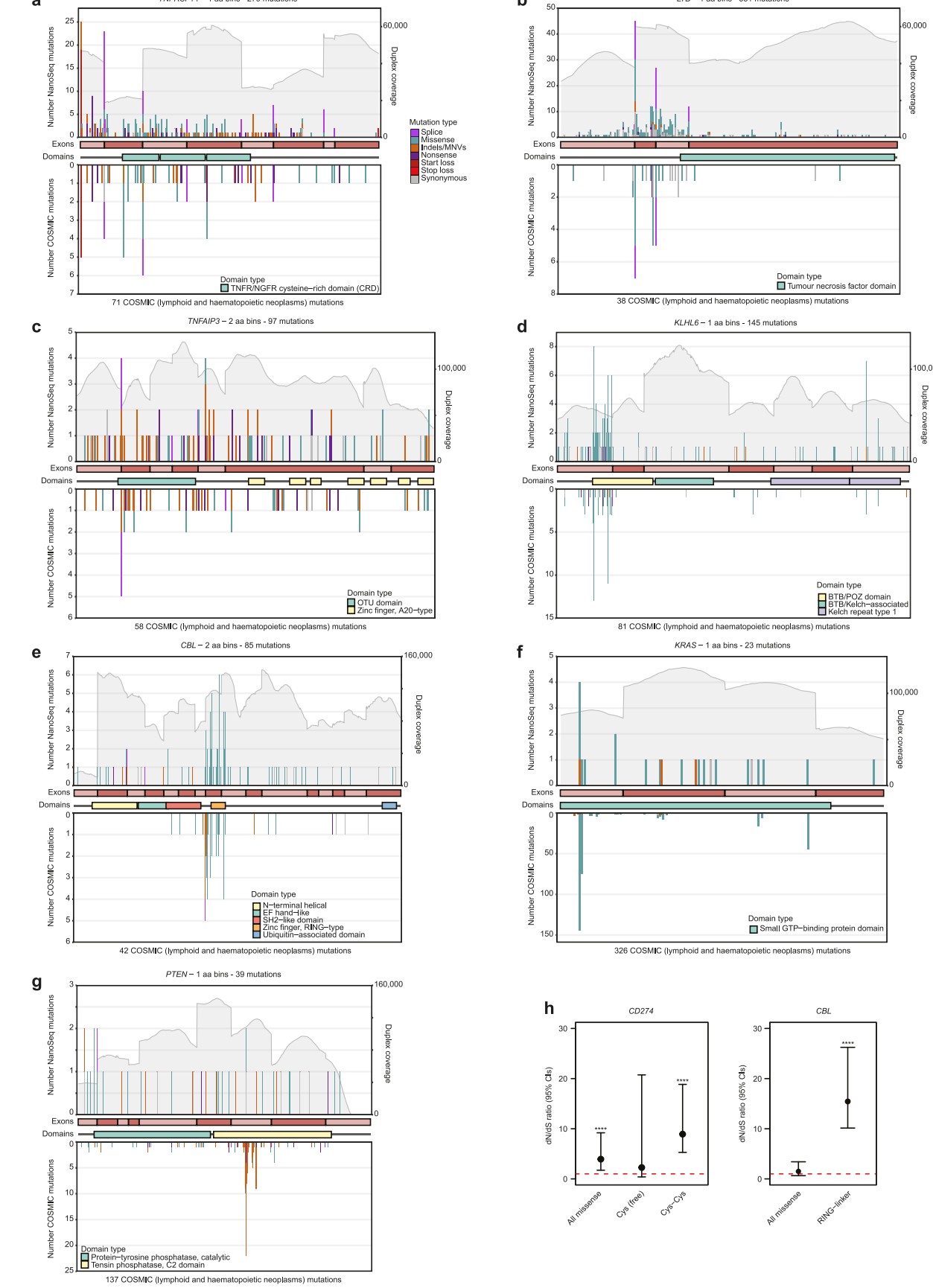

**Extended Data Fig. 3 |** See next page for caption.

**Extended Data Fig. 3 | Coding mutation distributions and selection analyses within genes. a-g,** Coding mutation barplots for seven selected driver genes: *TNFRSF14*, *LTB*, *TNFAIP3*, *KLHL6*, *CBL*, *KRAS* and *PTEN*. The *x*-axis represents coordinates along the coding sequence. Exons and protein domains are indicated along the *x*-axis. The *y*-axis represents the number of mutations, either in the AITD samples used in this study (top) or in leukaemias and lymphomas that have undergone whole exome or whole genome sequencing in the COSMIC[102] database (bottom). Mutations are coloured according to the mutation consequence category. Grey shading indicates cumulative duplex coverage across AITD samples. **h,** d*N*/d*S* ratios and 95% confidence intervals estimated by withingenednds for classes of missense mutations in *CD274* (left) and *CBL* (right). Cys-Cys and Cys (free) denote cysteine residues involved / not involved in disulphide bonds, respectively. The horizontal red dashed line indicates neutral d*N*/d*S* = 1. ****: $Q < 10^{-4}$; ***: $Q < 10^{-3}$; **: $Q < 0.01$; *: $Q < 0.1$; •: $P < 0.01$; *Q* values were calculated with the Benjamini-Hochberg method.

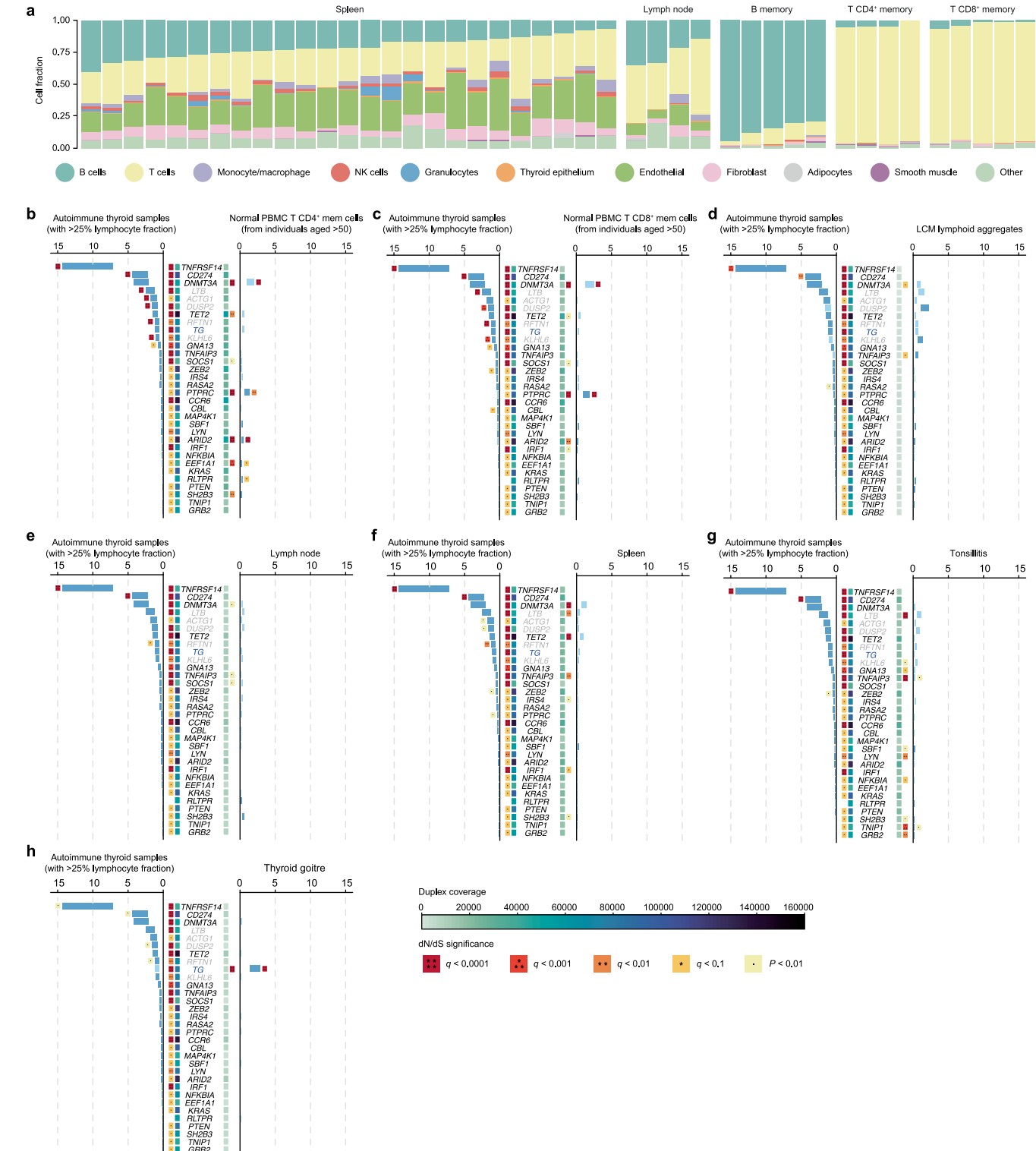

**Extended Data Fig. 4 | Comparison of driver mutant cell fractions and selection between AITD and control data sets. a**, Cell type composition of bulk spleen and lymph node biopsies and sorted memory B, CD4⁺ and CD8⁺ memory T cells obtained from PBMC (peripheral blood mononuclear cell) samples from healthy donors aged >50 estimated by targeted methylation sequencing. Within each group, samples are sorted by decreasing B cell fraction. **b-h**, Comparison of mutant cell fractions and strength of selection between lymphocyte-rich thyroid autoimmunity samples and seven control data sets: sorted T memory cells (**b**, CD4⁺ and **c**, CD8⁺) derived from PBMC samples from healthy donors aged >50 ($n$ = 20 donors); **d**, microdissected lymphoid aggregates from colon, ileum and tonsil ($n$ = 5 donors); **e**, lymph nodes ($n$ = 4 donors); **f**, spleen ($n$ = 30 donors); **g**, chronically inflamed tonsil

samples obtained from tonsillectomies ($n$ = 5 donors); and **h**, thyroid samples with goitres but no evidence of AITD ($n$ = 5 donors). Gene names highlighted in grey denote genes that reach significance solely due to the signal from exons affected by somatic hypermutation, whereas those in blue indicate lineage defining genes with increased indel mutagenesis. Boxes either side of gene names show duplex coverage and d$N$/d$S$ significance within each sample cohort. $P$ values were calculated using dndsshm and pairwisednds and corrected using Benjamini-Hochberg's method. Blue bars denote non-synonymous mutant cell percentages (upper and lower bounds), with genes showing significantly increased selection in one sample set compared to the other sample set (by pairwisednds) denoted by significance boxes.

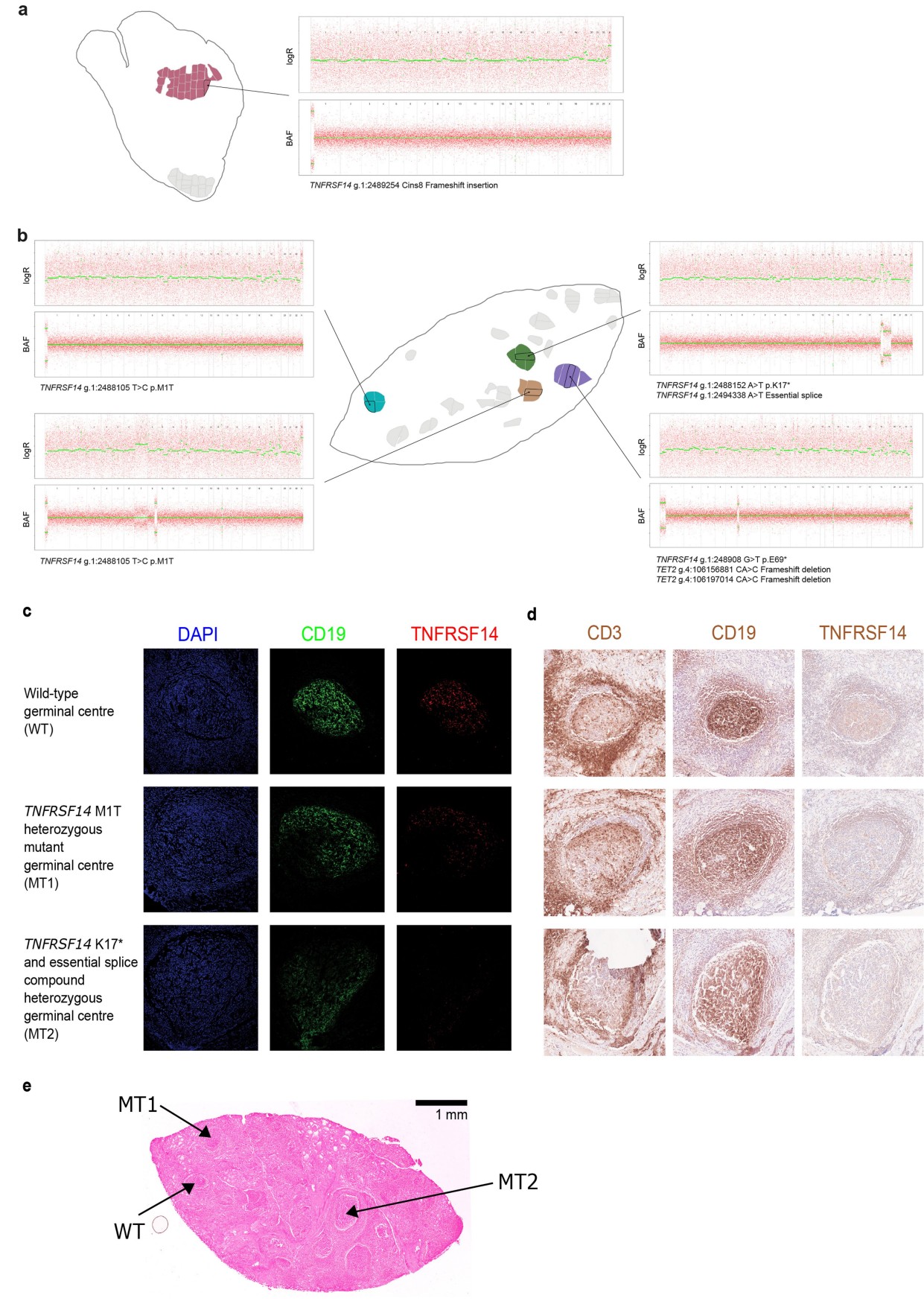

**Extended Data Fig. 5 |** See next page for caption.

**Extended Data Fig. 5 | Copy number variants identified in microdissected mutant clones and immunostaining of *TNFRSF14* wildtype and mutant germinal centres. a**,**b**, For two donors, (**a**) G5 and (**b**) H3, spatial maps denote lymphoid aggregates with copy number variants. Log[R] and B-allele fraction plots are shown for a representative microdissection (indicated by arrow and black outline) from the highlighted lymphoid aggregates.

**c**, Immunofluorescence for TNFRSF14 wildtype and mutant germinal centres from donor H3. Microscope and light source settings were kept constant across the images captured from the three germinal centres. **d**, Immunohistochemistry for the same three germinal centres shown in **c. e**, H&E overview showing the locations of the germinal centres shown in **c** and **d**.

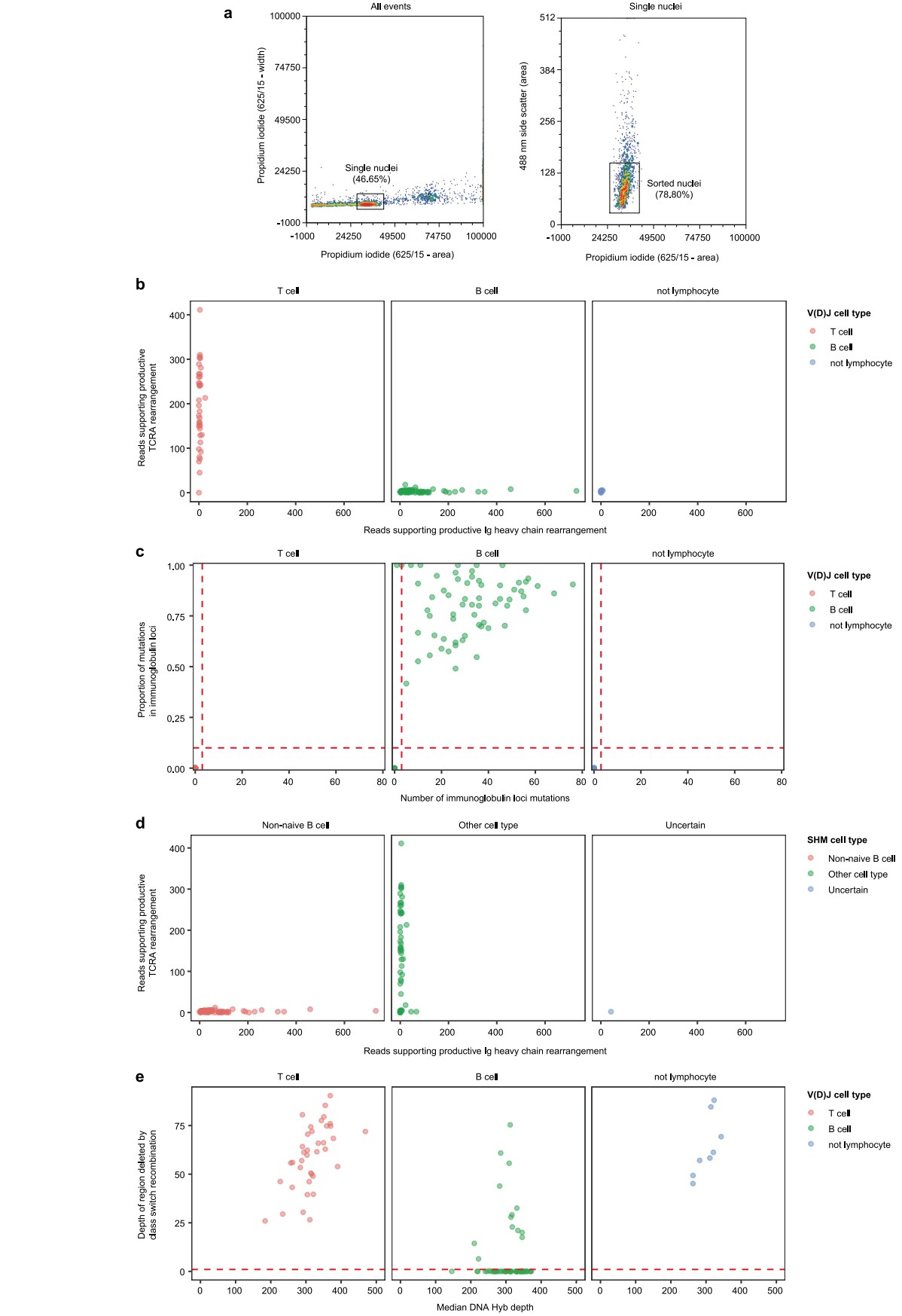

**Extended Data Fig. 6 |** See next page for caption.

**Extended Data Fig. 6 | Cell type identification from single nucleus sequencing data. a**, Fluorescence-activated cell sorting strategy for isolating single nuclei. **b**, Scatter plot showing read support for productive V(D)J recombination at the immunoglobulin heavy chain and TCRA loci, faceted by V(D)J inferred cell type. **c**, Scatter plot showing the number of mutations genotyped within the immunoglobulin heavy and light chain loci (14:106,053, 226-107,288,019 and 22:22,380,474-23,265,203 respectively) and the proportion of all genotyped mutations detected in the immunoglobulin loci, faceted by V(D)J inferred cell type. Red dashed lines denote cut-offs of 3 genotyped mutations in immunoglobulin loci and 10% of all genotyped mutations occurring in immunoglobulin loci. **d**, Scatter plot showing read support for productive V(D)J recombination at the immunoglobulin heavy chain and TCRA loci, faceted by somatic hypermutation inferred cell type. **e**, Scatter plot of the median depth obtained by whole genome sequencing (WGS) and the mean sequencing depth obtained in the region deleted by class switch recombination (14:106,303, 099-106,322,323). The red dashed line denotes median depth of 1 within the class switch recombination region.

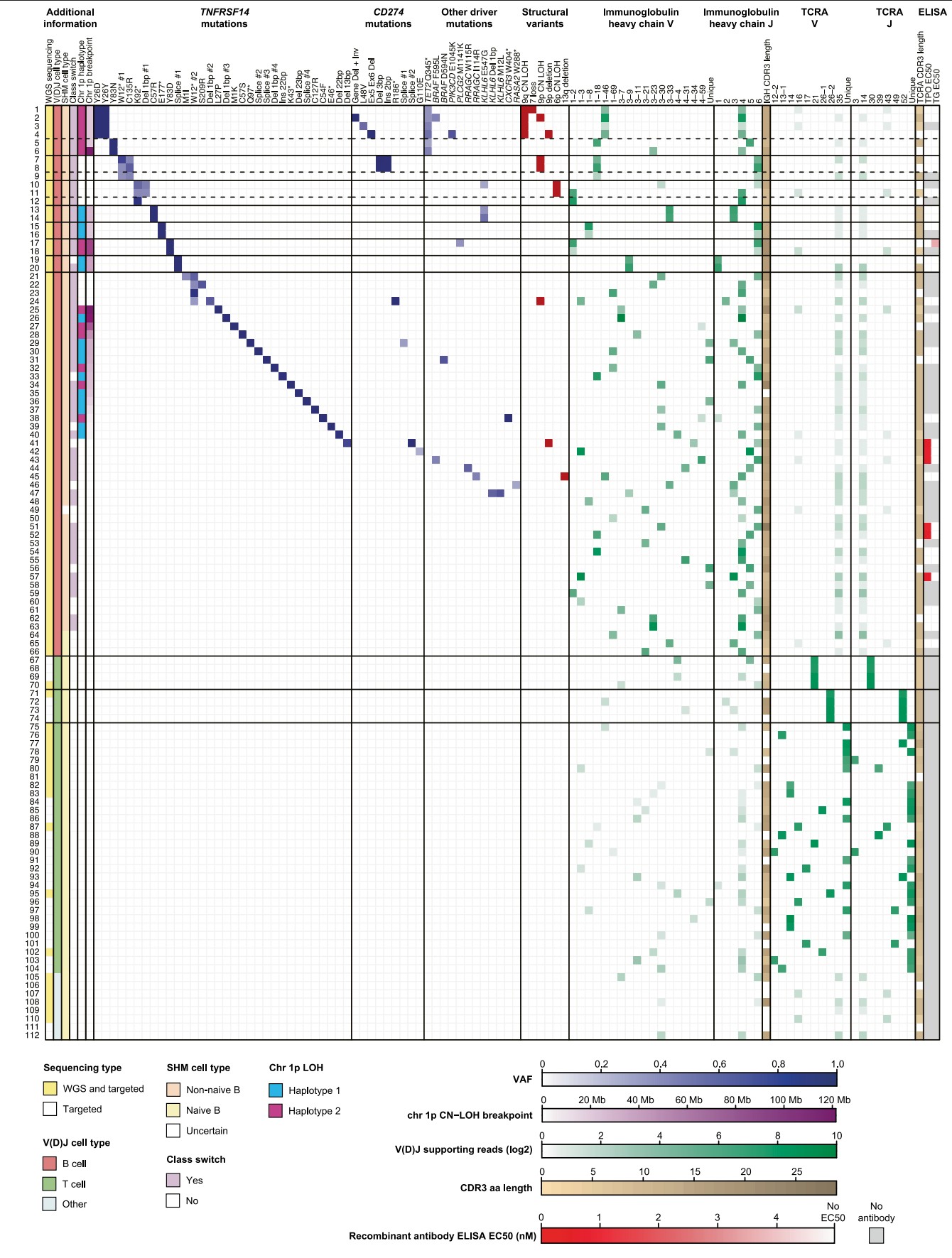

**Extended Data Fig. 7** | See next page for caption.

**Extended Data Fig. 7 | Full single nucleus sequencing heatmap for donor H1.**
Heatmap of driver mutations and V(D)J recombination events from all 112 cells from donor H1 characterised by single nucleus sequencing. From left to right, panels denote: sequencing strategy (WGS only or WGS and targeted); cell type inferred by V(D)J recombination and somatic hypermutation; class switch recombination status; the haplotype and breakpoint position for 1p copy-neutral loss of heterozygosity (CN-LOH) events; variant allele fractions for mutations in *TNFRSF14*, *CD274* and selected other genes; V and J segments used in productive immunoglobulin heavy chain rearrangements and the length of the immunoglobulin heavy chain CDR3 sequence; V and J segments used in productive TCRA rearrangements and the length of the TCRA CDR3 sequence; and EC50 values for TG and TPO ELISA using recombinant antibodies. Horizontal lines demarcate clades (or sub-clades for dashed lines) of related cells.

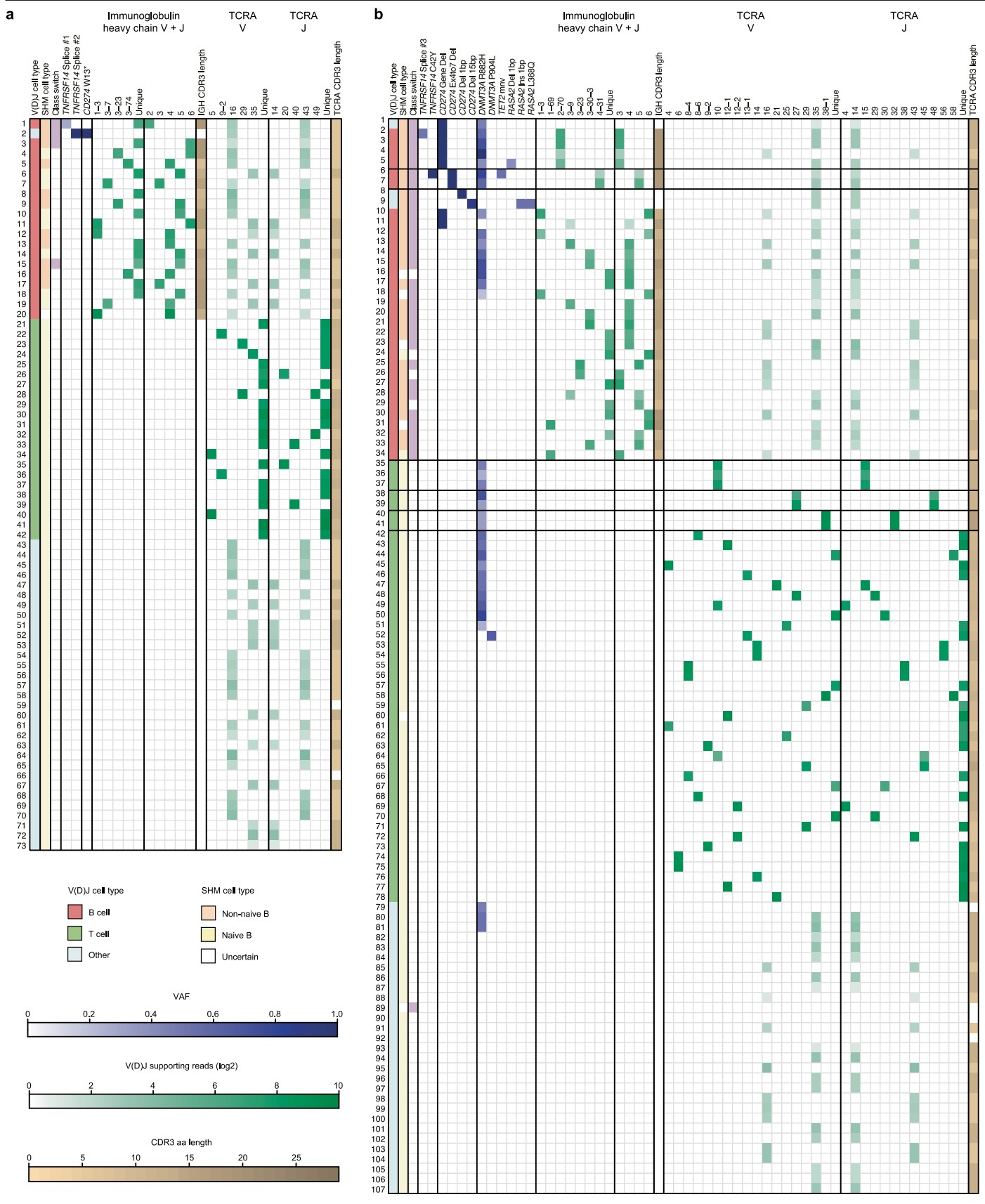

**Extended Data Fig. 8** | See next page for caption.

**Extended Data Fig. 8 | Single nucleus sequencing heatmaps for donors H2 and H8.** Heatmaps of driver mutations and V(D)J recombination events characterised by targeted single nucleus sequencing for: **a**, 73 cells from donor H2; and **b**, 107 cells from donor H8. From left to right, panels denote: cell type inferred by V(D)J recombination and somatic hypermutation; class switch recombination status; variant allele fractions for mutations in *TNFRSF14*, *CD274* and selected other genes; V and J segments used in productive immunoglobulin heavy chain rearrangements and the length of the immunoglobulin heavy chain CDR3 sequence; V and J segments used in productive TCRA rearrangements and the length of the TCRA CDR3 sequence. Horizontal lines demarcate clades of related cells.

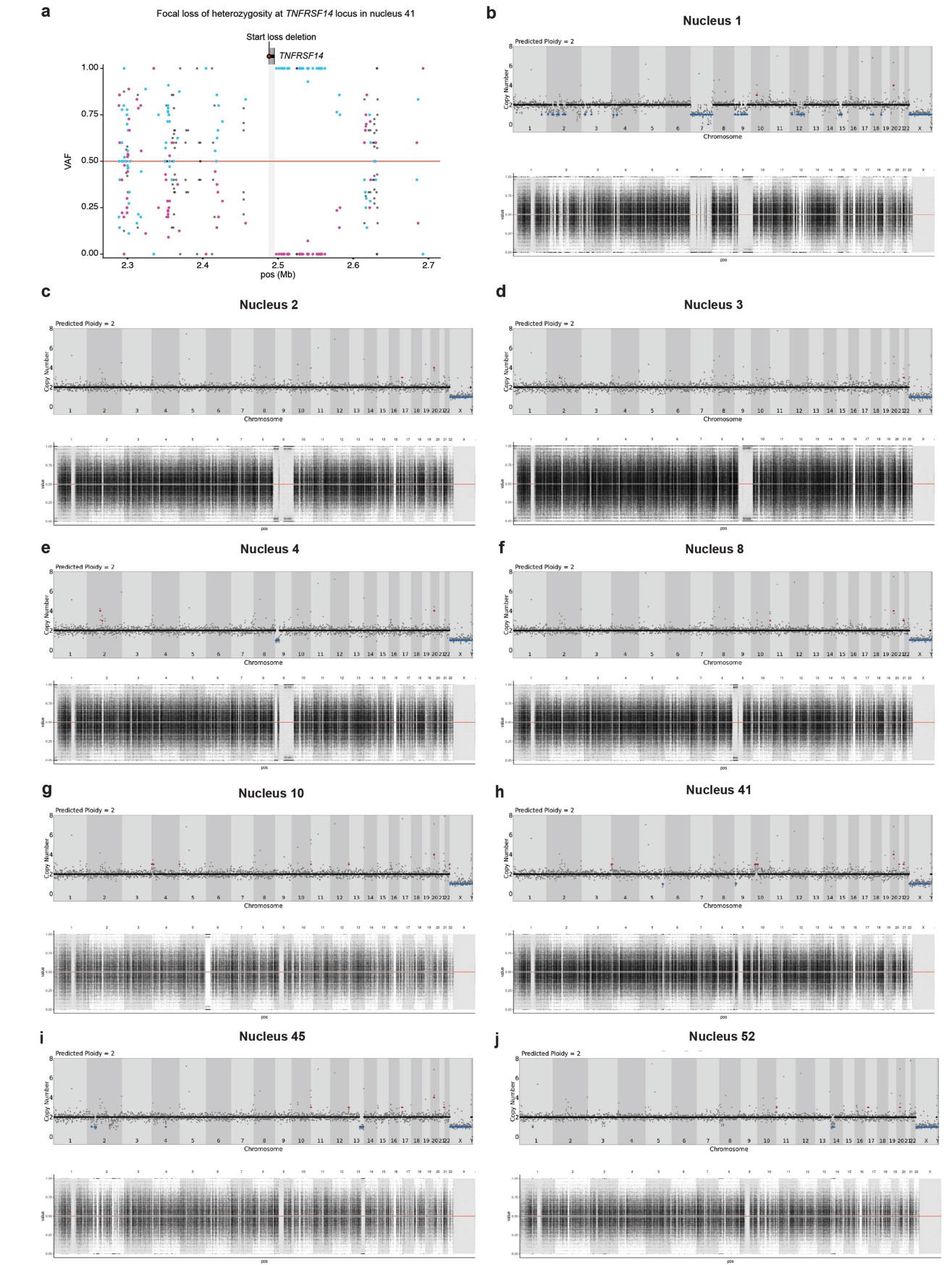

**Extended Data Fig. 9 | Additional structural variants identified in single nucleus sequencing data from donor H1. a**, Focal loss of heterozygosity at *TNFRSF14* locus in nucleus 41. **b-j**, Log[R] (top) and B-allele fraction (bottom) plots showing the support for structural variants other than 1p CN-LOH.

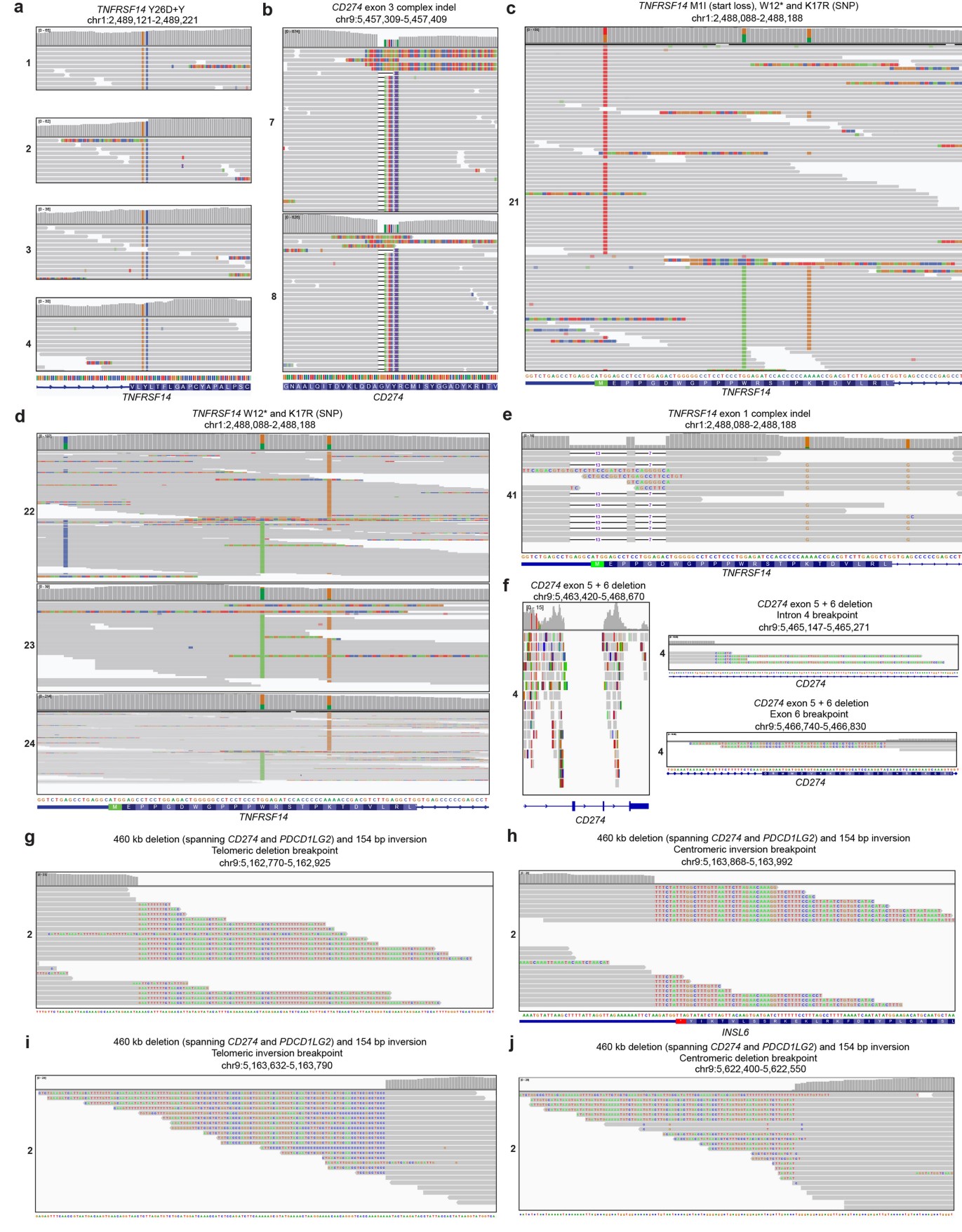

**Extended Data Fig. 10** | See next page for caption.

**Extended Data Fig. 10 | Characterisation of complex mutations identified in single nucleus sequencing data. a-j**, Integrative Genomics Viewer (IGV)[71] plots for selected complex mutations. **a**, *TNFRSF14* Y26D and Y26Y mutation in nuclei 1-4. **b**, *CD274* complex indel in nuclei 7 and 8. **c**, *TNFRSF14* M1I (start loss) is out of phase with *TNFRSF14* W12* mutation and K17R SNP in nucleus 21. **d**, Convergent independent nonsense mutations affecting *TNFRSF14* W12. Nuclei 22 and 24 have W12* mutation out of phase with K17R SNP, whereas nuclei 23 (and 21, shown in **c**) have W12* in phase with K17R SNP. **e**, Complex indel causing start loss of *TNFRSF14* is in phase with K17R SNP and intron 1 splice donor mutation in nucleus 41. **f**, *CD274* deletion spanning exons 5 and 6 in nucleus 4. Entire region and reads spanning deletion breakpoints are shown. **g-j**, Breakpoint spanning reads of a 460 kb large deletion (which includes a 154 bp inversion of the 3' UTR of *INSL6*) on chromosome 9 that results in complete loss of *CD274* and *PDCD1LG2*, which encodes PD-L2.

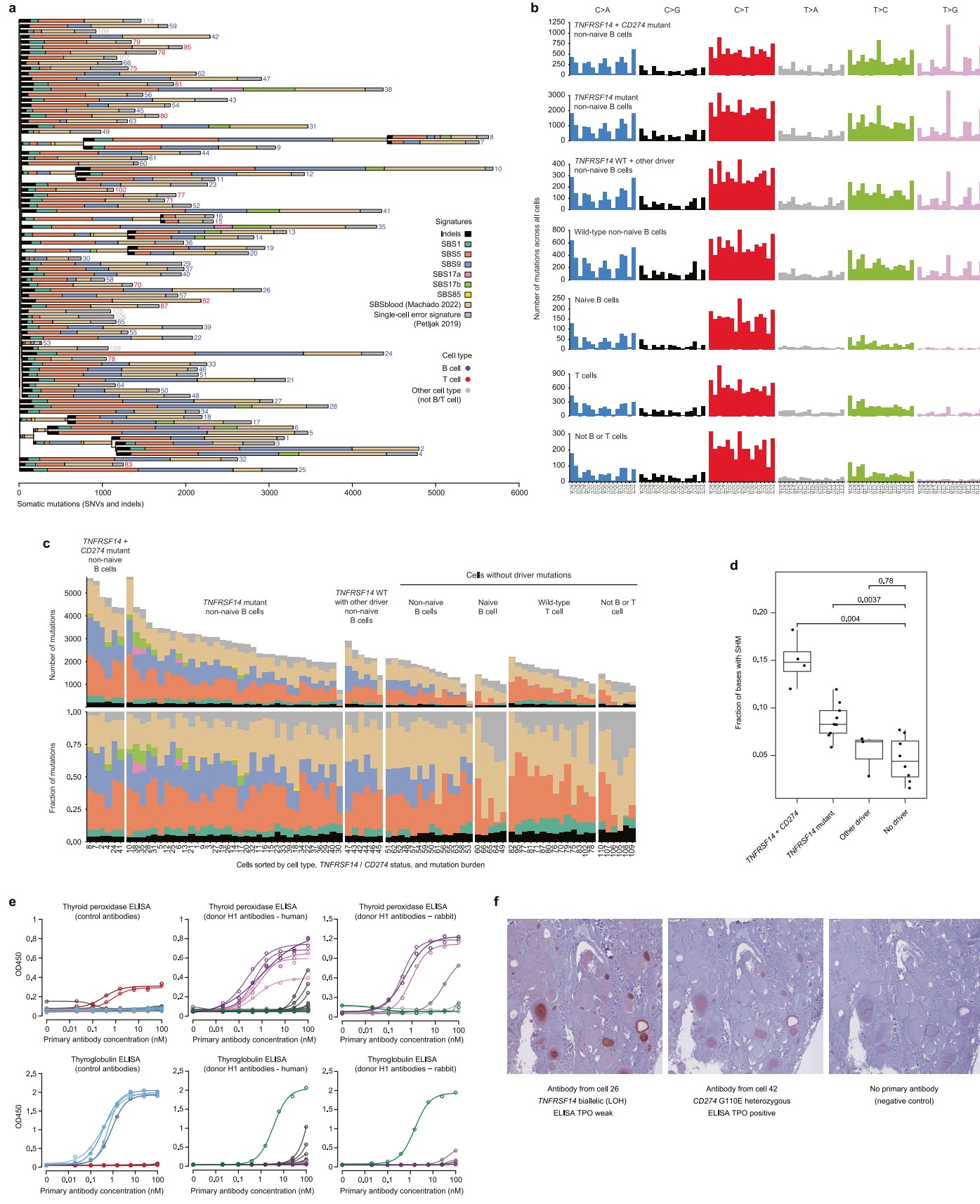

**Extended Data Fig. 11** | See next page for caption.

**Extended Data Fig. 11 | Mutational signature attribution and recombinant antibody ELISA results from single-nucleus sequencing data from donor H1. a**, Dendrogram of whole-genome single-nucleus sequencing data from donor H1 with each branch coloured by the proportions of attributed mutational signatures. Cell types assigned by V(D)J recombination and somatic hypermutation are indicated by the coloured cell number at the tips. **b**, Composite trinucleotide spectra for all mutations identified by whole-genome single-nucleus sequencing data for nuclei separated into cell type and driver mutation status categories. Peaks in the T > G channel are characteristic of a mutational signature associated with somatic hypermutation, SBS9. **c**, Stacked bar charts showing: (top) the total number of mutations assigned to each signature, and (bottom) the proportion of mutations assigned to each signature for nuclei separated into cell type and driver mutation status categories. Within each category, cells are sorted by decreasing mutation burden. **d**, Fraction of bases affected by somatic hypermutation (SHM) in reconstructed B cell receptors from non-naive B cells. *P*-values from Wilcoxon tests comparing mutant non-naive B cells to wild-type non-naive B cells. Only one cell per clade was selected. **e**, Dose response curves for ELISA to test binding to thyroid peroxidase (TPO) (top) or thyroglobulin (TG) (bottom) by recombinant antibodies generated using reconstructed immunoglobulin light and heavy chain sequences from single-nucleus sequencing for donor H1. Dose response curves are shown for: (left) recombinant positive control antibodies (red = known to bind TPO; blue = known to bind TG); (middle) recombinant antibodies from reconstructed BCR sequences from donor H1 on a human IgG1 backbone (purple = found to bind TPO; green = found to bind TG); and (right) recombinant antibodies from reconstructed BCR sequences from donor H1 on a rabbit IgG1 backbone (same colouring as middle panel). The grey line showing weak binding in the rabbit IgG1 backbone is the recombinant antibody generated from cell 26. **f**, Immunohistochemistry of normal thyroid (N1) using the recombinant antibodies generated from reconstructed BCR sequences of cells 26 and 42 on a rabbit IgG backbone as a primary antibody, along with a negative control (no primary antibody).

# Reporting Summary

## Statistics

For all statistical analyses, confirm that the following items are present in the figure legend, table legend, main text, or Methods section.

| n/a | Confirmed | |
|---|---|---|
| ☐ | ☒ | The exact sample size (*n*) for each experimental group/condition, given as a discrete number and unit of measurement |
| ☒ | ☐ | A statement on whether measurements were taken from distinct samples or whether the same sample was measured repeatedly |
| ☐ | ☒ | The statistical test(s) used AND whether they are one- or two-sided *Only common tests should be described solely by name; describe more complex techniques in the Methods section.* |
| ☐ | ☒ | A description of all covariates tested |
| ☐ | ☒ | A description of any assumptions or corrections, such as tests of normality and adjustment for multiple comparisons |
| ☐ | ☒ | A full description of the statistical parameters including central tendency (e.g. means) or other basic estimates (e.g. regression coefficient) AND variation (e.g. standard deviation) or associated estimates of uncertainty (e.g. confidence intervals) |
| ☐ | ☒ | For null hypothesis testing, the test statistic (e.g. *F*, *t*, *r*) with confidence intervals, effect sizes, degrees of freedom and *P* value noted *Give P values as exact values whenever suitable.* |
| ☐ | ☒ | For Bayesian analysis, information on the choice of priors and Markov chain Monte Carlo settings |
| ☒ | ☐ | For hierarchical and complex designs, identification of the appropriate level for tests and full reporting of outcomes |
| ☒ | ☐ | Estimates of effect sizes (e.g. Cohen's *d*, Pearson's *r*), indicating how they were calculated |

*Our web collection on statistics for biologists contains articles on many of the points above.*

## Software and code

Policy information about availability of computer code

| Data collection | No software was used for data collection. |
|---|---|
| Data analysis | Nanorate sequencing data analysis utilised the NanoSeq pipeline (v3.3.0, https://github.com/cancerit/NanoSeq). Targeted methylation analyses were performed with the UXM_deconv pipeline (https://github.com/nloyfer/UXM_deconv/) and the EpiDish R package (https://github.com/sjczheng/EpiDISH/tree/devel, v2.23.1), using the Robust Partial Correlations method and our modified atlas. |

<br>

Bespoke analyses were performed with R (v4.5) and relevant code can be found as Supplementary Code (HTML). R packages used include: tidyverse (v2), GenomicRanges (v1.62.1), dndscv (v0.0.1.0), scales (v1.4.0), patchwork (v1.3.2), viridis (v0.6.5), RColorBrewer (v1.1-3), lattice (v0.22-6), latticeExtra (v0.6-31), vcfR (v1.15.0), MASS (v7.3-65), jsonlite (v2.0.0), ggforce (v0.5.0), stringi (v1.8.7), gtools (v3.9.5), drc (v3.0-1), pander (v0.6.6), ape (v.5.8-1), ggtree (v4.0.4), ggh4x (v0.3.1) and ggpubr (v0.6.2).

<br>

For single nuclei data, all analyses are available on GitHub (https://github.com/alextidd/nicola_et_al_2026). For the single nuclei data, the BaseJumper tool suite was used (https://github.com/alextidd/bj-somatic-variantcalling, commit hash: 2a5dbfd; https://github.com/alextidd/bj-dna-qc, commit hash: 3149537). A Nextflow pipeline was designed to genotype SNPs and NanoSeq mutations, and to calculate coverage in V(D)J regions of single nuclei (https://www.github.com/alextidd/nf-resolveome, commit hash: a785010).

<br>

Full code repository for all analyses included in the manuscript will be available at https://doi.org/10.5281/zenodo.19366069.

<br>

Bespoke single nuclei analyses were performed with R (v4.4.0). R packages used include: magrittr (v2.0.4), tidyverse (v2.0.0), data.table (v1.18.2.1), ape (v5.8), patchwork (v1.2.0), RColorBrewer (v1.1-3), lsa (v0.73.3), slider (v0.3.2), ggh4x (v0.2.8), janitor (v2.2.0), knitr (v1.51), seqinr (v4.2-36), VGAM (v1.1-12), MASS (v7.3-60.2), devtools (v2.4.6), optparse (v1.7.5), hdp (v0.1.5), sigfit (v2.2), GenomicRanges (v1.56.1),

rtracklayer (v1.64.0), biomaRt (v2.60.1), Rsamtools (v2.20.0), ggtree (v3.12.0), BiocManager (v1.30.26) and treemut (v1.1).

Phylogeny construction was performed using the Sequoia package with MPBoot (v1.1.0). Mutational signature extraction was undertaken with hdp (v0.1.5, https://github.com/nicolaroberts/hdp) and SigProfiler (https://cancer.sanger.ac.uk/signatures/tools/). De novo signatures were deconvoluted to reference signatures in COSMIC v3.5 database (https://cancer.sanger.ac.uk), and other published signatures. Signature fitting was completed with sigfit (v2.2, https://github.com/kgori/sigfit). Further detail can be found in Supplementary Note 2.

For manuscripts utilizing custom algorithms or software that are central to the research but not yet described in published literature, software must be made available to editors and reviewers. We strongly encourage code deposition in a community repository (e.g. GitHub). See the Nature Portfolio guidelines for submitting code & software for further information.

## Data

Policy information about availability of data

All manuscripts must include a data availability statement. This statement should provide the following information, where applicable:

- Accession codes, unique identifiers, or web links for publicly available datasets
- A description of any restrictions on data availability
- For clinical datasets or third party data, please ensure that the statement adheres to our policy

All sequencing data has been deposited in EGA and is available within the following 8 datasets:
EGAD00001016058 Polyclonal selection of immune checkpoint mutations in thyroid autoimmunity - ExomeNanoSeq
EGAD00001016059 Polyclonal selection of immune checkpoint mutations in thyroid autoimmunity - TargetedNanoSeq
EGAD00001016060 Polyclonal selection of immune checkpoint mutations in thyroid autoimmunity - TargetedEMSeq
EGAD00001016061 Polyclonal selection of immune checkpoint mutations in thyroid autoimmunity - PTA_WGS
EGAD00001016062 Polyclonal selection of immune checkpoint mutations in thyroid autoimmunity - PTA_DNAHyb
EGAD00001016063 Polyclonal selection of immune checkpoint mutations in thyroid autoimmunity - PTA_RNA
EGAD00001016064 Polyclonal selection of immune checkpoint mutations in thyroid autoimmunity - LCMB_WGS
EGAD00001016065 Polyclonal selection of immune checkpoint mutations in thyroid autoimmunity - LCMB_WES

Xenium data can be found on the EMBL-EBI BioImage Archive (https://www.ebi.ac.uk/biostudies/bioimages/studies/S-BIAD3103, accession number S-BIAD3103, doi 10.6019/S-BIAD3103)

## Research involving human participants, their data, or biological material

Policy information about studies with human participants or human data. See also policy information about sex, gender (identity/presentation), and sexual orientation and race, ethnicity and racism.

| Reporting on sex and gender | We report sex, not gender, only for the purposes of calculating mutant cell fraction for X-linked genes (as this will vary in homozygous and hemizygous contexts). |
| --- | --- |
| Reporting on race, ethnicity, or other socially relevant groupings | We do not make any report on ethnicities or other socially relevant groupings. |
| Population characteristics | AITD cohort: 14 donors, 4 male, 10 female, age range from 10 - 20 years old to 70 - 79 years old.<br>Normal PBMC cohort: 28 donors, 13 male, 15 female, median age = 70 years old, range = 20 - 91 years old.<br>Microdissected lymphoid aggregate cohort: 5 donors, 3 male, 2 female, median age = 47 years old, range = 19 - 72 years old<br>Lymph node cohort: 4 donors, 2 male, 2 female, median age = 39 years old, range = 19 - 69 years old<br>Spleen cohort: 30 donors, 13 male, 17 female, median age = 50 years old, range = 20 - 69 years old<br>Tonsillitis cohort: 5 donors, 2 male, 3 female, median age = 28 years old, range = 20 - 38 years old<br>Thyroid goitre cohort: 5 donors, 1 male, 4 female, median age 71 years old, range = 56 - 80 years old<br>Normal thyroid cohort: 3 donors, 1 male, 2 female, median age = 47 years old, range = 19 - 54 years old<br><br>Associated metadata includes clinical diagnosis and autoantibody status (anti-TPO) where available. |
| Recruitment | AITD cohort: Donors were selected on the basis of having at least one of: a clinical diagnosis of autoimmune thyroid disease (either Hashimoto disease or Graves disease), histopathological evidence of intra-thyroidal lymphocytic infiltration, and positive serology for anti-TPO. Samples were provided in the course of thyroid biopsy or resection and donors gave consent for research purposes..<br>Normal PBMC cohort: Donors with available PBMC samples who were over 50 years old and without a clinical history of autoimmunity were purchased from a commercial supplier.<br>Microdissected lymphoid aggregate cohort: Donors with available tissue samples (tonsil, ileum, sigmoid colon) and without a clinical history of autoimmunity were purchased from a commercial supplier. Laser capture microdissection was used to excise lymphoid aggregates where present<br>Lymph node cohort: Donors with available tissue samples and without a clinical history of autoimmunity were purchased from a commercial supplier.<br>Spleen cohort: Three donors with available tissue samples and without a clinical history of autoimmunity were purchased from a commercial supplier. A further 27 donors were eligible organ transplant donors who donated tissue for research purposes. All donors had no clinical history of cancer or autoimmune disease.<br>Tonsillitis cohort: Donors with available chronically inflamed tonsil samples and without a clinical history of autoimmunity were purchased from a commercial supplier.<br>Thyroid goitre cohort: Donors were selected by having available thyroid tissue and a clinical and histopathological diagnosis of a nodular thyroid goitre. Samples were provided in the course of thyroid biopsy or resection and donors gave consent for |

research purposes.
Normal thyroid cohort: Donors with available tissue samples and without a clinical history of autoimmunity or other thyroid disease were purchased from a commercial supplier.

Ethics oversight

The use of these samples was approved by the South West - Central Bristol Research Ethics Committee (REC 19/SW/0031). The use of these samples was approved by the London - Surrey Research Ethics Committee (REC 17/LO/1801). The use of these samples was approved by the South Central - Oxford B Research Ethics Committee (REC 21/SC/0158). The use of these samples was approved by the East of England - Cambridge South Research Ethics Committee (REC 15/EE/0152).

Note that full information on the approval of the study protocol must also be provided in the manuscript.

# Field-specific reporting

Please select the one below that is the best fit for your research. If you are not sure, read the appropriate sections before making your selection.

☒ Life sciences  ☐ Behavioural & social sciences  ☐ Ecological, evolutionary & environmental sciences

For a reference copy of the document with all sections, see nature.com/documents/nr-reporting-summary-flat.pdf

# Life sciences study design

All studies must disclose on these points even when the disclosure is negative.

| | |
|---|---|
| Sample size | Information on sample sizes is provided for all analyses. The cohort size was limited by the scarcity of available inflamed thyroid samples across several sources and budgetary considerations. |
| Data exclusions | No samples were excluded from this study. |
| Replication | We haven't conducted explicit replication as this was not applicable for an observational study |
| Randomization | No randomization was performed as this was not applicable for an observational study |
| Blinding | No blinding was undertaken because this is an observational study. |

# Reporting for specific materials, systems and methods

We require information from authors about some types of materials, experimental systems and methods used in many studies. Here, indicate whether each material, system or method listed is relevant to your study. If you are not sure if a list item applies to your research, read the appropriate section before selecting a response.

## Materials & experimental systems

| n/a | Involved in the study |
|---|---|
| ☐ | ☒ Antibodies |
| ☒ | ☐ Eukaryotic cell lines |
| ☒ | ☐ Palaeontology and archaeology |
| ☒ | ☐ Animals and other organisms |
| ☒ | ☐ Clinical data |
| ☒ | ☐ Dual use research of concern |
| ☒ | ☐ Plants |

## Methods

| n/a | Involved in the study |
|---|---|
| ☒ | ☐ ChIP-seq |
| ☐ | ☒ Flow cytometry |
| ☒ | ☐ MRI-based neuroimaging |

# Antibodies

Antibodies used

Anti-CD20 primary antibody (PA0359, Leica Biosystems, Germany)
Anti-CD3 primary antibody (PA0122, Leica Biosystems, Germany)
Anti-BCL6 primary antibody (PA0204, Leica Biosystems, Germany)
Fluorescein-conjugated oligonucleotide probes targeting Kappa and Lambda light chain mRNA (PB0645 and PB0669, Leica Biosystems, Germany)
Anti-fluorescein antibodies (AR0222, Leica Biosystems, Germany)
Anti-CD19 primary antibody (ab270715, Abcam, UK)
Anti-CD3 primary antibody (ab11089, Abcam, UK)
Anti-TNFRSF14 primary antibody (ab314494, Abcam, UK)
Anti-thyroglobulin (ab156008, Abcam, UK)
Anti-thyroid peroxidase (ab109383, Abcam, UK)
Anti-CD19 primary antibody (14-0194-82, Invitrogen, Thermo Fisher Scientific Inc, Waltham, MA, USA)
Anti-TNFRSF14 primary antibody (ab314494, Abcam, UK)

Anti-Rat IgG H&L secondary antibody (Alexa Fluor® 568, ab175476, Abcam, UK)
Anti-Rabbit IgG H&L secondary antibody (Alexa Fluor® 647, ab150083, Abcam, UK).

Custom recombinant antibodies were generated by Biointron Biological Inc (Shanghai, China) based on reconstructed heavy and light chain sequences. Positive control nucleotide sequences were obtained from the literature, by searching GenBank for paired heavy and light chain sequences known to bind human TPO or thyroglobulin: AJ238327.1 and AJ238330.1 (clone ICA5, anti-human TPO); AJ399834.1 and AJ399876.1 (clone T8, anti-human TPO); AY365327.1 and AY365334.1 (clone #6, anti-human thyroglobulin); AY365330.1 and AY365338.1 (clone #26, anti-human thyroglobulin).

For ELISA, recombinant antigens used were: human thyroid peroxidase (NM_000547, OriGene, MD, USA, TP310659) and human thyroglobulin (NM_003235, OriGene, MD, USA, TP316216). Recombinant monoclonal antibodies and isotype control antibodies were also used (Biointron Biological Inc, Shanghai, ChinaBiointron, B117901, B646501, B730001). Horseradish peroxidase-conjugated secondary antibodies: goat anti-human IgG (Fc-specific) (Sigma-Aldrich, Merck KGaA, Germany A0170, 1:10,000 dilution) or goat anti-rabbit IgG antibody (Arigo Biolaboratories, Taiwan, China, ARG65351, 1:10,000 dilution).

| Validation | Data provided in the manuscript and on manufacturer's website. |
|---|---|

## Plants

| Seed stocks | *Report on the source of all seed stocks or other plant material used. If applicable, state the seed stock centre and catalogue number. If plant specimens were collected from the field, describe the collection location, date and sampling procedures.* |
|---|---|
| Novel plant genotypes | *Describe the methods by which all novel plant genotypes were produced. This includes those generated by transgenic approaches, gene editing, chemical/radiation-based mutagenesis and hybridization. For transgenic lines, describe the transformation method, the number of independent lines analyzed and the generation upon which experiments were performed. For gene-edited lines, describe the editor used, the endogenous sequence targeted for editing, the targeting guide RNA sequence (if applicable) and how the editor was applied.* |
| Authentication | *Describe any authentication procedures for each seed stock used or novel genotype generated. Describe any experiments used to assess the effect of a mutation and, where applicable, how potential secondary effects (e.g. second site T-DNA insertions, mosiacism, off-target gene editing) were examined.* |

## Flow Cytometry

### Plots

Confirm that:

☒ The axis labels state the marker and fluorochrome used (e.g. CD4-FITC).

☒ The axis scales are clearly visible. Include numbers along axes only for bottom left plot of group (a 'group' is an analysis of identical markers).

☒ All plots are contour plots with outliers or pseudocolor plots.

☒ A numerical value for number of cells or percentage (with statistics) is provided.

### Methodology

| Sample preparation | Nuclei dissociation was performed using a modified Slide-tags protocol, a single-nucleus barcoding technique developed for multimodal spatial genomics. The main deviation from this protocol was the absence of mounting tissue sections on the proprietary pucks necessary for spatial mapping. Dissociated nuclei were stained with a fluorescent intercalating dye (propidium iodide) for 30 minutes at 4 degrees and then filtered through a 30 µm filter prior to flow sorting. |
|---|---|
| Instrument | Single nuclei were sorted using a Bigfoot spectral cell sorter (Invitrogen, Thermo Fisher Scientific Inc, Waltham, MA, USA). |
| Software | Sorting was carried out using Sasquatch (SQ) software (Invitrogen, Thermo Fisher Scientific Inc, Waltham, MA, USA). Representative gating plots were prepared using FCS Express 7 Research. |
| Cell population abundance | Flow sorting was only being used to separate intact individual nuclei from doublets, non-nuclear debris and nuclei that were not intact due to cryosectioning. Representative proportions of intact nuclei are shown in Extended Data Fig. 6a. |
| Gating strategy | Propidium iodide was excited using a 561 nm laser and emission collected using a 625/15 bandpass filter. Gating was set on PI (area) vs PI (width) to discriminate doublets. Single nuclei were sorted into pre-calibrated 96 well plates on a single sorting mode. Representative gating strategy is shown in Extended Data Fig. 6a. |

☒ Tick this box to confirm that a figure exemplifying the gating strategy is provided in the Supplementary Information.

