## [Peer Review File · Nature]

Polyclonal selection of immune checkpoint mutations in thyroid autoimmunity

Corresponding Author: Dr Inigo Martincorena

Co-Corresponding Author: Dr Andrew R.J. Lawson

Version 0:

Reviewer comments:

Referee #1

(Remarks to the Author)

Nicola, Lawson, Martincorena et al follow up on their recent studies in which they mapped thousands of somatic mutations in oral, blood, and oesophageal samples, revealing clones with extensive selection patterns and risk factors. Here they use the same ultra-accurate sequencing method (NanoSeq) and additional sequencing methods to sequence B cells from thyroid of patients with autoimmune thyroid disease and visible germinal centers in tissue biopsies, to test the hypothesis that “somatic mutations in immune regulatory genes may enable self-reactive lymphocytes to bypass tolerance checkpoints”. They show B cell clones sharing LoF mutations in TNFRSF14 and CD74 (PD-L1), some with biallelic and/or several mutations in the same gene. Their conclusion is that “their studies support the hypothesis that somatic mutations in autoimmune lymphocytes allow them to escape tolerance constraints” and “provide new insights into the molecular basis of autoimmune disease.

These are interesting and striking observations that also build on earlier foundational work by David Schatz and others, who showed over two decades ago that BCL6-positive germinal center B cells can acquire mutations in non-BCR genes due to off-target activity of AID during somatic hypermutation. The present study advances this concept by showing that specific mutations may indeed confer a selective advantage, likely promoting their recycling to undergo multiple rounds of somatic hypermutation within germinal centers. There are two major concerns: The first is that the study suffers from a lack of proper controls – most notably tissues (not blood) from healthy controls, including tissues with and without chronic germinal center reactions. The second is that the framing of the conclusions implies a causal link between somatic mutations and B cell-driven autoimmunity, but this has not been established. Whilst the study describes selected somatic mutations in a tissue that is target of autoimmune disease, it does not provide proof that these mutations are causally related with disease.

Major concerns:

General:

- The data are consistent with a model in which somatic mutations contribute to germinal center B cell persistence or fitness, but there is no direct evidence provided that the observed mutations give rise to autoantibody-secreting plasma cells and thus to autoimmunity. It would be important to show i) the positively selected mutations are found in plasma cells and ii) the mutated plasma cells produce autoantibodies.
- The choice of controls – PBMCs that thus lack germinal center B cells and tissue resident cells - also limits the strength of the comparisons. A more informative baseline might be germinal centers from healthy lymphoid tissues (e.g., tonsils, mesenteric lymph nodes, or Peyer's patches), where similar mutational patterns could plausibly emerge as a feature of chronic stimulation. This distinction is important given existing evidence that chronic germinal center activity, even in autoimmunity-prone settings such as lupus, does not necessarily result in pathogenic autoantibody production: in both murine and human studies, anti-nuclear antibodies often arise from extrafollicular responses. Moreover, as the authors themselves mention, mechanisms exist within germinal centers to negatively select or redeem self-reactive clones, so they cannot leave germinal centres and differentiate into plasma cells.
- The data showing that B cells as opposed to other cells abundant in their biopsies are the cells carrying the somatic mutations is weak. Only the last figure addresses this for one patient. The authors sequence biopsies that contain large B cell clusters likely to contain germinal centres. It would be important to add germinal center markers and sequence cells

within them and compare them with areas that do not contain germinal centers in the same gland.

- The Figures are difficult to read and interpret and the figure legends do not provide sufficient information. This contributes to the longer-than-usual list of questions and issues as described below that require clarification.

Figure by Figure

Figure 1

- This figure describes ALL cells in thyroid samples from AITD patients. The authors should acknowledge that a significant proportion of mutations may occur in non-B cells and non-lymphoid cells.
- A healthy thyroid tissue control would be important, particularly when the frequency of somatic mutations is so variable between 3 AITD patients. Most mutations are identified in H1 while H2 and H3 have much lower levels. This should be clearly acknowledged in the text.
- Details are needed to understand prioritisation of the 4 genes mentioned. It is difficult to follow. How much above 1 does dN/dS values have to be to be considered meaningful? dN/dS values don't seem that high for TET2 and TNFAIP3 (Figure 1d). Are these true enrichments? What is the enrichment in dN/dS values in each patient? Again, healthy controls would be needed.
- Figure 1E: Not clear whether the “upper and lower bounds” refer to statistical uncertainty with the method used or upper and lower bounds between the three patients. The three patients shown in Figure 1F (and extended figure 1) seem to have much higher (H1) and lower (H2 and H3) bounds than what is presented in Figure 1E e.g. For TNFRSF14. Same in Figure 2c and other places in the manuscript: what is “upper and lower” bounds?
- Figure 1F: Is not referenced in the text.
- Text line 136/137: “Although each mutation is only present in a small proportion of cells, cumulatively across many clones the estimated fraction of mutant cells in these biopsies reaches remarkably high levels. For donor H1, at least 32.7% and 5.0% of all cells bear TNFRSF14 and CD274 mutations respectively (Methods). The other donors had lower, but still substantial, TNFRSF14 (H2 = 3.0%, H3 = 5.5%) and CD274 (H2 = 4.4%, H3 = 0.3%) mutant cell fractions”
 - o For donor H3 the level of CD274 mutant cell fraction seems barely detectable. There seems to be a high level of variability between patients (also seen in later figure) that the authors should acknowledge. TET2 and TNFAIP3 also seem very low across all donors. Levels in healthy tissue would be needed.
- Can lower levels of HVEM and PDL1 protein be detected on affected areas? Also relevant to Figure 4/ Extended Figure 6, show which areas are selected for imaging and across multiple patient samples or areas.

Figure 2

- Figure 2b: Targeted methylation sequencing is not mentioned in the text. Hard to follow what the authors did when it only appears in figure legends, or much later than where the results are described.
- Figure 2c:
 - o Figure legend states that grey denotes genes that reach significance only due to SHM – but earlier the authors describe that they developed a method that takes SHM into account and so this should not happen? How do the authors distinguish genes that are due to SHM from the other genes in black?
- Figure 2d:
 - o The text says (line 207/208) “Clones with mutations in immune checkpoint genes were observed across biopsies from all donors with sufficiently high lymphocyte abundance and depth of sequencing (Fig. 2d).”
 - o The plot shows combined pilot and extension datasets. The extension dataset looks less convincing - it does not confirm the results found in the pilot: The only donor that has detectable levels of non-synonymous mutations is HB – the rest have 0-9 non-synonymous mutations in the two top genes.
 - o There is a high degree of patient variability that does not seem to be explained by the degree of lymphocyte infiltration. Can the authors please clearly state this and discuss?
 - o Although patients are sorted by lymphocyte infiltration, it does not seem like there is clear correlation between lymphocyte infiltration and mutation counts. E.g., patient G3 has higher lymphocyte infiltration but low mutation counts. Patient H9 has lower lymphocyte infiltration but higher mut counts. Patient H5 sample did undergo laser capture to capture more lymphocytes and should thus have higher lymphocyte infiltration and a higher chance of finding mutations, yet none were observed for TNFRSF14. For Patient G3 (also laser capture) found 1 in TNFRSF14.
 - o Reading of the figure would be helped by clearly stating in the figure legend that mutation cell fraction is in colour and that the number refers to non-synonymous mutations.
- Text (line 213/214) “Importantly, TNFRSF14, CD274 and TNFAIP3 were also significant in the extension cohort alone, validating them in an independent cohort” – However no data for this is shown. How are these genes also significant? By dN/dS ratio?
- Extended data figure 1b: How is the % of lymphocytes determined? By nanoseq alone? Or by targeted methylation seq. which is not described? Please indicate the colour-scale bar for the cell fraction carrying a mutation and mention clearly in the figure legend what the numbers and the colours mean.

Figure 3

It is interesting that there seems to be selection for truncating mutations (mentioned also earlier in the text) and in cysteine-cysteine bonds, but it is hard to understand the significance of this finding as there are no controls and despite great variability between patients, the figure shows all patients pooled together. Could the authors analyse whether there would be an enrichment in these mutations on a patient-by-patient basis?

Are the findings in Figure 3b and Ext. data Fig 3h significant? Not mentioned in legends.

Text (line 240/241): “cause C-terminal truncation, resulting in reduced desensitisation and internalisation of CCR6 (mediated by β -arrestin) and upregulation of CCL20-mediated NF- κ B signalling”. Despite published information on these proteins the

exact effect of the mutations is speculative. The authors do not show experimentally the effect of the mutations and so conclusions on the effects must be toned down.

Extended data figure 3: While it is interesting that for some genes the hotspot seems to be the same as in leukaemia, it is again difficult to judge the significance due to lack of healthy controls.

IRF1 and CD45 are mentioned, but it is unclear how these would lead to autoimmunity if LOF?

Text (Line 274-279) and 382/384: The authors state themselves that there is selection for several of the “driver” genes during normal ageing. The patients used here are also not that young, so how can the authors conclude that this is specific to AITD without controls? This also weakens the argument that the somatic mutations indeed are “driver” mutations

Text (Line 286/287): “This is in stark contrast to the high number of driver mutations in TNFRSF14 and CD274 in the AITD samples, as both genes are very rarely mutated in non-autoimmune lymphocytes”. This is an overstatement with only 2-3 patients having a high level of these mutations.

Figure 3c: Duplex coverage is also very different in the two data sets. i.e., Not a one-to-one comparison

Figure 4

The authors need to tone down their conclusions. Conclusions often described as “revealed that” in definitive terms when results are only suggestive.

Figure 4a: It cannot be concluded that there two different clones. There is some overlap in the microdissections with the two types of mutations, although essential splice enriched at the top and frameshift enriched at the bottom. No definite conclusion can be reached either on whether the mutations occur in a lymphocyte; non-lymphoid tissue has not been sequenced for comparison.

Why were only 3 mutations from Nanoseq re-found? Sequencing depth?

What about the lymphoid cluster at the bottom of the tissue. Unaffected?

Figure 4b: Text (Line 319/320). The figure and the text need to be reordered for clarity. The text jumps very suddenly to conclude that the lymphoid-enriched structures are germinal centres without first showing this. Figure 4e needs to be shown before 4b, and possibly Extended Figure 6; specific germinal centre stains are needed.

Extended figure 6: Please show which “GCs” / areas these images are from. e.g, Images with lower magnification. Text mentions CD3 staining, but there is no CD3 staining in the figure. CD19 staining is not enough to claim that these are GCs. CD19 staining seems to be less bright in mutant “GCs” than in WT “GC” so apparently less bright TNFRSF14 staining might stem from lower quality tissue

Figure 4e: Xenium data should should be shown before clonal data, to first show that these are likely germinal centres.

Figure 4e is mentioned in the text before 4c and 4d. Should also show T cells as these are also likely to contribute to disease

Text: “Plasma cells are the most common, consistent with figure 1a”. However, no plasma cell stain in figure 1a. Are conclusions drawn from Igk/Igl staining? Igk/Igl deposition could be immunocomplexes. It is difficult to interpret whether IgK/IgI+ stains are indeed plasma cells when image is not magnified and no CD138 staining is included. In addition, the tissue in Fig. 1a looks very different from what is shown in Figure 4e. In Figure 1a: Large B cell follicles with Bcl6 staining (i.e., germinal centre B cells) and T cell areas. Figure 4e shows a very different picture with no clear B cell or T cell areas. Why is the structure so different?

Text (Line 320): authors mention single or small number of clones, however, two germinal centres have 3 clones (out of 7 identified). 2488106G>A is found everywhere and might not even be specific to “GCs”

Many germinal centres or microdissections do not contain any mutant clones. Could the authors please mention and comment in the text?

Additionally, a comparison between microdissections with and without mutant clones is missing. Without it, it is difficult to interpret whether these mutations are indeed driving anything.

Figure 4d: Text (Line 336): “dominated by a single clone” although there still is a high degree of overlap between mutations. Could be multiple clones. i.e., Conclusions in line 338/340 on whole genome sequencing (WGS) cannot be drawn on 1 clone. WGS1 seems to have larger VAF of p.Y26D than Q345* and pM11 and pW12* – indicating that it is not just one clone, but multiple.

Figure 5

Text (Line 360): Could the authors please clarify the text for readability. Mention first that sequenced cells are classified into B, T, and other and based on which characteristics. Then, mention how many mutations were found in each.

Mutations identified in T cells and non-lymphocytes need to be mentioned.

Extended Figure 7: Should show a comparison between mutated and unmutated B cell clones: are SHM levels the same? Class switch recombination? (Text says 90.2% but this is not shown in the figures). Again, controls are missing, making it hard to get an idea of how unique this is and if relevant.

Figure 5a: Text says that mutant clones are highly polyclonal but they actually seem to be less polyclonal than somatically unmutated B cells. Each somatic mutation combination has one set of VDJ combinations. For instance, TNFRSF14 Y26D

and Y25Y(?) clones all have V1-48 and J-4.

Do the authors suggest that somatic mutations occurred during early embryogenesis, during later somatic mutations in stem cells or after B cell development/VDJ-recombination? Line 369 suggests early embryogenesis; Line 388 suggests during first decade of life (why first decade is unclear); From line 388 and onwards, authors describe somatic mutations to occur after B cell maturation, ie not independent lineages, which seems to conflict with earlier statements.

If indeed an embryonically introduced somatic mutation conferred advantage, one would expect to find clones of different VDJ combinations, meaning that somatic mutation occurrence AND advantage are independent of VDJ recombination. By contrast, their results are more consistent with somatic mutations occurring in single VDJ-clones through germinal center-related somatic hypermutation (or through another process) happening after VDJ-recombination.

Text (Line 405/406): "Altogether, these data provide evidence of strong selection driving large numbers of clonally independent B cells to escape peripheral tolerance constraints during the pathogenesis of autoimmune thyroid disease." This is an overstatement not backed up by data. There seem to be alternative explanations for these results, no healthy controls, large degree of variability within patients but analysis focused on one with strong TNFRSF14 mutation load. Mutations seem to have occurred to a large extent (with exception of two clones with shared TET2 mutation) after VDJ-recombination. Most importantly, there is no evidence that these mutated B cells leave germinal centers and go on to differentiate into plasma cells and produce autoantibodies.

Discussion: Text (Line 430): "extreme convergence. Tens-hundreds of clones per donor" This is not shown for B cells but all cell types.

(Remarks on code availability)

Referee #2

(Remarks to the Author)

I co-reviewed this manuscript with one of the reviewers who provided the listed reports.

(Remarks on code availability)

Referee #3

(Remarks to the Author)

Nicola et al. present a compelling study that describes the somatic mutation landscape of lymphocytes in autoimmune thyroid disease and provides evidence of recurrent mutations in B cell clones targeting immune-checkpoint pathways in the course of disease evolution. Using highly sensitive duplex exome and targeted NanoSeq with orthogonal single-nucleus DNA sequencing and spatial transcriptome approaches the authors identify a high burden of mutations that converge on TNFRSF14 and CD274 within B lymphocytes. Their model supports stepwise polyclonal mutagenesis in which numerous B cells accrue lesions, including biallelic loss of TNFRSF14, consistent with selection for checkpoint evasion across infiltrating B cells. While the work certainly sets a high bar for technical innovation and substantially advances our understanding of the link between somatic mutations in lymphocytes and autoimmune pathogenesis, several aspects of interpretation of the data would benefit from further refinement. Furthermore, while the advance is substantial, the technical limitations of the approach preclude dissection of the specific nature of B lymphocytes targeted by the somatic mutations (immature/mature/germinal center/terminally differentiated).

Major Concerns:

1. Figure 1a – why does the anti-Kappa/lambda IHC staining not identify the same B cell regions as the CD20 staining does? Please clarify concordance or discuss biological/technical reasons for discordance.
2. As presented, the data don't resolve whether mutations arise/are selected in immature, GC, memory, or terminally differentiated plasma cells. I believe it is a crucial point to address, if at all possible. Please explain if this deconvolution can be performed and discuss the plausible selection pressures for TNFRSF14/CD274 lesions across these compartments.
3. Given that the small number of patients on which duplex sequencing and NanoSeq was performed, independent replication with predefined mutation/CNV search in larger number of patients that includes earlier stages of disease is appropriate. Doing this across different stages of disease could begin to address dynamics of accumulation and clonal evolution.
4. Single nucleus sequencing enabled the authors to couple the analysis of B cell VDJ repertoire to somatic mutations and provided critical insight into whether the mutations accumulated in clonally (by BCR) expanded B cells. Analysis of additional patient biopsies with snDNA-VDJ seq (or 5' GEX plus targeted amplicons or using other methods) is critical to determine mutation prevalence within and across expanded clonotypes.
5. Extended figures need complete legends (sample number, statistical tests, all annotations, etc).
6. Several methods defer to "Lawson et al., in preparation." Please include sufficient detail here (or Supplementary Methods) to permit evaluation/future reproduction.

Minor Concerns

1. Line 66 – Consider "neglect" rather than "ignorance"?
2. Line 94 – Consider "most common" instead of "commonest"?
3. Figure 4E – Define criteria used to separate mature versus immature plasma cells.

4. Please describe how TNFRSF14 and CD274 mutations were assigned to T cells and what the expected consequences are. What is the T cell clonality and driver mutational profile for these events? Please defined the confidence in assignment of lymphocyte identity across ST vs. B/T aggregates.

(Remarks on code availability)

Referee #4

(Remarks to the Author)

I co-reviewed this manuscript with one of the reviewers who provided the listed reports.

(Remarks on code availability)

Referee #5

(Remarks to the Author)

This manuscript by Nicola, Lawson and colleagues leverages ultra-deep high-accuracy duplex sequencing and other technologies to assay rare mutations in autoimmune thyroid disease. In thyroid tissue from patients with Hashimoto thyroiditis and Graves disease, afforded by their sequencing approach, the authors identify a high frequency of very low allele fraction somatic mutations in immune checkpoint genes in polyclonal lymphocyte populations. They demonstrate that non-synonymous mutations in TNFRSF14, CD274 and other immune checkpoint genes are under positive selection and that such mutations are not observed in aged B cells from a PBMC population. Using an orthogonal approach, they genotype these mutations in single cells and identify B cells as the population of interest. While each mutation is individually low frequency in the population, they are highly polyclonal and collectively affect a significant fraction of the B memory cell population. These likely contribute to Hashimoto and Graves pathology by abrogating autoinhibitory processes that govern self-reactive B cells. The unexpectedly rich landscape of polyclonal immune checkpoint mutations in B cells very likely plays a central role in the pathogenesis of Hashimoto and Graves.

By extension, this work tests the longstanding hypothesis that autoimmune diseases can arise through somatic mutation of genes in lymphocytes that negatively regulate self-reactive clones. The authors' results are highly impactful and change our understanding of autoimmune diseases, which will lead to many follow-up studies to validate the role of polyclonal somatic immune mutations across other autoimmune diseases, and to tease apart the precise role and timing of polyclonal mutant immune cells in disease etiology.

Overall, the study is very carefully performed, and the manuscript overall is clearly written. My one main reservation relates to the appropriateness of the negative controls of non-autoimmune B cell populations the authors used as a reference.

Major comments:

1. The lack of an appropriate negative control is a concern. While the authors did not observe the same positively selected mutations in normal PBMC memory B cells from aged donors (Fig. 3C), I feel this is an inadequate comparison. A more appropriate negative control would be a similarly local sample (biopsy or LCM) with proliferating B cells in place and possibly under selection from non-autoimmune donors. Such populations could be sampled from lymph nodes, spleen, or bone marrow. The mutations observed are already a polyclonal mix from such a small locale. Attempting to identify such mutations from the blood may significantly underestimate their presence in non-autoimmune implicated B cells. While the genes implicated conform to the hypothesis, an alternative interpretation, if these mutations are locally present in non-autoimmune tissue, is that these confer a selective advantage in B cell maturation but are not enough to drive pathogenesis of autoimmunity (i.e., other autoinhibitory mechanisms are present). It would add significantly to the study if the authors were able to close this gap.

Minor comments:

1. Related to the point above, it would also be valuable to see a set of negative control genes highlighted more clearly. At this depth of coverage and accuracy, what is the rate of mutation in genes not expressed in B cells or those that are expressed but not necessarily implicated in immunity per se (e.g., housekeeping genes)?

2. The repeated reliance on another forthcoming paper (Lawson et al, in preparation) to support several claims somewhat detracts from the present study. If these are not published together, data supporting claims made here, should be presented here or at least summarized to the level needed for this paper to stand on its own.

a. The authors refer to a modified version of dNdScv (dNdSshm) and refer to the other manuscript for details. They do describe adaptations to dNdScv in the methods section – is that dNdSshm? If not, this method should be described in broad terms there.

b. The authors claim “In a separate study [Lawson et al, In preparation], we show that several of these genes acquire driver mutations in B or T memory cells during normal immune ageing, in the absence of autoimmune disease” (line 274) is valuable information for this study, but no supporting data is shown.

3. The authors have spatial transcriptomics data from thyroid sections annotated with spatially distinct B cell populations. The lack of connection to changes in cellular phenotype despite having such data is a missed opportunity. Are there any transcriptional changes in B cell clusters with TNFRSF14 mutations when compared to adjacent populations without such a mutation? Could these data be used to confirm the reduction of TNFRSF14 or CD274 gene products in these populations?

4. Can the authors more clearly lay out differences between Hashimoto thyroiditis and Graves disease? These are now analyzed as one disease, yet they have an opposite effect on hormone secretion. Are there differences in genes under

selection, or in properties of mutant clones (e.g. clone size).

5. This is perhaps outside of the scope of the study, but it would underline the significance of the authors' findings if they could confirm if TNFRSF14 and CD274 mutant clones are indeed self-reactive. This could e.g. be addressed through a patient-derived thyroid organoid model, B cell isolation, and co-culture (PMID 34916298).

6. How are the cumulative fractions of mutant cells calculated? What assumptions go in there? E.g. are the authors implicitly assuming that mutant clones have only one mutation per cell here? This should be detailed in the methods section.

7. The manuscript at multiple positions jumps forward to later results, mainly to make the point early on that these mutations occur in B cells (e.g. lines 139-141). This is somewhat inelegant – it would be better to tell a linear story.

8. Lines 358-361 – in the single-cell sequencing section, the authors assay 112 cells and use an elegant approach to assign cell type. A few sentences to describe this could be added to the main text. E.g. it would be helpful to the reader to know that 66 of the cells are B cells to really follow what's written in lines 358-361.

9. The text and scale bars in the figures are often too small to read. Fig. 3c (and Ext. Data Fig. 4) needs a color legend for the significance boxes. Figure 4 is hard to follow with Zooming in extensively.

10. Data availability: the authors should make the raw sequencing data available through a repository.

(Remarks on code availability)

Referee #6

(Remarks to the Author)

I co-reviewed this manuscript with one of the reviewers who provided the listed reports.

(Remarks on code availability)

Version 1:

Reviewer comments:

Referee #1

(Remarks to the Author)

This reviewer was concerned with the evidence that the mutations cause disease. The first major concern raised highlighted the importance of showing positively selected mutations in antibody secreting plasma cells. The authors somewhat rephrased our request as “additional evidence of mutations occurring in B cells (...)” as the point to be addressed. They have now edited the text from “B cells” to “B cells or plasma cells” but have not provided additional evidence that these mutations indeed are found in plasma cells.

The only evidence seems to come from a single donor (H1) in which Xenium spatial transcriptomics suggests that virtually all cells in the tissue are plasma cells. This is surprising, as there are no B cells nor germinal center B cells identified in this tissue that could give rise to these plasma cells (there is no or little identification of T cells or thyroid epithelial cells either). At the very least, for this to be believable, they should show staining (IF or IHC) with key markers (eg. CD3, CD19, CD138, Blimp-1) to verify whether most of this tissue indeed is made up of plasma cells.

We agree that B cells could in theory contribute to breach in tolerance through additional mechanisms to antibody secretion e.g. through antigen presentation. However, the authors have not provided evidence of this either.

We appreciate the effort the authors have made to include better controls and inclusion of a large variety of additional tissue samples. Nevertheless, a more transparent approach in comparing mutational burden in 2 genes would be to compare tissues sampled the same way. I.e. Tissue sampled from AITD thyroid by microdissection and specified to contain >25% lymphocytes should ideally be compared to control tissue also captured by microdissection and with a >25% lymphoid content (i.e. ensuring the samples are taken from white pulp rather than red pulp in the spleen, and ideally germinal centers (GC) in tonsil (since human spleen barely contains reactive lymphocytes or GCs). The number of donors sequenced should also be made clear in the text: i.e. 3 donors for colon, 1 for Peyer's patches in ileum and 1 for tonsil were sequenced. At present, the text mentions n=5 (line 276).

(Remarks on code availability)

Referee #2

(Remarks to the Author)

I co-reviewed this manuscript with one of the reviewers who provided the listed reports.

(Remarks on code availability)

Referee #3

(Remarks to the Author)

The authors have done an outstanding job addressing the vast majority of the reviewer concerns. The addition of spatial transcriptomics and additional single-cell analyses substantially strengthens the manuscript.

The one response I had some remaining concerns with relates to the κ/λ staining in Fig. 1a. The explanation provided remains somewhat indirect, as it does not fully clarify whether the κ/λ signal primarily reflects plasma cells versus extracellular Ig deposition, nor why CD20⁺ follicular regions appear devoid of κ/λ staining. This does not detract from the central conclusions of the manuscript; however, a brief clarification in the text or figure legend would improve interpretability for readers. Additional detail in the Methods section describing the staining protocol and antibodies used would also be helpful, as this information is currently lacking.

Overall, this is a highly compelling and important study.

(Remarks on code availability)

Referee #4

(Remarks to the Author)

I co-reviewed this manuscript with one of the reviewers who provided the listed reports.

(Remarks on code availability)

Referee #5

(Remarks to the Author)

In this revised manuscript, the authors have addressed all our concerns, as well as the concerns of the other reviewers. I particularly appreciate the authors' efforts in including extensive non-autoimmune B cell negative controls. I congratulate the authors on their important study.

I have only a few minor issues remaining:

1. Extended Data Figure 4 and Figure 3d should have a color legend. The significance ****/***/**/* annotations are too small to be visible on their own, and the duplex coverage blue squares need a scale bar as well.
2. Two typos: (i) line 108: "This was was used"; (ii) line 577: "highlighted blue indicate" -> "highlighted in blue"

(Remarks on code availability)

The R notebook provides is very clear and well documented.

One minor comment: the authors mention they provide the code for dNdSshm, but we couldn't find it. As we understand, the code provided refers to a separate dnds_shm function that is loaded from another file, but that file was not provided.

Referee #6

(Remarks to the Author)

I co-reviewed this manuscript with one of the reviewers who provided the listed reports.

(Remarks on code availability)

Dear Inigo

Your manuscript, "Polyclonal selection of immune checkpoint mutations in thyroid autoimmunity", has now been seen by 6 referees, whose comments are attached below. While they find your work of potential interest, as do we, they have raised important concerns that in our view need to be addressed before we can consider publication in Nature.

You will see that there is great interest in this study among the reviewers, however, they have some concerns about the strength of the conclusions that can be made at this stage. As a key issue, we would like to highlight the appropriateness of the controls (mentioned by two reviewers), which we feel would need to be fully alleviated in a revised manuscript.

Should further experiments and/or analyses allow you to address these and all other criticisms, we would be happy to consider a revised manuscript (unless something similar has been accepted at Nature or appeared elsewhere in the meantime). We hope to receive your revised paper within four to six months. If you cannot complete the required revisions within this time frame, please let us know when you would anticipate being able to submit a revised manuscript.

As an additional request, I would also kindly ask that you include some general comments on the sensitivity of variant calling (e.g. at the top of your point-by-point response). This question was raised in confidential comments by one reviewer, and it would be good to assure them that the approach has sufficient sensitivity.

Any revised manuscript should conform to our format instructions and publication policies (see [here](https://www.nature.com) [[nature.com](https://www.nature.com)]). We also strongly suggest that your revised manuscript has tracked changes, which is increasingly requested by referees to aid in their re-review.

(...)

In the meantime, we hope that you will find our referees' comments helpful. Please do not hesitate to contact me if there is anything you would like to discuss.

Yours sincerely

Michelle

Dr Michelle Trenkmann
Senior Editor
Nature

orcid.org/0000-0002-3332-6344
www.nature.com

Referee expertise:

Referee #1: autoimmunity, genetics

Referee #3: autoimmunity, B cell development

Referee #5: somatic evolution

General comment on the sensitivity of NanoSeq:

In their confidential comments to the editor, one of the reviewers asked for additional information on the sensitivity of NanoSeq. The editor has requested that we provide this information at the start of our response to reviewers.

NanoSeq is a highly accurate duplex sequencing method with error rates $\sim 1e-9$ errors per bp in single molecules of DNA. Since this error rate is ~ 100 times lower than the mutation load of adult cells, NanoSeq enables the accurate detection of somatic mutations in single molecules of DNA. This means that mutations can be detected in a sample even when present in a single molecule of DNA in the sequencing library, which has enabled the findings of this study.

We have described the NanoSeq method in detail in two separate publications (Abascal et al, Nature, 2021, PMID:33911282; and Lawson et al, Nature, 2025, PMID:41062696). Standard sequencing methods require multiple independent molecules (or reads) to call a mutation and can only detect mutations in a sample if they typically affect $>1-5\%$ of the cells in the sample. Thus sequencing studies using standard sequencing methods are typically blind to somatic mutations affecting $<1\%$ of the cells of a sample. Accurate single-molecule DNA sequencing (with NanoSeq or other duplex sequencing methods) circumvents this limitation.

In practice, the sensitivity of NanoSeq to rare mutations is not limited by error rates and is only limited by the depth of sequencing achieved (i.e. by the number of molecules sampled and sequenced from every locus). For example, targeted NanoSeq of *TNFRSF14* at 1,000 duplex depth (dx) means that 1,000 individual copies of *TNFRSF14* were sampled from a cell population and read with very high sensitivity and accuracy. The *probability* of detecting a given mutation is then proportional to its variant allele frequency (VAF) in the population and the duplex depth achieved, but, critically, a single supportive molecule is sufficient to call a mutation unlike for standard sequencing technologies. Importantly, since the sampling is unbiased and detection of mutations is done on single molecules of DNA, NanoSeq enables unbiased discovery of driver mutations and unbiased estimation of mutant cell fractions independently of the sizes of a clone. For example, imagine two cell populations, one with a single large clone with a mutation at 10% VAF, and another sample with 10,000 microscopic clones with different mutations in the gene each at $VAF=1e-5$. In the first population, NanoSeq at 1,000dx will yield $\sim 100/1000$ mutant reads. In the second population, each individual clone has a 1% ($1e-5 * 1,000dx$) detection probability (due to sampling), but on aggregate 10% of copies sequenced from the gene will carry a mutation. In both cases, NanoSeq provides the correct estimated aggregate VAF of 10%, independently of the clonality of the sample. In contrast, standard bulk sequencing will correctly detect the 10% clone in the first sample but will not be able to call any mutation in the second sample. This makes NanoSeq (and other single-molecule mutation detection methods) a powerful tool for driver discovery in highly polyclonal samples that are below the detection sensitivity of standard sequencing methods. For more details, we kindly direct the reviewer to the original NanoSeq publications (Abascal et al, Nature, 2021, PMID:33911282; and Lawson et al, Nature, 2025, PMID:41062696).

Referees' comments:

Referee #1 (Remarks to the Author):

Nicola, Lawson, Martincorena et al follow up on their recent studies in which they mapped thousands of somatic mutations in oral, blood, and oesophageal samples, revealing clones with extensive selection patterns and risk factors. Here they use the same ultra-accurate sequencing method (NanoSeq) and additional sequencing methods to sequence B cells from thyroid of patients with autoimmune thyroid disease and visible germinal centers in tissue biopsies, to test the hypothesis that “somatic mutations in immune regulatory genes may enable self-reactive lymphocytes to bypass tolerance checkpoints”. They show B cell clones sharing LoF mutations in *TNFRSF14* and *CD74* (PD-L1), some with biallelic and/or several mutations in the same gene. Their conclusion is that “their studies support the hypothesis that somatic mutations in autoimmune lymphocytes allow them to escape tolerance constrains” and “provide new insights into the molecular basis of autoimmune disease.

These are interesting and striking observations that also build on earlier foundational work by David Schatz and others, who showed over two decades ago that BCL6-positive germinal center B cells can acquire mutations in non-BCR genes due to off-target activity of AID during somatic hypermutation. The present study advances this concept by showing that specific mutations may indeed confer a selective advantage, likely promoting their recycling to undergo multiple rounds of somatic hypermutation within germinal centers. There are two major concerns: The first is that the study suffers from a lack of proper controls – most notably tissues (not blood) from healthy controls, including tissues with and without chronic germinal center reactions. The second is that the framing of the conclusions implies a causal link between somatic mutations and B cell-driven autoimmunity, but this has not been established. Whilst the study describes selected somatic mutations in a tissue that is target of autoimmune disease, it does not provide proof that these mutations are causally related with disease.

Major concerns:

General:

- The data are consistent with a model in which somatic mutations contribute to germinal center B cell persistence or fitness, but there is no direct evidence provided that the observed mutations give rise to autoantibody-secreting plasma cells and thus to autoimmunity. It would be important to show i) the positively selected mutations are found in plasma cells and ii) the mutated plasma cells produce autoantibodies.

Authors' response:

We thank the reviewer for their positive summary of our study and for raising two interesting and important questions: (1) additional evidence of the mutations occurring in B cells and being involved in the pathogenesis of the disease, and (2) additional control data. We deal with the first point here and with the second one in the next question.

Here we address the first question in 3 parts:

1. Original evidence that the mutations occur in germinal centre B cells and plasma cells.
2. Additional single-cell DNA sequencing data for the revision.
3. Pathogenic role of mutant B cells.

1. Original evidence that the mutations occur in germinal centre B cells and plasma cells

Several lines of evidence support that the *TNFRSF14* and *CD274* mutations occur in B cells and plasma cells. *TNFRSF14* and *CD274*, together with *TET2* and *TNFAIP3* are the commonest driver genes of MALT lymphomas of the thyroid (Wu et al, 2021, PMID:34021249), which develop from post-germinal center B cells, particularly in the background of Hashimoto thyroiditis. This already strongly suggests that these mutations likely occur in antigen-experienced B cells in our samples, but we demonstrated this in two complementary ways. First, for donor H3, which has clear germinal centres dominated by

single mutant clones, the combination of Xenium spatial transcriptomics and laser microdissection maps several of these driver mutations to germinal centre B cells in this donor (additional quantitative data is provided in another answer to this reviewer). Second, in single-cell DNA sequencing data for 112 cells (66 B cells, 38 T cells and 8 other cells) from donor H1, *TNFRSF14* and *CD274* mutations were observed exclusively in antigen-experienced B cells or plasma cells, all of which have undergone somatic hypermutation of their BCR and 90% (37 out of 41 *TNFRSF14*-mutant B cells) have undergone class switch recombination. Further, Xenium spatial transcriptomics on this donor confirms that most lymphoid cells in this donor (H1) are plasma cells, including in regions dominated by *TNFRSF14* mutant clones.

2. New data mapping *TNFRSF14* and *CD274* mutations to antigen-experienced B cells and plasma cells

The single-cell DNA sequencing data provides the most direct demonstration of the mutations occurring in antigen-experienced B cells and plasma cells. Single-cell DNA sequencing using PTA is a new and costly technology, so in the submission we only included data from donor H1, as this was the donor with the highest frequency of *TNFRSF14* and *CD274* mutations. However, to further address the reviewer comment, in the revision we have generated data from an additional 180 cells from 2 other donors (73 from donor H2 and 107 from donor H8), taking the total number of single cells with DNA sequencing data to 292 cells from 3 donors. These data strongly support the original conclusion, identifying 13 additional single cells with *TNFRSF14* and/or *CD274* mutations, all of which are confirmed to be B cells from analysis of their BCR sequences.

These additional data are now described in the main text and in a new Extended Data Figure (EDF8).

3. Likely pathogenic role of mutant B cells

B cells are often considered to play at least 3 different roles in the pathogenesis of AITD: (1) by contributing to the breakdown of tolerance in their interaction with Th cells, (2) by acting as APCs, stimulating T cells, and (3) by differentiating into autoantibody producing plasma cells.

Several indirect lines of evidence support that the mutations that we have observed contribute to the pathogenesis of AITD:

1. The discovery that AITD B cells frequently carry loss-of-function somatic mutations in immune checkpoints and other lymphoma driver genes, particularly in *TNFRSF14* and *CD274*, which are key genes involved in peripheral tolerance and the interaction with Th cells (Wu et al, 2021, PMID:34021249), strongly suggests that these mutations could contribute to the deregulation of Th cells and the breakdown of B-Th peripheral tolerance observed in AITD.
2. Additional sources of evidence for a likely causal role of these mutations in AITD include: the causal GWAS link between *TNFRSF14* and AITD (as well as other autoimmune diseases) (Saevarsdottir et al, 2020, PMID:32581359), HVEM mouse KO models (Wang et al, 2005, PMID:15696194), and the frequent emergence of AITD secondary to anti-PD-L1 inhibitors (Chen et al, 2022, PMID:35597121).
3. Self-reactivity and autoantibodies: the reviewer specifically asks whether there is evidence that the observed mutations occur in self-reactive antibody producing plasma cells. We would argue that B cells can be pathogenic in AITD not just through the generation of autoantibody producing plasma cells but also through their action as APCs, cytokine production and dysregulation of Th cells. However, we agree that demonstrating that at least some mutant clones are self-reactive (e.g. recognising TPO or TG), would strengthen the evidence of a pathogenic role. Suggestive evidence that this is the case comes from Armengol et al, 2001 (PMID:11549579), which we cite in the revision. This study demonstrated that many tertiary lymphoid structures (TLS) in biopsies from Hashimoto patients recognise TPO or TG. We have already shown that many TLSs are clonal and carry *TNFRSF14* mutations, so it is likely that a fraction of mutant clones will recognise TPO and TG.

To directly examine this question, in the revised manuscript we provide data from a new set of experiments. Using the single-cell DNA sequencing data generated for donor H1 (a donor dominated by plasma cells), we reconstructed heavy and light chain BCR sequences from 32 B cells. We then generated recombinant antibodies from these 32 sequences on a human IgG backbone using a commercial provider (Biointron), and each of these antibodies was tested for binding to TPO and TG using 12-point ELISA assays. Interestingly, at least 7 of the 32 recombinant antibodies were self-reactive against either TPO (n=6) or TG (n=1). 3 of the 7 self-reactive clones carried mutations in *TNFRSF14* and/or *CD274*, and 1 in *BRAF*. 6 of the 7 had detectable class switch recombination. Altogether, this confirms that a sizable fraction of mutant clones are self-reactive against TPO or TG in this donor. For full details please see the updated Methods, Fig 5 and EDF11.

It is important to remember that whereas TPO and TG are two classical autoantigens in AITD, many self-reactive clones could be recognising other autoantigens. To explore whether some clones without ELISA reactivity to TPO or TG could be recognising other self-antigens, we also ordered 8 of the BCR sequences in a rabbit IgG backbone. 5/8 of these antibodies were self-reactive in normal thyroid biopsies by IHC. Reassuringly, this included 3 antibodies reported as anti-TPO positive by ELISA and one anti-TG positive antibody. IHC also identified a further *TNFRSF14*-mutant clone with evidence of self-reactivity (Fig 5f), consistent with evidence of weak binding to TPO in the rabbit IgG ELISA (EDF11d). Since not all autoantigens may be present in the biopsies with detectable levels at the time of sampling or may not be appropriately preserved or retrieved by the IHC protocol, the true fraction of self-reactive clones is likely to be higher. Interestingly, 3 ELISA-positive anti-TPO clones did not carry a detectable driver mutation. This is not entirely surprising as in the context of an established polyclonal autoimmune response the localised loss of peripheral tolerance is expected to enable other clones to proliferate.

We thank the reviewer for motivating us to carry out this experiment, which we believe has strengthened our study.

Authors' action: We thank the reviewer for their positive comments and for their constructive suggestions, which have helped us strengthen our study.

In the revised manuscript we have made several changes to address these comments, including:

1. Inclusion of the new single-cell DNA sequencing data from 2 more donors (from just 1 in the original submission), which further strengthens the mapping of mutations to antigen-experienced B cells (EDF8).
 2. Inclusion of Xenium transcriptomic data from 7 donors to complement the cell type characterization of our tissue sections (discussed in a later answer; EDF1).
 3. Heavy and light chain BCR reconstruction from 32 B cells followed by recombinant antibody synthesis and anti-TPO and anti-TG ELISA binding tests and IHC, which demonstrates the self-reactivity of 8 independent clones (Fig. 5f, EDF11).
 4. Changes to the text to clarify the questions raised by the reviewer, including toning down the discussion and including a reference to Armengol et al, 2001.
- The choice of controls – PBMCs that thus lack germinal center B cells and tissue resident cells - also limits the strength of the comparisons. A more informative baseline might be germinal centers from healthy lymphoid tissues (e.g., tonsils, mesenteric lymph nodes, or Peyer's patches), where similar mutational patterns could plausibly emerge as a feature of chronic stimulation. This distinction is important given existing evidence that chronic germinal center activity, even in autoimmunity-prone settings such as lupus, does not necessarily result in pathogenic autoantibody production: in both murine and human studies, anti-nuclear antibodies often arise from extrafollicular responses. Moreover, as the authors themselves mention, mechanisms exist within germinal centers to

negatively select or redeem self-reactive clones, so they cannot leave germinal centres and differentiate into plasma cells.

Authors' response: We thank the reviewer for raising this question. We note that reviewer 3 raised a similar question and the editor has also requested that we provide additional data.

Providing new targeted NanoSeq data from several secondary lymphoid organs is a large task, but we have made a considerable effort to address this as comprehensively as possible.

To address this question, we have performed deep targeted NanoSeq on a range of additional samples:

1. Lymph nodes from 4 healthy donors sequenced to a cumulative 10,740 duplex depth (dx).
2. Spleen from 30 healthy donors sequenced to a cumulative 21,748 dx.
3. 183 laser microdissection cuts of lymphoid aggregates from a range of tissue samples from 5 healthy donors, including colon (50 cuts from 3 donors; 1,093 dx), Peyer's patches from ileum (46 cuts from 1 donor; 478 dx), and tonsil (87 cuts from 1 donor; 2,085 dx).
4. Tonsil samples obtained from the tonsillectomies of 5 donors with a history of chronic tonsillitis (cumulative 19,803 dx).
5. Thyroid samples from 5 donors with nodular goitres but without AITD (cumulative 7,024 dx).

These data confirm that mutations in *TNFRSF14* and *CD274* are very rare and not under selection across a range of tissues, including healthy secondary lymphoid organs, tertiary lymphoid structures, chronically inflamed tonsils and goitrous thyroids. The Q-values for *TNFRSF14* and *CD274* mutations are 1 and 1 in non-autoimmune goitrous thyroids, 0.14 and 1 in peripheral B-memory cells from healthy individuals, 1 and 1 in healthy spleens, 1 and 1 in healthy lymph nodes, 1 and 1 in chronically inflamed tonsils and 1 and 1 in 183 microdissected TLSs from healthy donors. Additionally, the mutant cell fraction estimates for *TNFRSF14* and *CD274* (aggregated across donors) are <0.2% for all control datasets (Fig. 3c). This is an important point. The level of convergent positive selection that we are reporting in *TNFRSF14* and *CD274* in AITD is very high and absent from normal samples.

This strongly complements the original control data from flow-sorted PBMCs from healthy donors and further demonstrates the remarkable and highly specific nature of the *TNFRSF14* and *CD274* mutations that we report in AITD. We thank the reviewer for raising this comment, which we think has strengthened our manuscript.

Authors' action: Given that the request of additional control data was made by two reviewers and by the editor, we have made a large effort to address this as comprehensively as we could with five additional control datasets. In total, the new control data includes samples from 44 donors and a cumulative duplex depth (across 725 genes) of 62,971 dx.

We have updated the manuscript to describe these additional control samples and have added a new main text figure panel (Fig. 3c, shown below) and an expanded Ext Data Figure (EDF4). These data strongly support the original conclusions of the study.

Figure. Mutant cell percentage estimates for *TNFRSF14* and *CD274* in AITD samples with >25% lymphocyte fraction and 6 control data sets. Upper and lower bound estimates are shown for each group (aggregated across donors weighted by duplex coverage) and midpoint estimates for each donor (points).

- The data showing that B cells as opposed to other cells abundant in their biopsies are the cells carrying the somatic mutations is weak. Only the last figure addresses this for one patient. The authors sequence biopsies that contain large B cell clusters likely to contain germinal centres. It would be important to add germinal center markers and sequence cells within them and compare them with areas that do not contain germinal centers in the same gland.

Authors' response: We thank the reviewer but we disagree that the evidence that the mutations occur in B cells is weak. As explained above, several lines of evidence demonstrate that the *TNFRSF14* and *CD274* mutations occur in B cells and plasma cells, and we have strengthened this in revision with new Xenium and PTA data:

1. Drivers of MALT lymphomas (suggestive prior evidence): *TNFRSF14* and *CD274*, together with *TET2* and *TNFAIP3* are the commonest driver genes of MALT lymphomas of the thyroid (Wu et al, 2021, PMID:34021249), which develop from post-germinal center B cells. This strongly suggests that these mutations likely occur in mature B cells in our samples, but we demonstrated this in the original submission in two complementary ways, described below.
2. Spatial transcriptomics (Xenium) + laser microdissection: for donors H3, which had clear germinal centres dominated by single mutant clones, the combination of Xenium spatial transcriptomics and laser microdissection convincingly maps several of these driver mutations to germinal centre B cells in this donor (please see Fig 4). In many of these microdissections the fraction of mutant cells is sufficiently high that these mutations must be present in B cells based on their cell type composition estimated by Xenium.

3. Whole-genome single-cell DNA sequencing (PTA) from donor H1: In the original submission, we included single-cell DNA sequencing data for 112 B cells from donor H1 (66 B cells, 38 T cells and 8 other cells). In this dataset, *TNFRSF14* and *CD274* mutations occurred exclusively in mature B cells, as evidenced by somatic recombination of their BCR, the presence of somatic hypermutation and class switch recombination. Xenium spatial transcriptomics on this donor confirms that most lymphoid cells in this donor (H1) are plasma cells, including in regions dominated by *TNFRSF14* mutant clones.
4. New single-cell DNA sequencing data from an additional 180 cells from 2 more donors: To directly address the reviewer's comment that we only provide single-cell data for one donor, in the revised manuscript we include single-cell data from an additional 180 cells for 2 donors, taking the total number of single cells with DNA sequencing data to 292 cells from 3 donors. These data strongly support the original conclusion, identifying 13 additional single cells with *TNFRSF14* and/or *CD274* mutations, all of which are confirmed to be B cells from analysis of their BCR sequences. Of note, the number of *TNFRSF14* and/or *CD274* mutant cells identified in these additional donors is consistent with the number predicted by the mutant cell fractions obtained from Targeted NanoSeq. Please see the new EDF8.

Authors' action: We have updated the manuscript to clarify the different lines of evidence that demonstrate that the *TNFRSF14* and *CD274* mutations occur in mature B cells and plasma cells, including the new PTA data for an additional 2 donors. These data are also described in EDF8. In the revision we have also extended the original Xenium data from 2 donors to 7 donors using a broader Xenium 5k panel, to enrich the cell type characterisation across donors, shown in new EDF1.

- The Figures are difficult to read and interpret and the figure legends do not provide sufficient information. This contributes to the longer-than-usual list of questions and issues as described below that require clarification.

Authors' response: We apologise for any confusion and are grateful for the suggestions below to make the paper more accessible outside of the somatic evolution field.

Figure by Figure

Figure 1

- This figure describes ALL cells in thyroid samples from AITD patients. The authors should acknowledge that a significant proportion of mutations may occur in non-B cells and non-lymphoid cells.

Authors' action: We apologise for the confusion. We have made this clearer in the revised text.

- A healthy thyroid tissue control would be important, particularly when the frequency of somatic mutations is so variable between 3 AITD patients. Most mutations are identified in H1 while H2 and H3 have much lower levels. This should be clearly acknowledged in the text.

Authors' response: Thank you for the suggestion. As described in our responses above, we have added 5 additional control datasets in the revised manuscript, including deep targeted NanoSeq of non-autoimmune goitrous thyroid biopsies from 5 donors. We think that the original text and figures already highlighted that the number of mutations detected varies significantly across donors (e.g. half of Fig 1 was devoted to donor level figures), but we have ensured that this is clear in the revised manuscript.

Although the dramatic number of mutations and the extremely high dN/dS ratios observed in donor H1 may make the number of mutations in other donors appear low, we would like to emphasise that the numbers are remarkably *high* across donors (and statistically highly significant by dN/dS tests). For example, we detect ~40 independent driver mutations in *TNFRSF14* in donors H2 and H3 each, ~60 *CD274* independent driver mutations in H2 and ~10 in H3. These are strikingly high numbers.

- Details are needed to understand prioritisation of the 4 genes mentioned. It is difficult to follow. How much above 1 does dN/dS values have to be to be considered meaningful? dN/dS values don't seem that high for TET2 and TNFAIP3 (Figure 1d). Are these true enrichments? What is the enrichment in dN/dS values in each patient? Again, healthy controls would be needed.

Authors' response: We apologise for the confusion. These 4 genes are simply the genes that reach exome-wide significance for positive selection. This is a standard approach for driver discovery in somatic evolution and in cancer genomics, so common in the somatic evolution field that, in retrospect, we probably failed to explain it sufficiently for other audiences. Briefly, the 4 genes above are those that show evidence of positive selection with a Q-value cutoff of 0.01 (equivalent to a 1% false discovery rate, which is a stringent cutoff). The signal of selection in the 4 genes is overwhelming, with Q-values of $<1e-15$ for *TNFRSF14* and *CD274*, $1.25e-06$ for *TET2*, and $9.06e-06$ for *TNFAIP3*. As expected, their signal becomes even stronger with the additional coverage and samples of the extension cohort, with Q-values of $1.64e-08$ and $1.95e-14$ for *TET2* and *TNFAIP3* respectively. These Q-values were provided in the submission manuscript within the Supplementary Code file, but we now provide them in the main text for clarity.

The reviewer also asks (understandably) whether the dN/dS ratios are sufficiently high in these genes. Again we apologise for any lack of clarity. The dN/dS values are, indeed, very high, but we acknowledge that this is unclear in the previous figures and we thank the reviewer for raising this. We have modified Fig. 1 to include 95% confidence intervals and a log scale to make the high degree of enrichment in all four driver genes more apparent. The dN/dS values for truncating (loss-of-function) mutations are ~ 20 for *TET2* and *TNFAIP3* in Fig 1, which are very high values (even if they appear low next to values ~ 1000 for *TNFRSF14*). A dN/dS value of 20 means that there are 20-times (2,000%) more truncating mutations in these genes than would be expected without selection, which are very large enrichments for the somatic evolution field.

Authors' action: We have now made this clearer in the manuscript, including by showing dN/dS ratios with confidence intervals in a log-scale in Fig. 1 to more clearly show the dN/dS ratios for *TET2* and *TNFAIP3* (shown below).

Figure. dN/dS ratios with 95% confidence intervals per mutation consequence category (horizontal line indicates neutral dN/dS=1) for the four genes found to be under significant positive selection ($q < 0.01$) in exome NanoSeq data across all cells in the pilot AITD cohort

- Figure 1E: Not clear whether the “upper and lower bounds” refer to statistical uncertainty with the method used or upper and lower bounds between the three patients. The three patients shown in Figure 1F (and extended figure 1) seem to have much higher (H1) and lower (H2 and H3) bounds than what is presented in Figure 1E e.g. For *TNFRSF14*. Same in Figure 2c and other places in the manuscript: what is “upper and lower” bounds?

Authors' response: We apologise for the confusion and thank the reviewer for raising this. These upper and lower bounds represent the estimated percentage of cells carrying a mutation in these genes

assuming that each cell carries just 1 mutant copy of the gene (higher bound for the mutant cell fractions) or two (lower bound for the mutant cell fraction). We have used these bounds and described them in detail in previous studies (in particular in Martincorena et al, Science 2018, PMID:30337457).

Authors' action: We have clarified this in the main text and the Methods section.

- Figure 1F: Is not referenced in the text.

Authors' response: Please note that Fig 1f was cited in the first paragraph of page 5.

- Text line 136/137: “Although each mutation is only present in a small proportion of cells, cumulatively across many clones the estimated fraction of mutant cells in these biopsies reaches remarkably high levels. For donor H1, at least 32.7% and 5.0% of all cells bear TNFRSF14 and CD274 mutations respectively (Methods). The other donors had lower, but still substantial, TNFRSF14 (H2 = 3.0%, H3 = 5.5%) and CD274 (H2 = 4.4%, H3 = 0.3%) mutant cell fractions”
 - For donor H3 the level of CD274 mutant cell fraction seems barely detectable. There seems to be a high level of variability between patients (also seen in later figure) that the authors should acknowledge. TET2 and TNFAIP3 also seem very low across all donors. Levels in healthy tissue would be needed.

Authors' response: We believe that we already acknowledged the inter-donor variability prominently, both in the main text (as shown in the quoted sentence above, where we provide the actual mutant cell fractions per donor) as well as in the main text figures. Half of the space in Fig 1 is devoted to donor-level figures, and much of Fig 2 focuses on this variation, including histology images per donor (Fig 2a), cell type composition by methylation sequencing (Fig 2b), mutant cell fractions and exact numbers of mutations per gene in each donor (Fig 2d). The variation in the frequency of *TNFRSF14*, *CD274* and other drivers is important and interesting, but it is also not surprising or concerning given that the fraction of B and plasma cells vary greatly across donors, and given the variability in disease severity and histopathological features across donors (such as TLSs and follicular destruction).

Authors' action: To further emphasise the variation across donors, as requested by the reviewer, we have changed several sentences in the main text (e.g. the estimated fraction of mutant cells in these biopsies reaches remarkably high levels, *albeit with considerable variation across patients*). In the revision we have also included new Xenium data from 7 donors, to better characterise the heterogeneity in histopathological features across the cohort. As requested by the reviewer and as described above, we now also include additional control datasets showing that these mutations are very rare and not under selection in other settings.

- Can lower levels of HVEM and PDL1 protein be detected on affected areas? Also relevant to Figure 4/ Extended Figure 6, show which areas are selected for imaging and across multiple patient samples or areas.

Authors' response: Thank you. Yes, that is indeed the case. We already showed this in the submitted manuscript using IHC and IF on clonal areas. Fig 4a shows an IHC example from donor G5 of a region dominated by a HVEM-mutant clone with reduced HVEM protein levels mapping to the mutant clone boundaries identified by laser microdissection. We included another example, from a different donor (H3), in EDF6 (now incorporated in new EDF5) showing reduced HVEM levels by IF in heterozygous and homozygous mutant germinal centres compared to a WT one.

Authors' action: We have extended EDF6 to include IHC from the same regions shown by IF for donor H3 to provide additional confidence on the IF results.

Figure 2

- Figure 2b: Targeted methylation sequencing is not mentioned in the text. Hard to follow what the authors did when it only appears in figure legends, or much later than where the results are described.

Authors' response: Thank you. The targeted methylation data was described in the original manuscript at the bottom of page 3 in reference to Fig 1, and at the top of page 6 in reference to Fig 2. The first reference in the main text was the following (page 3):

In addition to histopathological evaluation, targeted methylation sequencing was used to quantify cell type composition (Methods), which confirmed high levels of B cell (median = 17.2%, range = 12.4% - 41.7%) and T cell infiltration (median = 49.8%, range = 27.7% - 54.8%), with a low proportion of follicular epithelial cells (median = 4.8%, range = 1.7% - 8.5%) (Fig. 1b). This was in stark contrast to healthy thyroid samples that were dominated by follicular epithelium (median = 65.5%, range = 56.6% - 80.1%).

Authors' action: To better explain our methodology, in the revised manuscript we have expanded the **Methods** section describing the targeted methylation assay. We have also added a reference to the original work underpinning this method (Loyfer et al., 2023, PMID:36599988). The current version now reads *“In addition to histopathological evaluation, targeted methylation sequencing can be used to estimate the cell type composition of a sample (Loyfer et al, 2023). This was used to quantify cell type composition (Methods)...”*. We hope the current version is clearer and we thank the reviewer for this constructive comment.

- Figure 2c:
 - Figure legend states that grey denotes genes that reach significance only due to SHM – but earlier the authors describe that they developed a method that takes SHM into account and so this should not happen? How do the authors distinguish genes that are due to SHM from the other genes in black?

Authors' response: We apologise for the lack of clarity. In response to this question, in the revised **Methods** section we include a more detailed description of dNdSshm. Initially, this method was described in detail in a manuscript in preparation that we are hoping to be able to cite in the final version of this manuscript. But we now provide a detailed description in the current manuscript.

Briefly, a small minority of exons in B cells are affected by off-target somatic hypermutation. These exons are easy to identify leveraging a deep targeted NanoSeq study of sorted B-mem cells from PBMCs from healthy donors. Exons affected by SHM present a characteristic mutation spectrum (consistent with on-target SHM in the IGH loci), as well as a highly elevated background mutation rate (clearly distinct from selection as manifested by a large excess of synonymous mutations and intronic mutations in the exon flanks). As expected, the regions we identified as being affected by off-target SHM only showed an increase in the synonymous and flanking intronic mutation burden in our sorted B-memory cell data and did not have elevated burdens in our sorted CD4-memory or CD8-memory populations. In dNdSshm, we simply exclude exons affected by SHM from the dNdScv driver discovery analysis, obtaining reliable selection results free from the confounding effect of off-target SHM (these genes are shown in black in Fig 2c). However, selection can theoretically also act on exons affected by SHM, which should manifest as a considerable excess of non-synonymous mutations over the already-increased number of synonymous mutations in these exons. To account for this possibility, in dNdSshm we also run a local dN/dS model (called dNdSloc, see Martincorena et al, Cell 2017, PMID:29056346), which performs an additional selection test while removing the effect of the locally elevated mutation rate and of the distinctive mutation spectrum of SHM. Genes significant for this SHM-aware test of selection are shown in grey in Fig 2. While these genes show suggestive signs of selection, we recommend more caution while interpreting them as the background mutation model used by dNdSloc may not fully account for all of the features of off-target SHM, as we explain in the manuscript. Of note, most of the genes found under positive selection in AITD are unaffected by off-target SHM, including *TNFRSF14* and *CD274*.

Authors' action: We thank the reviewer for highlighting that Fig 2c was confusing. We now provide a detailed explanation of how dNdSshm handles somatic hypermutation in a revised **Methods** section. Thank you.

- Figure 2d:
 - The text says (line 207/208) “Clones with mutations in immune checkpoint genes were observed across biopsies from all donors with sufficiently high lymphocyte abundance and depth of sequencing (Fig. 2d).”
 - The plot shows combined pilot and extension datasets. The extension dataset looks less convincing - it does not confirm the results found in the pilot: The only donor that has detectable levels of non-synonymous mutations is HB – the rest have 0-9 non-synonymous mutations in the two top genes.

Authors' response: We apologise for the confusion and welcome the opportunity to improve the relevant text in the revised manuscript.

Although the reviewer is correct that mutations in *TNFRSF14* and *CD274* are less common (in absolute terms) in the bulk sequencing data of the extension cohort, this is unsurprising and does not undermine the results of the study for the reasons provided below.

- First, it is important to emphasise that *TNFRSF14* and *CD274* are very strongly validated by the extension cohort samples (i.e. they are under highly significant positive selection in the extension cohort samples alone, with Q-values $< 1e-15$ in both cases, see Supplementary Code file for all details), with dN/dS ratios for truncating mutations of 53 and 27, respectively. For clarity, this means that loss of function mutations in these two genes are 53-fold and 27-fold more frequent than expected under neutrality. Q-values remain highly significant in the extension cohort even after excluding donor H8, mentioned by the reviewer: Q-value = $4.3e-11$ for *TNFRSF14* and $3.3e-6$ for *CD274*. These Q-values and dN/dS ratios are *very strong* signals of selection. This is also apparent when compared to similar depths of sequencing in the 6 control datasets, all of which show non-significant selection for both genes (Q-value > 0.1 across each of the control datasets for *TNFRSF14* and *CD274*).
- Second, it is important to remember that the extension cohort was designed not as a validation of the pilot cohort, but as a collection of biopsies with much less advanced AITD features, to test the extent to which our results extended to earlier or less advanced disease. As shown in Fig 2a and b, the samples in the extension cohort have much lower levels of thyroid destruction and, crucially, much lower levels of B cell infiltration, which naturally reduces the expected number of driver mutations from B cells in the extension cohort detected with a given duplex depth.
- Third, as can be seen in Fig 2d, the extension cohort was sequenced at a lower depth than the pilot samples, which again linearly reduces the expected number of driver mutations in immune checkpoint genes. Even then, however, the results of the extension cohort show that most samples in the extension have clones with driver mutations in *TNFRSF14* (7 of the 11 samples) or *CD274* (8 of the 11 samples). In fact, only one sample (G4) does not show any mutation in *TNFRSF14*, *CD274* or *TNFAIP3*, and this is precisely the sample with the lowest estimated fraction of lymphocytes ($\sim 2.5\%$), with B cells being undetectable (estimated fraction of 0 using EMseq, see Supplementary Table 3).
- To provide a more quantitative estimate of the fraction of B cells or plasma cells with mutations in these genes, we can divide the mutant cell fractions estimated from the sequencing data by the estimated fraction of B and plasma cells in each sample, using both the EMseq targeted methylation data and the Xenium spatial transcriptomics data. Whereas this is only an approximate analysis (which is the reason why we did not include it originally) this reveals that the estimated fraction of B or plasma cells with mutations in *TNFRSF14* or *CD274* is often substantial across samples other than H1 with high ($>25\%$) lymphocyte fractions (H2: 23% *TNFRSF14* mutant, 31% *CD274* mutant; H3: 11% *TNFRSF14* mutant, 0.3% *CD274* mutant; G1: 6% *TNFRSF14* mutant, 14% *CD274* mutant; G3: 0.4% *TNFRSF14* mutant, 0.7% *CD274* mutant; G5: 8.2% *TNFRSF14* mutant, 0.4% *CD274* mutant; H8: 8% *TNFRSF14* mutant; 25%

CD274 mutant). Of note, this estimate of *TNFRSF14* mutant B cells for G5 is consistent with the mutant cell fraction in microdissected lymphocytes (10.6%, see EDF1c and Supplementary Code) and is substantially higher than the mutant cell fractions observed across the control cohorts (Fig. 3c).

Authors' action: We hope that this additional context is helpful. In the revised manuscript, we have changed the sentence highlighted by the reviewer to more clearly acknowledge the presence of samples without these mutations and to provide the additional context above. We have also included a new sentence providing the approximate estimate of mutant B cells above, while acknowledging the caveats around B-cell fraction estimation from targeted methylation.

- There is a high degree of patient variability that does not seem to be explained by the degree of lymphocyte infiltration. Can the authors please clearly state this and discuss? Although patients are sorted by lymphocyte infiltration, it does not seem like there is clear correlation between lymphocyte infiltration and mutation counts. E.g., patient G3 has higher lymphocyte infiltration but low mutation counts. Patient H9 has lower lymphocyte infiltration but higher mut counts. Patient H5 sample did undergo laser capture to capture more lymphocytes and should thus have higher lymphocyte infiltration and a higher chance of finding mutations, yet none were observed for *TNFRSF14*. For Patient G3 (also laser capture) found 1 in *TNFRSF14*.

Authors' response: Thank you. It is important to clarify that there are technical and biological reasons for the observed interpatient variability.

- Technical sources: to explain the interindividual variation in the number of driver mutations detected, two variables are more important than the total lymphocyte fraction per donor: (1) the B cell fraction, and (2) the duplex depth per patient. For example, imagine that 10% of B cells carry a heterozygous driver mutation in two donors. Imagine that donor A was sequenced to 10,000dx and had a 20% B cell fraction, whereas donor B was sequenced to 2,000dx and had a 5% B cell fraction. If we assume that clones are very small, we then expect donor A to have $10000 * 0.10 * 0.20 / 2 = 100$ detected driver mutations and donor B to have $2000 * 0.10 * 0.05 / 2 = 5$ detected driver mutations. Thus, it is important to consider both the B cell fraction and the duplex depth (both of these variables are now included in the revised heatmap).
- Biological sources of variation: beyond the technical reasons above, interpatient variability should not be surprising given the histopathological differences observed across donors. In fact, we would argue that the convergence across donors is quite remarkable. Considering the estimated B cell fraction and duplex depth, we estimate that most donors (except H1) have *TNFRSF14* mutations in ~5-10% of B cells.

Authors' action: We have updated the heatmap in Fig 2d to show the B cell fraction as well as the total lymphocyte fraction, although we note that these are estimates based on targeted methylation data and do not have resolution of B cell subtypes, for example. In the revised text, we have also clarified the importance of duplex depth and clone sizes for the interindividual variation in the absolute number of driver mutations detected. Both factors account for a majority of the variance in the interindividual variation in driver mutations, but additional biological variation is also relevant and expected between donors given their different stages of disease progression, histopathology and genetics.

- Reading of the figure would be helped by clearly stating in the figure legend that mutation cell fraction is in colour and that the number refers to non-synonymous mutations.

Authors' action: Thank you. We have incorporated this suggestion.

- Text (line 213/214) “Importantly, *TNFRSF14*, *CD274* and *TNFAIP3* were also significant in the extension cohort alone, validating them in an independent cohort” – However no data for this is shown. How are these genes also significant? By dN/dS ratio?

Authors' response: Thank you. *TNFRSF14*, *CD274* are very strongly validated by the extension cohort samples (i.e. they are under highly significant positive selection in the extension cohort samples alone, with Q-values < 1e-15 in both cases, see Supplementary Code file for all details), with dN/dS ratios for truncating mutations of 53-fold and 27-fold, respectively. Q-values remain highly significant in the extension cohort even after excluding donor H8, mentioned by the reviewer: Q-value = 4.3e-11 for *TNFRSF14* and 3.3e-6 for *CD274*. These Q-values and dN/dS ratios are very strong signals of selection. The Q-value for *TNFAIP3* in the extension cohort alone is 6.57e-04 and its dN/dS enrichment of loss of function mutations is 8.5-fold.

Authors' action: We now provide these Q-values in the text rather than solely in the Supplementary Code as we did originally.

- Extended data figure 1b: How is the % of lymphocytes determined? By nanoseq alone? Or by targeted methylation seq. which is not described? Please indicate the colour-scale bar for the cell fraction carrying a mutation and mention clearly in the figure legend what the numbers and the colours mean.

Authors' action: Thank you for these suggestions. We can confirm that the estimated lymphocyte percentages shown in the manuscript were derived from targeted methylation data, using the same approach that we described in detail in a separate paper (Lawson et al, Nature 2025, PMID:41062696). We agree with the reviewer that this needs to be better described in the Methods section and we have now added a section devoted to it.

Figure 3

- It is interesting that there seems to be selection for truncating mutations (mentioned also earlier in the text) and in cysteine-cysteine bonds, but it is hard to understand the significance of this finding as there are no controls and despite great variability between patients, the figure shows all patients pooled together. Could the authors analyse whether there would be an enrichment in these mutations on a patient-by-patient basis?

Authors' response: It is important to clarify that cohort level analyses of selection are the standard tests in studies of somatic evolution (e.g. PMID:23770567, PMID:30337457, PMID:41372419), as somatic data is almost always too sparse at an individual level to provide statistically meaningful information. For example, in cancer genomics, evidence of selection on driver mutations can only be found by aggregating mutations from many patients (even if selection is active in every donor), as any given gene only carries 0-2 mutations per patient in the vast majority of cases (e.g. PMID:23770567, PMID:29056346).

The reviewer also says that “*it is hard to understand the significance of this finding as there are no controls*”. Here, it is important to clarify that positive selection is tested against a null model of neutrality, using synonymous mutations as internal controls. Specifically, dN/dS ratios test whether the density of certain mutations (e.g. truncating or cysteine-cysteine bonds) is higher than expected by chance given the density of neutral (synonymous) mutations in the gene. These tests are very well established and used widely in the somatic mutation field. Fig 3b shows the dN/dS ratios for these classes of mutations, which show the fold-enrichment over neutral mutations. They show that missense mutations in cysteine-cysteine pairs in *TNFRSF14* are enriched ~10-35 fold over neutral expectation - Q-values < 1e-5 for the three CRDs-, which is a very strong signal of selection. A separate control dataset is thus not necessary for a selection analysis (it is also never needed for driver discovery in cancer genomes). However, we hope that the addition of 5 new control datasets in the revision, together with the original control B-mem data, further helps reassure the reviewer that the signal of positive selection is very strong in AITD and absent in the control datasets.

- Are the findings in Figure 3b and Ext. data Fig 3h significant? Not mentioned in legends.

Authors' response: Thank you for this suggestion. The error bars show 95% confidence intervals, and so the fact that they do not overlap with the neutral value of $dN/dS=1$ indicates that they are statistically significant with $P\text{-value}<0.05$. We have added significance indicators to the figures and updated the figure legends to clarify this.

Authors' action: We have updated the legends of these two figures to highlight the red horizontal dashed line represents a neutral value of $dN/dS=1$. Additionally, the significance of each test is now indicated within the figure.

- Text (line 240/241): “cause C-terminal truncation, resulting in reduced desensitisation and internalisation of CCR6 (mediated by β -arrestin) and upregulation of CCL20-mediated NF- κ B signalling”. Despite published information on these proteins the exact effect of the mutations is speculative. The authors do not show experimentally the effect of the mutations and so conclusions on the effects must be toned down.

Authors' response: Thank you. We note that several studies not cited in the manuscript (due to constraints on the number of references) have provided considerable experimental evidence on the activating role of C-terminal truncation of CCR6. For example, an in vitro study demonstrated that the W355X nonsense mutation, which we observe in three donors, leads to increased resistance to apoptosis particularly upon ligand stimulation (PMID:35142152). A further 14 nonsense mutations and frameshift indels are observed across the cohort, all of which would be expected to similarly truncate the CCR6 C-terminal phosphorylation motif. In fact, no truncating mutations were identified distal to the phosphorylation motif. However, we agree with the reviewer's request and we have toned down the sentence as requested.

Authors' action: We have replaced the text referring to *CCR6* truncating mutations “resulting in reduced desensitisation” with “expected to result in reduced desensitisation”. We have also added a second citation here that demonstrates the functional consequences of one of truncating mutations we observe.

- Extended data figure 3: While it is interesting that for some genes the hotspot seems to be the same as in leukaemia, it is again difficult to judge the significance due to lack of healthy controls.

Authors' response: As in one of the answers above, it is important to say that detection of positive selection does not need an external control. When we report a hotspot under selection, we are using a well established dN/dS -based method for statistical analysis of hotspot enrichment that uses synonymous mutations as internal controls and corrects for trinucleotide mutability (*sitednds* and *codondnds* functions in the *dNdScv* package; Lawson et al, PMID:41062696). Thus, statistical detection of positive selection does not need external controls. However, demonstrating that the positive selection detected in AITD is specific to the disease does require a control dataset. As described above, we now provide 6 different deeply sequenced control datasets, all of which show lack of significant positive selection in *TNFRSF14* and *CD274* (Q-value > 0.1).

Authors' actions: As described in our responses above, we have added 5 additional control datasets in the revised manuscript, including deep targeted NanoSeq of non-autoimmune thyroid biopsies, chronically inflamed tonsils, spleens and lymph nodes from healthy donors, and microdissected TLSs.

- IRF1 and CD45 are mentioned, but it is unclear how these would lead to autoimmunity if LOF?

Authors' response: Due to space constraints we did not discuss all genes in the main text, although we discussed some of them in greater detail in a Supplementary Note. It is important to clarify, however, that not all the genes found under selection in AITD need to be causally implicated in the pathogenesis of AITD, and in fact we already show how some of the less frequent driver genes are also under selection in lymphocytes during healthy ageing. Of note, *CD45/PTPRC* is under stronger positive selection in both CD4 and CD8 memory T cells from non-autoimmune donors during ageing than in our bulk AITD

data (see EDF4), and so its signal in AITD is likely to come from T cells and may be unrelated to AITD pathogenesis. Given the overwhelming signal of selection in *TNFRSF14* and *CD274*, their specificity to AITD, and their known roles in B-T tolerance, we have focused most of the discussion on the possible causal role of somatic driver mutations in AITD on these two genes.

- Text (Line 274-279) and 382/384: The authors state themselves that there is selection for several of the “driver” genes during normal ageing. The patients used here are also not that young, so how can the authors conclude that this is specific to AITD without controls? This also weakens the argument that the somatic mutations indeed are “driver” mutations

Authors’ response: This question was already discussed in the submitted manuscript but we are happy to provide an additional clarification.

To clarify, when we refer to a mutation as a “driver mutation” we mean that the mutation is under positive selection, i.e. driving clonal expansions, as demonstrated by dN/dS analyses. In the last 10 years, we and others have shown that somatic driver mutations are common and under genuine positive selection in normal tissues (e.g. PMID:25999502, PMID:30337457, PMID:32350471). Sensitively studying this phenomenon in ageing lymphocytes had not been possible beyond clonal haematopoiesis (where selection occurs in HSCs), as somatic mutations in normal lymphocyte clones from blood are below the detection sensitivity of standard bulk sequencing methods. To address this, in a separate study (in preparation), we have applied whole-exome NanoSeq and deep targeted NanoSeq to sorted populations of lymphocytes from PBMCs. This has revealed evidence of positive selection on a collection of immune regulatory genes.

These control data are valuable to highlight how dramatically different the landscape of selection is in AITD compared to ageing lymphocytes. As shown in the paper (Fig 3c), LoF mutations in *TNFRSF14* and *CD274*, are extremely rare and show no signs of selection in circulating B memory cells, in stark contrast to our data in AITD. The addition of 5 additional control datasets in the revised manuscript further demonstrates how exceptional the frequency and selection of *TNFRSF14* and *CD274* mutations are in AITD.

However, as explained in the manuscript, a few of the other driver genes found under selection in AITD appear under similar positive selection (by dN/dS and by mutant cell fraction) in AITD lymphocytes and in controls. This means that mutation of these genes does not confer an additional selective advantage to lymphocytes in the context of AITD, beyond their advantage in normal ageing. This does not strictly rule out a possible role in disease for these genes, but their lack of enrichment in disease suggests that they could theoretically be bystanders in the context of AITD, expanded during ageing but not involved in the pathogenesis of AITD. For this reason, in the original manuscript, we discussed these genes cautiously and made no claims regarding their possible involvement in disease.

Authors’ action: We have made some changes to the text to further clarify the points above. We have also included the 5 new control datasets in the revised version, which are discussed in the main text and summarised in Fig 3c (reproduced above) and EDF4.

- Text (Line 286/287): “This is in stark contrast to the high number of driver mutations in *TNFRSF14* and *CD274* in the AITD samples, as both genes are very rarely mutated in non-autoimmune lymphocytes”. This is an overstatement with only 2-3 patients having a high level of these mutations.

Authors’ response: We hope that our answers above clarify this point. The evidence of positive selection in *TNFRSF14* and *CD274* is overwhelming in both the pilot and extension cohorts, as we have explained in our previous answers. Although the numbers of mutations observed in some donors may seem low when compared to the dramatic numbers in the most mutated donors, these mutations are greatly enriched over neutral expectation even in the group of less mutated donors. This is demonstrated by very high dN/dS ratios and highly significant Q-values, even when excluding the most highly mutated

donors. We do not know of another condition outside of cancer where convergent positive selection on clinically relevant genes is as strong as reported here.

- Figure 3c: Duplex coverage is also very different in the two data sets. i.e., Not a one-to-one comparison

Authors' response: Thank you. This is not a problem in itself, but the duplex depth and the B cell fraction are important factors to consider when interpreting the number of mutations observed per gene per donor, as explained above. Please note that the variation in duplex depth is already taken into account when we report mutant cell fractions (the fraction of cells in a sample carrying a mutation), as well as dN/dS ratios and Q-values.

It is also important to say that the 6 control datasets were sequenced to much higher depths than the pilot exome data that first revealed strong selection in *TNFRSF14* and *CD274*, precisely to maximise our sensitivity in the control datasets. This is summarised below and highlighted in the revised manuscript:

Pilot exome data: 3,125 dx. 102 non-synonymous mutations in *TNFRSF14* and 40 in *CD274*.

Pilot targeted data: 29,584 dx. 229 *TNFRSF14* and 125 *CD274*.

Control B-mem (healthy donors aged ≥ 50): 19,975 dx. 10 *TNFRSF14* and 6 *CD274*.

Normal lymph nodes: 10,740 dx. 1 *TNFRSF14* and 1 *CD274*.

Normal spleen: 21,748 dx. 0 *TNFRSF14* and 4 *CD274*.

Microdissected lymphoid aggregates: 3,656 dx. 0 *TNFRSF14* and 0 *CD274*.

Tonsillitis: 19,803 dx. 4 *TNFRSF14* and 6 *CD274*.

Nodular goitre thyroid: 7,024 dx. 1 *TNFRSF14* and 1 *CD274*.

Figure 4

- The authors need to tone down their conclusions. Conclusions often described as “revealed that” in definitive terms when results are only suggestive.

Authors' response: Thank you. In the revised manuscript we have toned down definitive statements when results are only suggestive.

- Figure 4a: It cannot be concluded that there two different clones. There is some overlap in the microdissections with the two types of mutations, although essential splice enriched at the top and frameshift enriched at the bottom. No definite conclusion can be reached either on whether the mutations occur in a lymphocyte; non-lymphoid tissue has not been sequenced for comparison.

Authors' response: Thank you. As we explained below in more detail, the available allele frequencies strongly imply that: (1) there are two separate clones occupying partly overlapping areas of this lymphoid aggregate, and (2) that the vast majority of cells in this region carry these mutations.

The first evidence is the variation of VAFs across biopsies. Fig 4a clearly shows that the VAFs of the two essential splice mutations co-vary across sections, and that they anticorrelate with the VAF of the frameshift insertion. This behaviour suggests that these are two separate clones. However, more definitive evidence comes from the actual VAF values. As it can be seen in the figure, the VAFs of the two essential splice site mutations reach values around 30% in several LCM sections, while the frameshift insertion has VAFs around 10% in those biopsies. This can only be explained by a single-clone model if the frameshift insertion occurs in a subclone of a parental clone with the two essential splice site mutations. However, the opposite pattern is seen at the bottom of the lymphoid aggregate, where the VAF of the insertion reaches values higher than 50% and the VAFs of the two essential splice site mutations drop below 10%. Mathematically, this is inconsistent with any single clone model and requires at least two separate clones (for more details on the use of VAFs for subclonal deconvolution see PMID:22608083 and PMID:28270531).

It is also important to note that copy number analysis of these microdissections confirms that the *TNFRSF14* locus is diploid in these cuts, with CN-LOH occurring in the clone with the frameshift insertion (see the original EDF5) but not in the cuts dominated by the two essential splice site mutations, as described in the main text. Together, this means that there are 4 separate loss-of-function hits in *TNFRSF14* in this region, which again would be highly unlikely for a single diploid clone.

Finally, VAFs can be used to infer MCFs (mutant cell fractions) by correcting for local ploidy (see PMID:22608083). Since the VAFs support the existence of two separate clones, the mutant cell fractions of the two clones can be summed in each section to estimate the fraction of cells in each microdissection having biallelic loss of *TNFRSF14*. Doing this on the microdissections in Fig 4a clearly reveals that >80-90% of all cells in most cuts of this lymphoid aggregate have mutations in *TNFRSF14*. This provides strongly suggestive evidence that the mutations occur in lymphocytes. More definitive evidence for this statement comes from the fact that co-occurring SHM mutations in the IGH locus of these cuts occur at similarly high VAFs, which using the pigeonhole principle (PMID:22608083) implies that these mutations occur in B cells in this lymphoid aggregate.

Authors' action: We hope that these additional explanations are helpful. As explained above, the pattern of VAF variation across microdissections is not compatible with any single clone nesting model, and the aggregate MCFs together with SHM mutations imply that they occur in B cells (through the pigeonhole principle). Please note that in the manuscript we refer to the single-cell DNA sequencing data as the more definitive evidence that these mutations occur in B cells. In the revision, and following the reviewer's request, we have also added additional PTA data from two other donors to complement the original PTA data from donor H1.

- Why were only 3 mutations from Nanoseq re-found? Sequencing depth?

Authors' response: It is due to a combination of depth and clone sizes. The LCM approach samples very few clones compared to the deep targeted NanoSeq approach. The more microdissections we do and the more polyclonal they are, the higher the chances of detecting a mutation previously found in the targeted NanoSeq data. However, the sizes of the clones are also important. The 3 mutations from the bulk targeted NanoSeq also found in the LCM have higher VAFs in the targeted NanoSeq data, reflecting larger clones. This is consistent with the LCM data in Fig 4a, which shows that these two clones are unusually large.

- What about the lymphoid cluster at the bottom of the tissue. Unaffected?

Authors' response: This is an interesting case. As far as we can tell it does not carry coding driver mutations in any of the genes identified in the study. This is not necessarily unexpected as only a fraction of B cells carry driver mutations in our data. However, it is also possible that there are clones carrying driver mutations in genes yet to be identified under selection, or driver events undetectable by targeted NanoSeq or whole-exome LCM sequencing, such as methylation events or structural rearrangements (e.g. gene fusions), which should be interesting lines of research for follow up studies.

Authors' action: We thank the reviewer for raising this interesting question. We have added a brief note in the main text to make the final point above.

- Figure 4b: Text (Line 319/320). The figure and the text need to be reordered for clarity. The text jumps very suddenly to conclude that the lymphoid-enriched structures are germinal centres without first showing this. Figure 4e needs to be shown before 4b, and possibly Extended Figure 6; specific germinal centre stains are needed.

Authors' action: Thank you for the helpful suggestion. We have now generated Xenium 5k data for a total of 7 donors. These data are helpful to complement the histology images in Fig 1 and 2, and the targeted methylation estimates of cell type composition. The new Xenium data is now presented in the

revised manuscript following the histology description in Fig 2, as a new EDF1. This allows us to refer to germinal centres earlier in the manuscript, pointing to the Xenium results.

- Extended figure 6: Please show which “GCs” / areas these images are from. e.g, Images with lower magnification. Text mentions CD3 staining, but there is no CD3 staining in the figure. CD19 staining is not enough to claim that these are GCs. CD19 staining seems to be less bright in mutant “GCs” than in WT “GC” so apparently less bright TNFRSF14 staining might stem from lower quality tissue

Authors’ action: Thank you. We have included additional IHC images to the supplementary figure as requested by the reviewer (new EDF5). Specifically, we have added a low magnification image of H&E-stained adjacent slides (EDF5e) and panels of IHC stains of adjacent slides, including CD3 (EDF5d). We note that the IHC results recapitulate the pattern of CD19 and TNFRSF14 staining seen in the original IF images.

- Figure 4e: Xenium data should be shown before clonal data, to first show that these are likely germinal centres. Figure 4e is mentioned in the text before 4c and 4d. Should also show T cells as these are also likely to contribute to disease

Authors’ action: In the revision we have included the Xenium 5k data for 7 donors (new EDF1) earlier in the manuscript, after the description of the histology and the methylation estimates of cell type composition in Fig 2.

- Text: “Plasma cells are the most common, consistent with figure 1a”. However, no plasma cell stain in figure 1a. Are conclusions drawn from Igk/Igl staining? Igk/Igl deposition could be immunocomplexes. It is difficult to interpret whether IgK/Igl+ stains are indeed plasma cells when image is not magnified and no CD138 staining is included. In addition, the tissue in Fig. 1a looks very different from what is shown in Figure 4e. In Figure 1a: Large B cell follicles with Bcl6 staining (i.e., germinal centre B cells) and T cell areas. Figure 4e shows a very different picture with no clear B cell or T cell areas. Why is the structure so different?

Authors’ action: Thank you for highlighting this area of confusion. The evidence that plasma cells are much more frequent than B cells in donor H1 comes mainly from the Xenium data, including the new Xenium 5k data. The reviewer also asks why the histology looks different in Fig 1a and Fig 4e for donor H1. The reason is that these images come from different biopsies from this donor, and that while in Fig 4e we show the low magnification image, in Fig 1a we zoomed on an area containing germinal centres. However, the full section overview is comparable between the two biopsies, shown below. Additionally, please also find below a figure demonstrating high expression of SDC1/CD138 in the Xenium data for donor H1. Expression of additional plasma cell markers are shown in the Xenium data in EDF1b.

Figure. a, Low magnification image of diagnostic biopsy for donor H1. **b**, Expression of SDC1/CD138 in the 380-plex Xenium Immune-Oncology Panel for donor H1.

- Text (Line 320): authors mention single or small number of clones, however, two germinal centres have 3 clones (out of 7 identified). 2488106G>A is found everywhere and might not even be specific to “GCs”

Authors’ response: Thank you. The main text in line 320 said: “*This revealed that individual germinal centres are largely dominated by single (or a small number of) clones in this donor. Independent start loss mutations were observed across multiple aggregates.*”. What we meant there was that any given GC in this donor was dominated by one clone (or very few). This is apparent in Fig 3b as mutations in individual GCs reach VAFs ~0.4-0.5, meaning that >80% of cells in the germinal centre carry them (i.e. individual GCs are largely clonal in this donor). We hope this clarifies the meaning of that sentence.

The reviewer then mentions that 2488106G>A is found “everywhere”. That is correct, but this is not inconsistent with our original statement above that any given GC is “largely dominated by single (or small number of) clones in this donor”. It is also important to note that 2488106G>A corresponds to start loss mutations (MII), which is a frequent hotspot in *TNFRSF14*. These are almost certainly independent events given the lack of sharing of other mutations between these GCs. This is why in the second sentence cited above we explained that “*independent start loss mutations were observed across multiple aggregates*”.

Authors’ action: We hope these clarifications are helpful. In the revised manuscript we have made minor edits to these two sentences to avoid confusion, clarifying that “2488106G>A” corresponds to the recurrent MII hotspot.

- Many germinal centres or microdissections do not contain any mutant clones. Could the authors please mention and comment in the text?

Authors’ action: We now briefly comment on this in the main text.

- Additionally, a comparison between microdissections with and without mutant clones is missing. Without it, it is difficult to interpret whether these mutations are indeed driving anything.

Authors’ response: Could the reviewer clarify this point? We do not fully understand what they would like us to compare. To clarify, and as explained above, when we say that these mutations are “drivers”

we mean that they are under positive selection (i.e. these mutations are driving clonal expansions), as shown by dN/dS.

- Figure 4d: Text (Line 336): “dominated by a single clone” although there still is a high degree of overlap between mutations. Could be multiple clones. I.e., Conclusions in line 338/340 on whole genome sequencing (WGS) cannot be drawn on 1 clone. WGS1 seems to have larger VAF of p.Y26D than Q345* and pM11 and pW12* – indicating that it is not just one clone, but multiple.

Authors’ response: By “dominated” in this (and previous) sentences we mean that the majority (i.e. >50%) of the cells in the microdissection share the same mutation, as demonstrated by their VAFs. Other clones are detected in them, but at very low VAFs.

Regarding the second point, we apologise for the confusion caused by our wording, which was accidentally ambiguous. When we said “*Despite the widespread clonal mixing, three microdissections in this donor were found to be dominated by a single clone*”, we meant “*were each found to be dominated by a different clone*”. What we meant is that in each of these microbiopsies a *different* clone represents >50% of the cells of each microbiopsy. This allowed us to detect copy number changes and structural variants in them. Please note that we then said: “*This revealed that one of the three clones had acquired at least four driver mutations...*”, clarifying that we referred to 3 different clones.

Authors’ action: We thank the reviewer for identifying this ambiguity in our text. We have fixed it in the revision.

Figure 5

- Text (Line 360): Could the authors please clarify the text for readability. Mention first that sequenced cells are classified into B, T, and other and based on which characteristics. Then, mention how many mutations were found in each.

Authors’ action: Thank you for the helpful suggestion. We have made the suggested change, which we think has improved readability.

- Mutations identified in T cells and non-lymphocytes need to be mentioned.

Authors’ response: Thank you. None of the driver mutations from targeted NanoSeq were detected in the T cells or non-lymphocytes sequenced. This is shown in Ext Data Fig 8 and it has been made clearer in the revised main text.

- Extended Figure 7: Should show a comparison between mutated and unmutated B cell clones: are SHM levels the same? Class switch recombination? (Text says 90.2% but this is not shown in the figures). Again, controls are missing, making it hard to get an idea of how unique this is and if relevant.

Authors’ response: We thank the reviewer for this comment but we would like to clarify that we only referred to the presence of SHM and class switch recombination as further evidence that the mutant cells are matured B cells that have undergone a germinal centre reaction. We did not necessarily expect the mutant B cells to have more SHM or more frequent class switch recombination.

Nevertheless, the reviewer makes an interesting suggestion for an additional analysis. We compared the extent of SHM between mutant and wild-type cells using two different approaches. Firstly, we extracted mutational signatures from mutations called de novo from our whole-genome single nucleus DNA sequencing. As expected, previously described mutational signatures associated with somatic hypermutation (COSMIC signatures: SBS9 and SBS85) were exclusively observed in cells independently identified as being non-naive B cells by assessment of V(D)J recombination and genotyping of mutations identified by Targeted NanoSeq in the immunoglobulin loci. This analysis

identified significantly higher mutation burdens and levels of SBS9 in *TNFRSF14*-mutant cells compared to wild-type cells (Fig. 5e and EDF11a-c). Furthermore, these elevated levels of SHM were even more pronounced in cells with both *TNFRSF14* and *CD274* mutations. Secondly, to compare the extent of SHM within the BCR, we used the full receptor sequences reconstructed for the recombinant antibody experiment. This confirmed that driver-mutant cells have significantly higher levels of SHM than WT B cells (Fig. 5e). We thank the reviewer for this suggestion, which has enabled us to identify an important difference between mutant and wild-type cells.

Authors' action: We have added details of our signature extraction approach to Supplementary Note 2 and incorporated the results of the two analyses described above to the main text, Fig. 5e and EDF11a-c.

- Figure 5a: Text says that mutant clones are highly polyclonal but they actually seem to be less polyclonal than somatically unmutated B cells. Each somatic mutation combination has one set of VDJ combinations. For instance, *TNFRSF14* Y26D and Y25Y(?) clones all have V1-48 and J-4. Do the authors suggest that somatic mutations occurred during early embryogenesis, during later somatic mutations in stem cells or after B cell development/VDJ-recombination? Line 369 suggests early embryogenesis; Line 388 suggests during first decade of life (why first decade is unclear); From line 388 and onwards, authors describe somatic mutations to occur after B cell maturation., ie not independent lineages, which seems to conflict with earlier statements. If indeed an embryonically introduced somatic mutation conferred advantage, one would expect to find clones of different VDJ combinations, meaning that somatic mutation occurrence AND advantage are independent of VDJ recombination. By contrast, their results are more consistent with somatic mutations occurring in single VDJ-clones through germinal center-related somatic hypermutation (or through another process) happening after VDJ-recombination.

Authors' response: We thank the reviewer for highlighting areas of confusion in the original text, and we apologise for any unintentional lack of clarity.

We try to clarify a few of the points raised by the reviewer below:

- “Text says that mutant clones are highly polyclonal but they actually seem to be less polyclonal than somatically unmutated B cells”. Both statements are correct. Driver mutations lead to clonal expansions, which as a result make the *TNFRSF14*-mutant pool less polyclonal than the wildtype pool, but they are still highly polyclonal, as evidenced by finding 135 different *TNFRSF14* mutations in a single donor.
- “Line 369 suggests early embryogenesis”. We apologise for the confusion. Our original sentence said “Phylogenetic analyses using whole-genome sequencing data then confirmed that most mutant B cells share no more than a few mutations, consistent with them being entirely independent lineages since early embryogenesis”. What we meant here was that “most mutant B cell lineages share no more than a few mutations...”. That is, independent B cell lineages with different VDJ recombination events and different driver mutations also do not share other passenger mutations genome-wide, which means that they are independent lineages since early development.
- “By contrast, their results are more consistent with somatic mutations occurring [...] after VDJ-recombination”. We agree. As the reviewer points out, most driver mutations are either clonal with a VDJ recombination event or subclonal to it, which is consistent with most drivers occurring after VDJ recombination. Although in those cases where a driver mutation is clonal with a VDJ recombination event (i.e. occurring in the same internal branch of the phylogenetic tree), we cannot resolve their relative order.

The *TET2* Q345* mutation is an interesting exception to the general pattern above as it clearly occurred before VDJ recombination, and it gave rise to at least two separate VDJ lineages. *TET2* is a clonal haematopoiesis driver, so it is easy to see how it may have led to a HSC clonal expansion from which both lineages arose. In this particular case, the phylogenetic tree places this *TET2* Q345* mutation in a branch within 174 mutations from the root of the tree. Multiple previous studies have shown that at

birth, HSCs carry ~80-100 mutations and that they accumulate ~16-20 mutations per year. The *TET2* Q345* mutation is timed to 10-174 mutations from conception, which suggests that it occurred at some point from early development to <5 years of age. To be conservative in the manuscript we said “first decade of life”. We have added this additional detail to the main text.

Authors’ action: Thank you for pointing us to areas of the text that were confusing. We have revised this paragraph to avoid the confusion above and to provide a few more relevant details.

- Text (Line 405/406): “Altogether, these data provide evidence of strong selection driving large numbers of clonally independent B cells to escape peripheral tolerance constraints during the pathogenesis of autoimmune thyroid disease.” This is an overstatement not backed up by data. There seem to be alternative explanations for these results, no healthy controls, large degree of variability within patients but analysis focused on one with strong *TNFRSF14* mutation load. Mutations seem to have occurred to a large extent (with exception of two clones with shared *TET2* mutation) after VDJ-recombination. Most importantly, there is no evidence that these mutated B cells leave germinal centers and go on to differentiate into plasma cells and produce autoantibodies.

Authors’ response: We hope that we have addressed these comments in our previous answers. This includes:

1. **Controls:** Providing 5 additional control datasets sequenced to high duplex depth that show no selection on *TNFRSF14* or *CD274*.
2. **Variability:** Variability between donors is expected given the large heterogeneity in their disease progression and their cell type composition. In fact, we find the level of convergence on these two driver genes remarkable compared to the heterogeneity observed at the level of histology or clinical manifestation. We have also already clarified above that selection on these genes is very strong across the cohort, even after excluding the donors with most advanced pathology and higher numbers of driver mutations (as formally shown by dN/dS ratios and Q-values).
3. **Timing of mutations:** As explained, we agree with the reviewer that most driver mutations probably occurred after VDJ recombination (although we note that it is often impossible to determine whether they occurred a bit before or around the same time). This should not be a criticism of their relevance as their occurrence after VDJ would be perfectly compatible with the pathogenic model that we present in the discussion as a possible interpretation.
4. **Plasma cells:** Where we have been able to study these mutations by single cell sequencing, we have shown that they most often occurred in B cells with SHM and class switch recombination. We also now provide evidence that at least some of these clones recognise classical autoantigens (TPO and TG).
5. **Tone and speculation:** Although we hope that the reviewer and the editor agree that the new data in the revision considerably strengthens and extends our original claims, in the revised manuscript we have also revised the text to better convey areas of uncertainty, including regarding a possible causal role of the mutations observed.

We thank the reviewer for pointing us to areas that required additional data or clarification. We very much hope that our answers above are helpful, and we believe that the additional data generated for the revision has strengthened the manuscript.

- Discussion: Text (Line 430): “extreme convergence. Tens-hundreds of clones per donor” This is not shown for B cells but all cell types.

Authors’ response: We hope that the explanations above and the additional single-cell data for a total of 3 donors provides more confidence that these driver mutations occur in B cells.

Final comment: We would like to thank this reviewer for the time devoted to evaluating our manuscript so carefully and for the many constructive suggestions, even when critical. We think that the manuscript has considerably improved as a result, with refinements to the text to avoid confusion and tone, and

with the addition of considerable new supporting data (incl. the much more extensive controls, additional Xenium and PTA data for more donors, and the recombinant antibody experiment).

Referee #2 (Remarks to the Author):

I co-reviewed this manuscript with one of the reviewers who provided the listed reports.

Referee #3 (Remarks to the Author):

Nicola et al. present a compelling study that describes the somatic mutation landscape of lymphocytes in autoimmune thyroid disease and provides evidence of recurrent mutations in B cell clones targeting immune-checkpoint pathways in the course of disease evolution. Using highly sensitive duplex exome and targeted NanoSeq with orthogonal single-nucleus DNA sequencing and spatial transcriptome approaches the authors identify a high burden of mutations that converge on TNFRSF14 and CD274 within B lymphocytes. Their model supports stepwise polyclonal mutagenesis in which numerous B cells accrue lesions, including biallelic loss of TNFRSF14, consistent with selection for checkpoint evasion across infiltrating B cells. While the work certainly sets a high bar for technical innovation and substantially advances our understanding of the link between somatic mutations in lymphocytes and autoimmune pathogenesis, several aspects of interpretation of the data would benefit from further refinement. Furthermore, while the advance is substantial, the technical limitations of the approach preclude dissection of the specific nature of B lymphocytes targeted by the somatic mutations (immature/mature/germinal center/terminally differentiated).

Major Concerns:

1. Figure 1a – why does the anti-Kappa/lambda IHC staining not identify the same B cell regions as the CD20 staining does? Please clarify concordance or discuss biological/technical reasons for discordance.

Authors' response: We thank the reviewer for highlighting this. The differential staining of the light chains and CD20+ B cell populations is likely the result of (i) most plasma cells being CD20- (PMID: 26100534) and (ii) some of the observed light chain staining being the result of antigen-antibody immune complex deposition. Immune complex deposition has long been observed in AITD (Kalderon & Bogaars, 1977, PMID: 337799). These immune complexes are often associated with plasmacytic infiltration, but not exclusively.

Authors' action: To address several reviewer comments, in the revision we have generated Xenium 5k spatial transcriptomics data from 7 donors. This provides more detailed information on cell type composition and cell states than the IHC shown in Fig 1a. Low-magnification Xenium overviews from the 7 donors are shown in EDF1a and zoomed in images on germinal centres and plasma cell rich areas are shown in EDF1b, including showing spatial expression maps for CD20 (*MS4A1*) and other relevant markers. We thank the reviewer for raising this issue.

2. As presented, the data don't resolve whether mutations arise/are selected in immature, GC, memory, or terminally differentiated plasma cells. I believe it is a crucial point to address, if at all possible. Please explain if this deconvolution can be performed and discuss the plausible selection pressures for TNFRSF14/CD274 lesions across these compartments.

Authors' response: We thank the reviewer for raising this important point. Several lines of evidence map the *TNFRSF14* and *CD274* mutations to activated germinal centre B cells and plasma cells, in different donors, as summarised below:

1. Drivers of MALT lymphomas (suggestive prior evidence): *TNFRSF14* and *CD274*, together with *TET2* and *TNFAIP3* are the commonest driver genes of MALT lymphomas of the thyroid (Wu et al, 2021, PMID:34021249), which develop from post-germinal center B cells. This strongly suggests that these mutations likely occur in post-germinal center B cells in our samples, but we demonstrated this in the original submission in two complementary ways, described below.
2. Spatial transcriptomics (Xenium) + laser microdissection: for donors H3, which had clear germinal centres dominated by single mutant clones, the combination of Xenium spatial transcriptomics and laser microdissection convincingly maps several of these driver mutations

to germinal centre B cells in this donor (please see Fig 4). In many of these microdissections the fraction of mutant cells is sufficiently high that these mutations must be present in B cells based on their cell type composition estimated by Xenium.

3. Whole-genome single-cell DNA sequencing (PTA) from donor H1: In the original submission, we included single-cell DNA sequencing data for 112 B cells from donor H1 (66 B cells, 38 T cells and 8 other cells). In this dataset, *TNFRSF14* and *CD274* mutations occurred exclusively in mature B cells, as evidenced by somatic recombination of their BCR, the presence of somatic hypermutation and class switch recombination. Xenium spatial transcriptomics on this donor confirms that most lymphoid cells in this donor (H1) are plasma cells, including in regions dominated by *TNFRSF14* mutant clones. It is important to note that, whilst this suggests that the mutations are present in plasma cells in this donor, selection could have occurred in the germinal centre B cells that gave rise to them. In the revision, we have included a new analysis of the single-cell data from donor H1 showing that *TNFRSF14*-mutant cells have considerably more SHM and SBS9 mutations than wild-type mature B cells from the same donor, consistent with these *TNFRSF14* mutations having previously existed in germinal centre B cells (see Fig. 5e and EDF11a-c).
4. New single-cell DNA sequencing data from an additional 180 cells from 2 more donors: In the revised manuscript we include single-cell data from an additional 180 cells for 2 donors, taking the total number of single cells with DNA sequencing data to 292 cells from 3 donors. These data strongly support the original conclusion, identifying 13 additional single cells with *TNFRSF14* and/or *CD274* mutations, all of which are confirmed to be B cells from analysis of their BCR sequences. Of note, the number of *TNFRSF14* and/or *CD274* mutant cells identified in these additional donors is consistent with the number predicted by the mutant cell fractions obtained from Targeted NanoSeq. Please see the new EDF8.

Authors' action: To address this question, in the revised manuscript we have included several additional datasets and analyses, including: (1) Xenium 5k spatial transcriptomics from 7 donors to better characterise cell type composition and histopathology, (2) additional PTA data from 2 more donors to extend the single cell analyses to a total of 3 cases, (3) mutation signature analyses of the whole-genome PTA data from donor H1, which revealed higher levels of SHM and SBS9 in *TNFRSF14*-mutant cells.

We hope that the answers above, the additional Xenium data, and the new single cell data is satisfactory to the reviewer. We agree, however, that future studies with better cell type resolution (perhaps enabled by a future sample collection compatible with flow sorting or by advances in dual single cell DNA+RNA sequencing) will be fascinating to better understand the impact of the immune checkpoint mutations uncovered here in different cell types and cell states.

3. Given that the small number of patients on which duplex sequencing and NanoSeq was performed, independent replication with predefined mutation/CNV search in larger number of patients that includes earlier stages of disease is appropriate. Doing this across different stages of disease could begin to address dynamics of accumulation and clonal evolution.

Authors' response: We thank the reviewer for this suggestion. We agree that this is an interesting question and this is why we provided an extension cohort in our study. The 3 pilot samples (H1-H3) where we first uncovered this phenomenon were highly advanced cases with extensive lymphocytic infiltration and thyroid destruction. To explore whether selection on immune checkpoint mutations can also be found in earlier stages of disease and in patients with an initial diagnosis of Graves disease, we then obtained an additional 11 samples. Overall, the ages of the 14 samples studied span seven decades, and a wide histopathological spectrum of disease activity/stage, including end-stage florid plasmacytic infiltration (e.g. H1), active disease with the formation of tertiary lymphoid structures amongst preserved thyroid architecture (e.g. H3, G3), and early stage disease with intact thyroid architecture and

little lymphocytic infiltration (e.g. G2, G4). We feel this provides some insights into the somatic driver landscape across disease states.

Making more definitive conclusions about the somatic driver landscape throughout the evolution of the disease would require much larger cohorts, ideally with longitudinal sampling. This is beyond the discovery-focused scope of this study, but we agree with the reviewer that exploring the driver landscape throughout the evolution of the disease would be fascinating. To this end, we are currently setting up a new prospective collection of samples with cryopreserved viable material for cell sorting and with detailed metadata. However, this effort will take several years to yield enough samples as unlike the goitrous thyroids of Graves disease and nodular thyroid disease, obtaining further high-quality tissue samples of Hashimoto thyroiditis is not trivial, given that this inflammatory condition rarely undergoes primary thyroid resection. Tissue biopsies of Hashimoto disease are often obtained incidentally during thyroid resections for unrelated reasons.

4. Single nucleus sequencing enabled the authors to couple the analysis of B cell VDJ repertoire to somatic mutations and provided critical insight into whether the mutations accumulated in clonally (by BCR) expanded B cells. Analysis of additional patient biopsies with snDNA-VDJ seq (or 5' GEX plus targeted amplicons or using other methods) is critical to determine mutation prevalence within and across expanded clonotypes.

Authors' response: Thank you. In the revision, we have included single-cell PTA DNA sequencing data from an additional 180 cells from 2 additional donors, which we hope provides valuable additional evidence complementing the single-cell DNA sequencing data from donor H1 that we included in the original manuscript.

Please bear in mind that PTA single-cell DNA sequencing is a very new technology, expensive and low-throughput (plate based). Prior to using PTA, we tried alternative methods to detect somatic mutations from single-cell RNA sequencing data, including using long read and target capture, but the genotyping fraction on genes of interest like *TNFRSF14* or *CD274* was very low (often below 1% per cell), which made the resulting data inadequate for inclusion in the manuscript. We are currently investing in automation and miniaturisation to increase our ability to do single-cell PTA DNA+RNA on many hundreds or thousands of cells, but this will take time to be ready for use.

We hope that the inclusion of a total of 292 cells with high-quality DNA sequencing and low allele dropout helps satisfy this reviewer request, even though we agree that a much larger single-cell study will be fascinating to link the somatic mutations detected with VDJ clonotypes and with phenotypes, as single-cell DNA+RNA sequencing methods mature. For context, please note that recent high-impact papers have been done on tens or low hundreds of cells with PTA given the cost of the technology (e.g. Dong et al, Science, 2025, PMID:40093089; Miller et al, Nature, 2022, PMID:35444284), which is a small fraction of the data that we have included in our study.

On a related note that this reviewer may find of interest, in the revision we have also improved on the reconstruction and analysis of VDJ sequences from donor H1, combining DNA and RNA sequencing from each cell when possible, and testing autoreactivity to TPO and TG using recombinant antibodies generated from these sequences (Fig. 5f and EDF11d).

We hope that the novelty of the core discovery (the widespread existence of B cell clones with immune checkpoint somatic mutations in AITD), the unprecedented extent of convergent positive selection in an autoimmune disease (tens to hundreds of independent clones with convergent driver mutations per

donor), the combination of state of the art methodologies (exome-wide NanoSeq, whole-genome single-cell DNA sequencing, Xenium, LCM spatial DNA sequencing, etc), and the extensive additional data added to the revision, all together is sufficient to warrant publication. Even though we acknowledge that future follow up studies on larger cohorts and with scDNA+RNAseq will be fascinating to expand on these discoveries.

5. Extended figures need complete legends (sample number, statistical tests, all annotations, etc).

Authors' action: In the revision, we have improved the EDF legends as requested.

6. Several methods defer to “Lawson et al., in preparation.” Please include sufficient detail here (or Supplementary Methods) to permit evaluation/future reproduction.

Authors' action: Thank you. We now include further details on targeted methylation and dNdSshm in the Methods section.

Minor Concerns

1. Line 66 – Consider “neglect” rather than “ignorance”?

Authors' action: We have replaced “ignorance” with “neglect” as requested.

2. Line 94 – Consider “most common” instead of “commonest”?

Authors' action: We have replaced “commonest” with “most common” as requested.

3. Figure 4E – Define criteria used to separate mature versus immature plasma cells.

Authors' action: We have removed the specification between mature versus immature plasma cells. While the Xenium 5k panel (now used in extended data figure 1) contains a larger gene set for broader cell type specification, it contains fewer immune-focused genes required to robustly call plasma cell subtypes compared to the 380 immune panel (used in figure 4e). To provide consistency across datasets, we have removed the specification of mature versus immature plasma cells and are instead presented as a single population through the manuscript.

4. Please describe how TNFRSF14 and CD274 mutations were assigned to T cells and what the expected consequences are. What is the T cell clonality and driver mutational profile for these events? Please defined the confidence in assignment of lymphocyte identity across ST vs. B/T aggregates.

Authors' response: There may be some confusion here, as we did not find any *TNFRSF14* or *CD274* mutation in T cells. All the mutations in these two genes in the single-cell DNA sequencing data were found in B cells.

Full details on how we classified single cells into B cells, T cells or non-B/T cells and how we detected driver mutations in them can be found in the **Methods** section. Briefly, cells were classified as B or T cells based on the presence of somatic recombination of their BCR or TCR loci, respectively. Presence of somatic hypermutation (SHM) in the IGH locus was also found in most B cells, confirming them as non-naive B cells. Class switch recombination was also detected in a majority of B cells (donor H1). The identification of B and T cells in lymphoid aggregates was based on transcriptomic profiles studied with Xenium.

Driver mutations found by deep targeted NanoSeq of the same donor were genotyped across all single cells. This revealed that all *TNFRSF14* and *CD274* driver mutations detected in single-cells by this unbiased approach were found in B cells, as described in the main text. Please also note that, for reasons of space, **Fig. 5a** only shows B cells. However, the full heatmap with B cells, T cells and non-B/T cells sequenced by PTA in donor H1 is available in **EDF7** (along with additional PTA data for donors H2 and H8 in **EDF8**), and it shows that no driver mutations in *TNFRSF14* and *CD274* were found outside of B cells. We hope that this resolves the confusion.

Authors' action: In the revision, we have improved the description of the annotation of B and T cells from single-cell DNA sequencing data in the main text and in the Methods section to avoid confusion. We also more clearly explain that all mutations in *TNFRSF14* and *CD274* in the single-cell data were found in B cells, providing the additional information discussed above.

We would like to thank the reviewer for their time and constructive suggestions, which we think have helped us improve the manuscript.

Referee #4 (Remarks to the Author):

I co-reviewed this manuscript with one of the reviewers who provided the listed reports.

Referee #5 (Remarks to the Author):

This manuscript by Nicola, Lawson and colleagues leverages ultra-deep high-accuracy duplex sequencing and other technologies to assay rare mutations in autoimmune thyroid disease. In thyroid tissue from patients with Hashimoto thyroiditis and Graves disease, afforded by their sequencing approach, the authors identify a high frequency of very low allele fraction somatic mutations in immune checkpoint genes in polyclonal lymphocyte populations. They demonstrate that non-synonymous mutations in TNFRSF14, CD274 and other immune checkpoint genes are under positive selection and that such mutations are not observed in aged B cells from a PBMC population. Using an orthogonal approach, they genotype these mutations in single cells and identify B cells as the population of interest. While each mutation is individually low frequency in the population, they are highly polyclonal and collectively affect a significant fraction of the B memory cell population. These likely contribute to Hashimoto and Graves pathology by abrogating autoinhibitory processes that govern self-reactive B cells. The unexpectedly rich landscape of polyclonal immune checkpoint mutations in B cells very likely plays a central role in the pathogenesis of Hashimoto and Graves.

By extension, this work tests the longstanding hypothesis that autoimmune diseases can arise through somatic mutation of genes in lymphocytes that negatively regulate self-reactive clones. The authors' results are highly impactful and change our understanding of autoimmune diseases, which will lead to many follow-up studies to validate the role of polyclonal somatic immune mutations across other autoimmune diseases, and to tease apart the precise role and timing of polyclonal mutant immune cells in disease etiology.

Authors' response: We are very grateful for this positive assessment of our manuscript.

Overall, the study is very carefully performed, and the manuscript overall is clearly written. My one main reservation relates to the appropriateness of the negative controls of non-autoimmune B cell populations the authors used as a reference.

Major comments:

1. The lack of an appropriate negative control is a concern. While the authors did not observe the same positively selected mutations in normal PBMC memory B cells from aged donors (Fig. 3C), I feel this is an inadequate comparison. A more appropriate negative control would be a similarly local sample (biopsy or LCM) with proliferating B cells in place and possibly under selection from non-autoimmune donors. Such populations could be sampled from lymph nodes, spleen, or bone marrow. The mutations observed are already a polyclonal mix from such a small locale. Attempting to identify such mutations from the blood may significantly underestimate their presence in non-autoimmune implicated B cells. While the genes implicated conform to the hypothesis, an alternative interpretation, if these mutations are locally present in non-autoimmune tissue, is that these confer a selective advantage in B cell maturation but are not enough to drive pathogenesis of autoimmunity (i.e., other autoinhibitory mechanisms are present). It would add significantly to the study if the authors were able to close this gap.

Authors' response: We thank the reviewer for raising this question. Providing new targeted NanoSeq data from several secondary lymphoid organs is a large task, but we have made a considerable effort to address this as comprehensively as possible. Below we describe 5 new control datasets: lymph nodes, spleen, tonsillitis, LCM of lymphoid aggregates in healthy biopsies of the gastrointestinal tract, and non-autoimmune thyroid, sequenced to high duplex depth.

Sensitivity: Before we summarise the new control data, we would like to alleviate a concern from the reviewer. The reviewer notes that "*the mutations observed [in AITD] are already a polyclonal mix from such a small locale. Attempting to identify such mutations from the blood may significantly underestimate their presence in non-autoimmune implicated B cells*". This suggests to us that the

reviewer is concerned that we may be less sensitive to driver mutations in highly polyclonal samples. It is important to clarify that this is not a problem as targeted NanoSeq is a single-molecule sequencing method and not a VAF-based sequencing method. Whereas traditional VAF-based mutation detection has low or zero sensitivity in highly polyclonal samples, this is not the case for single-molecule mutation detection. A detailed explanation is provided at the start of the response to reviewers, at the request of the editor. Briefly, for Nanoseq and other single-molecule duplex sequencing methods, it is equally easy to correctly conclude that 10% of the cells of a biopsy carry mutations in a given gene if the sample contains a single large clone at 10% frequency, or 10,000 small clones at 0.001% frequency. In practice, we can reassure the reviewer that the rarity of mutations and the absence of significant positive selection on *TNFRSF14* and *CD274* in circulating B memory cells is real and not due to sensitivity problems in highly polyclonal samples.

New control datasets: Although we can reassure the reviewer that the lack of *TNFRSF14* and *CD274* driver mutations in B memory cells from non-autoimmune donors is not due to sensitivity, it is nevertheless valuable to explore whether these driver mutations are observed in other secondary lymphoid organs or tissues, as suggested by the reviewer. To address this question, we have performed deep targeted NanoSeq on a range of additional samples:

To address this question, we have performed deep targeted NanoSeq on a range of additional samples:

1. Lymph nodes from 4 healthy donors sequenced to a cumulative 10,740 duplex depth (dx).
2. Spleen from 30 healthy donors sequenced to a cumulative 21,748 dx.
3. 183 laser microdissection cuts of lymphoid aggregates from a range of tissue samples from 5 healthy donors, including colon (50 cuts from 3 donors; 1,093 dx), Peyer's patches from ileum (46 cuts from 1 donor; 478 dx), and tonsil (87 cuts from 1 donor; 2,085 dx).
4. Tonsil samples obtained from the tonsillectomies of 5 donors with a history of chronic tonsillitis (cumulative 19,803 dx).
5. Thyroid samples from 5 donors with nodular goitres but without AITD (cumulative 7,024 dx).

These data confirm that mutations in *TNFRSF14* and *CD274* are very rare and not under selection across a range of tissues, including healthy secondary lymphoid organs, tertiary lymphoid structures, chronically inflamed tonsils and goitrous thyroids (Q-value > 0.1 for both *TNFRSF14* and *CD274* across all control cohorts). This strongly complements the original control data from flow-sorted PBMCs B memory cells sorted from PBMCs from healthy donors and further demonstrates the remarkable and highly specific nature of the *TNFRSF14* and *CD274* mutations that we report in AITD. We thank the reviewer for raising this comment, which we think has strengthened our manuscript.

Authors' action: Given that the request of additional control data was made by two reviewers and by the editor, we have made a large effort to address this as comprehensively as we could with 5 additional control datasets. In total, the new control data includes samples from 44 donors and a cumulative duplex depth (across 725 genes) of 62,971 dx. Together with the original control data, this surpasses the aggregate duplex depth of the original AITD data.

We have updated the manuscript to describe these additional control samples and have added a new main text figure panel (Fig. 3c, shown below) and a new Ext Data Figure (EDF4).. These data strongly support the original conclusions of the study.

Figure. Mutant cell percentage estimates for *TNFRSF14* and *CD274* in AITD samples with >25% lymphocyte fraction and 6 control data sets. Upper and lower bound estimates are shown for each group (aggregated across donors weighted by duplex coverage) and midpoint estimates for each donor (points)

Minor comments:

1. Related to the point above, it would also be valuable to see a set of negative control genes highlighted more clearly. At this depth of coverage and accuracy, what is the rate of mutation in genes not expressed in B cells or those that are expressed but not necessarily implicated in immunity per se (e.g., housekeeping genes)?

Authors' response: Thank you. Although this is an easy analysis, we fear that it could be misleading for the readers. This is because the total number of mutations per gene is confounded by gene length, sequence composition, and duplex depth. This is why it is important to use dN/dS-like approaches that control for these confounding factors to provide a statistically robust measure of selection (or enrichment) over neutral chance. Please note that the dN/dS methods that we have used (dNdScv, dNdSshm, sitednds, withingenednds) control for the depth of sequencing and for the confounding factors above, including variable duplex coverage across genes. These methods also use synonymous mutations in each gene (and in other genes) as internal controls. This is more reliable and robust than comparisons across genes (e.g. drivers vs house keeping genes) given the multiple confounder factors across genes. Adequately accounting for the impact of the mutations is also important, as selection in some genes can act only at specific high-impact sites. For example, imagine a driver gene (A) of 100 codons and a passenger gene (B) of 1000 codons. All other things being equal, under neutrality we expect 10 times more mutations by chance in B than A. Imagine that 10 codons in gene A are potent driver sites with

mutations enriched by selection at these sites by 50-fold. In that case, we may expect to see 60 mutations in gene A (10 neutral and 50 drivers) for every 100 mutations in gene B (all passengers, 0 drivers).

We hope that this helps reassure the reviewer that our analyses are appropriately controlled. However, to satisfy this request below we show dN/dS ratios and mutant cell fractions (the estimated fraction of cells with a mutation in the gene) for the driver genes in AITD and for a randomly selected set of 30 other genes in the panel. This figure shows how most driver genes in AITD are mutated in >1% of cells in the biopsies, whereas these frequencies tend to be much lower for other genes. However, raw mutant cell fractions, as explained above, are misleading and long genes are expected to carry passenger mutations in a fraction of cells. dN/dS ratios control for this and are a more meaningful measure of selection and functional importance. Interestingly, the figure below also shows suggestive signals of selection for *CARD11* and *FAS*, which did not meet our stringent q-value cutoff. We had previously noted in Supplementary Note 1 the presence of several well-known *CARD11* hotspot mutations within the Targeted NanoSeq data.

Figure. Mutations and evidence of selection on AITD driver genes compared to other genes in the immune-gene panel not under significant selection. Top panel shows the number of mutations per gene coloured according to their protein coding impact. Second panel shows the dN/dS ratios estimated for different mutation classes (in linear scale) (capped to a maximum value of 50, as *TNFRSF14* dN/dS ratio for truncating mutations reaches 141). The third panel shows dN/dS ratios in log scale, showing only classes where the CI95% deviate from 1. Note that *CARD11* and *FAS* both show tentative signals of selection in AITD, but do not reach q-value significance with the conservative cutoff chosen in the study.

2. The repeated reliance on another forthcoming paper (Lawson et al, in preparation) to support several claims somewhat detracts from the present study. If these are not published together, data supporting claims made here, should be presented here or at least summarized to the level needed for this paper to stand on its own.
 - a. The authors refer to a modified version of dNdScv (dNdSshm) and refer to the other manuscript for details. They do describe adaptations to dNdScv in the methods section – is that dNdSshm? If not, this method should be described in broad terms there.

Authors' response: We thank the reviewer for this suggestion, with which we entirely agree. In the revised manuscript we include a more detailed description of dNdSshm in the Methods section. Initially, we had hoped to co-submit both manuscripts, which was the reason for citing the other manuscript in this way. Please note that we also provide the code for dNdSshm in the current manuscript (Supplementary Code).

Authors' action: We now provide a detailed description in the revised version of the **Methods** section and have reduced the references to Lawson et al.

- b. The authors claim “In a separate study [Lawson et al, In preparation], we show that several of these genes acquire driver mutations in B or T memory cells during normal immune ageing, in the absence of autoimmune disease” (line 274) is valuable information for this study, but no supporting data is shown.

Authors' response: Thank you. Please note that these data were already shown in the manuscript. The original Fig 3c showed a side-by-side comparison of AITD vs non-autoimmune B-memory cells (from Lawson et al, in prep). The figure shows the significance of each gene in each dataset by dNdSshm, and the estimated mutant cell fractions (the fraction of cells with mutations in each gene) estimated in each dataset. Additional comparisons for CD4-memory and CD8-memory cells for the list of AITD driver genes were also provided in EDF4. These data are also available in the Supplementary Code.

In keeping with suggestions from Reviewers 1 and 2, we now present additional comparative plots from a further 5 control populations (EDF4). These show a small number of genes to be under positive selection in non-autoimmune samples (including clonal haematopoiesis drivers like *DNMT3A* or *TET2*), as we also saw in the original B-memory control data. However, many drivers identified in AITD, including the immune checkpoints *TNFRSF14* and *CD274*, remain exclusively under selection in the AITD cohort.

Authors' action: We have now added further comparative plots further demonstrating key immune checkpoints are under positive selection in AITD, and not in control populations (Fig 3c and EDF4). The supporting data for all 6 control datasets can be found in the Supplementary Code.

3. The authors have spatial transcriptomics data from thyroid sections annotated with spatially distinct B cell populations. The lack of connection to changes in cellular phenotype despite having such data is a missed opportunity. Are there any transcriptional changes in B cell clusters with *TNFRSF14* mutations when compared to adjacent populations without such a mutation? Could these data be used to confirm the reduction of *TNFRSF14* or *CD274* gene products in these populations?

Authors' response: Thank you. We agree that this is an interesting question. However, the current dataset is unsuitable to study this for several reasons. First, the Xenium data slides are too distant from the LCM sections to enable the annotation of enough mutant and wild-type areas for this analysis. Second, the analysis is further complicated by subclonality, as most dissected areas are not clonal and contain other cell types, which are likely to confound the gene expression analysis. Together, we do not

have more than a few examples of semi-clonal mutant and wild-type areas with matching Xenium data, which means that we are underpowered to address this with the current data. Third, and importantly, we expect the phenotypic effect of these mutations to be context dependent. For example, *TNFRSF14* mutations may elicit their effect in germinal centres during the interaction of B cells with Th cells, or during the interaction of B cells with their antigens (e.g. by lowering the B cell activation threshold), but even if this leads to clonal expansions, the transcriptomic effects could be transient and may not be detectable in other contexts (e.g. in mutant vs wildtype plasma cells resulting from these clones). For these reasons, we feel that the current dataset is not appropriate for this analysis. Our hope is to revisit this question with single-cell DNA+RNA sequencing in a future study, to have the appropriate single-cell and cell state resolution, as well as the necessary statistical power from many independent cells or clones. However, generating these data requires setting up a new prospective sample collection, as high-quality single-cell DNA+RNA sequencing data will need cryopreserved viable cell suspensions, which we cannot generate with the current tissue blocks.

Although we do not have an analysis of the transcriptomic consequences of the driver mutations observed, in the revision we have been able to include an interesting analysis of the mutational signatures present in mutant vs wild-type B cells. This analysis reveals that *TNFRSF14* mutant B cells have much higher rates of SHM in their IGH as well as higher rates of SBS9, suggesting that mutant cells underwent more rounds of affinity maturation or spent a longer time in germinal centres, providing some additional information on the phenotypic impact of these mutations.

The reviewer also asks whether these data could help confirm the reduction of *TNFRSF14* or *CD274* gene products. This is an easier and more focused comparison not affected by some of the confounder issues above. Although truncating mutations may lead to lower transcript levels through nonsense mediated decay, the effects may be modest on RNA levels. However, we can address this question more definitively with IHC or IF. This has two advantages: (1) we are staining for protein abundance rather than transcripts, which should provide clearer signals, and (2) we saved unstained tissue sections for IHC or IF adjacent to the LCM sections used in the study, which allows for unambiguous mapping of clones between LCM and IHC sections. In the original manuscript, we presented immunohistochemistry highlighting reduced *TNFRSF14* expression in a lymphoid aggregate from donor G5 consisting of two distinct *TNFRSF14*-mutant populations that together amount to almost all B cells being *TNFRSF14*-null (Fig. 4a). In a second donor (H3), we used immunofluorescence to highlight progressively reduced *TNFRSF14* expression across three lymphoid aggregates ranging from *TNFRSF14*-wildtype, *TNFRSF14*-mutant (heterozygous), and *TNFRSF14*-mutant (compound heterozygous) (EDF5). To provide additional confidence in our immunofluorescence data, we have extended EDF5 to show paired IHC for the same trio of germinal centres.

Authors' action: As described above, in the revised manuscript we now include a new mutational signature analysis revealing an increased rate of SHM and SBS9 in mutant antigen-experienced B cells (Fig. 5e and EDF11). We also provide additional IHC data (EDF5) to complement previous IF data, showing a dosage-dependent effect of somatic mutations on *TNFRSF14* expression levels.

4. Can the authors more clearly lay out differences between Hashimoto thyroiditis and Graves disease? These are now analyzed as one disease, yet they have an opposite effect on hormone secretion. Are there differences in genes under selection, or in properties of mutant clones (e.g. clone size).

Authors' response: We thank the reviewer for raising this point. It is important to clarify the relationship between these diagnoses, particularly in our sample cohort. From a pathophysiological point

of view, Hashimoto and Graves are best considered as two different clinical manifestations of autoimmune thyroid disease (AITD), rather than two different diseases. Both involve a fundamental loss of tolerance and B and T cell infiltration of the thyroid parenchyma. Graves manifests as hyperthyroidism due to activating autoantibodies against the thyroid stimulating hormone receptor (TSHR), whereas Hashimoto manifests as hypothyroidism due to the progressive thyroid tissue destruction. This is even more relevant in our sample cohort, as pathognomonic features of Hashimoto thyroiditis, such as TPO/TG autoantibodies or intrathyroidal lymphocytic infiltration were present in all Graves cases. For these reasons, our clinical AITD collaborators (endocrinologists and histopathologists), have cautioned us against classifying samples into Hashimoto and Graves, and have requested that we refer to them as AITD independently of their original clinical diagnosis.

Nevertheless, this is an interesting suggestion, and we can certainly perform a separate dN/dS analysis for donors with an original diagnosis of Hashimoto or Graves. Reassuringly, this reveals strong positive selection for *TNFRSF14* and *CD274* mutations in both subsets of samples: q-values $<1e-15$ and $5.1e-5$ for *TNFRSF14* in the Hashimoto vs Graves subset, and $<1e-15$ and $7.4e-4$ for *CD274*.

Given the caveats above and the fact that we have more analyses and data in the manuscript than we can appropriately describe within the available space, our preference would not to include this analysis in the manuscript. But we think that this would be an interesting analysis for a study in a much larger cohort.

5. This is perhaps outside of the scope of the study, but it would underline the significance of the authors' findings if they could confirm if *TNFRSF14* and *CD274* mutant clones are indeed self-reactive. This could e.g. be addressed through a patient-derived thyroid organoid model, B cell isolation, and co-culture (PMID 34916298).

Authors' response: Thank you for the interesting suggestion. We agree that this would be very valuable. We had originally accepted that this was indeed outside of the scope of our study, but motivated by this question, in the revised manuscript, we provide a new set of experiments that support the self-reactivity of several *TNFRSF14*-mutant clones.

Using the single-cell DNA sequencing data generated for donor H1 (a donor dominated by plasma cells), we have reconstructed with high confidence the complete heavy and light chain BCR sequences from 32 B cells from this donor (see Methods for details). We then generated recombinant antibodies from these 32 sequences on a human IgG backbone using a commercial provider (Biointron), and each of these antibodies was tested for binding to TPO and TG using 12-point ELISA assays. Interestingly, at least 7 of these 32 recombinant antibodies were found to be self-reactive against either TPO (n=6) or TG (n=1). 3 of the 7 self-reactive clones carried mutations in *TNFRSF14* and/or *CD274*, and 1 in *BRAF*. 6 of the 7 had detectable class switch recombination. Altogether, this confirms that a sizable fraction of mutant clones (as well as wild type clones) are self-reactive against TPO in this donor.

It is important to remember that whereas TPO and TG are two classical autoantigens in AITD, many self-reactive clones could be recognising other autoantigens. To explore whether some clones without ELISA reactivity to TPO or TG could be recognising other self-antigens, we also ordered 8 of the BCR sequences in a rabbit IgG backbone. 5/8 of these antibodies were self-reactive in normal thyroid biopsies by IHC. Reassuringly, this included 3 antibodies reported as anti-TPO positive by ELISA and one anti-TG positive antibody. IHC also identified a further *TNFRSF14*-mutant clone with evidence of self-reactivity (Fig 5f), consistent with evidence of weak binding to TPO in the rabbit IgG ELISA (EDF11d). Since not all autoantigens may be present in the biopsies with detectable levels at the time of sampling or may not be appropriately preserved or retrieved by the IHC protocol, the fraction of self-reactive clones could be higher. Interestingly, 3 ELISA-positive anti-TPO clones did not carry a detectable driver

mutation. This is not entirely surprising as in the context of an established polyclonal autoimmune response the localised loss of peripheral tolerance is expected to enable other clones to proliferate.

We hope that the demonstration that a good fraction of mutant clones are self-reactive (particularly against TPO) strengthens the hypothesis that these mutant clones could contribute to the autoimmune response.

Authors' action: We now include new data and a new Fig. 5f and EDF11 detailing the heavy and light chain BCR reconstruction from 32 B cells, followed by recombinant antibody synthesis, anti-TPO and anti-TG ELISA binding tests, and IHC, to demonstrate the self-reactivity of 8 independent clones.

6. How are the cumulative fractions of mutant cells calculated? What assumptions go in there? E.g. are the authors implicitly assuming that mutant clones have only one mutation per cell here? This should be detailed in the methods section.

Authors' response: Thank you for highlighting that the Methods section does not describe this in sufficient detail.

The calculation of estimated mutant cell fractions was done as described in several previous studies. Briefly, we provide a lower bound and an upper bound aggregate MCF using two alternative assumptions. Assuming that each cell could carry up to two driver mutations in a given gene (biallelic loss), using the sum of duplex VAFs for all mutations in a gene in a sample provides a lower bound estimate of the aggregate mutant cell fraction. Assuming that each cell carries a single driver mutation in a gene, $2 * \text{sum}(\text{duplexVAF})$ provides an upper bound estimate for the mutant cell fraction. When we refer to MCFs in the manuscript we typically provide the range between both bounds (e.g. Fig. 1e&f, Fig 2c, and the old Fig 3c), unless stated otherwise (e.g. Fig 2d heatmap). It is important to note that this range is robust to a broad range of assumptions. For example, if mutant cells carried a single driver mutation and copy-neutral LOH or haploid copy number loss of the wild-type allele, the bounds above remain valid. We first described these equations and their rationale in detail in Martincorena et al (Science, 2018, PMID:30337457), and we have used the same approach in later studies, most recently using targeted NanoSeq (Lawson et al, Nature, 2025; PMID:41062696).

Authors' action: We now provide a detailed description in the revised version of the **Methods** section.

7. The manuscript at multiple positions jumps forward to later results, mainly to make the point early on that these mutations occur in B cells (e.g. lines 139-141). This is somewhat inelegant – it would be better to tell a linear story.

Authors' response: Thank you. We understand this criticism and we considered alternative approaches. However, not disclosing that the mutations occur in B cells early in the manuscript is likely to cause unnecessary confusion to readers, so we decided to trade elegance or linearity for clarity.

Authors' action: Following this suggestion, in the revision, we have reduced the number of references to later sections of the manuscript. For example, we now describe the Xenium data earlier (immediately after the description of the histology of the samples) to be able to refer to germinal centres early in the manuscript without reference to later sections. We have only retained one reference to a late section (to clarify that the mutations occur in B cells), as this helps understand and interpret the subsequent analyses and provides important context to the results. We hope that this compromise is acceptable, and we thank the reviewer for helping us streamline the text in other sections.

8. Lines 358-361 – in the single-cell sequencing section, the authors assay 112 cells and use an elegant approach to assign cell type. A few sentences to describe this could be added to the main text. E.g. it would be helpful to the reader to know that 66 of the cells are B cells to really follow what's written in lines 358-361.

Authors' action: Thank you for the excellent suggestion. We have added these details to the relevant main text section, which we think has made it clearer.

9. The text and scale bars in the figures are often too small to read. Fig. 3c (and Ext. Data Fig. 4) needs a color legend for the significance boxes. Figure 4 is hard to follow with Zooming in extensively.

Authors' action: Thank you. In the revision, we have tried to increase font sizes and improve legends on several figures, including in Fig 3c, as suggested.

10. Data availability: the authors should make the raw sequencing data available through a repository.

Authors' action: Thank you. We will of course do so. All sequencing data will be made available through EGA before publication but after review, to avoid multiple submissions of additional datasets during revision. We have also included a detailed supplementary code file (R markdown) to ensure the reproducibility of our analyses, which we hope will also help others in similar future studies.

We would like to thank the reviewer for their time and effort to provide detailed and constructive feedback, which we think has considerably improved the manuscript.

Referee #6 (Remarks to the Author):

I co-reviewed this manuscript with one of the reviewers who provided the listed reports.

Referee expertise:

Referee #1: autoimmunity, genetics

Referee #3: autoimmunity, B cell development

Referee #5: somatic evolution

Reviewer #1:

This reviewer was concerned with the evidence that the mutations cause disease. The first major concern raised highlighted the importance of showing positively selected mutations in antibody secreting plasma cells. The authors somewhat rephrased our request as “additional evidence of mutations occurring in B cells (...)” as the point to be addressed. They have now edited the text from “B cells” to “B cells or plasma cells” but have not provided additional evidence that these mutations indeed are found in plasma cells.

We agree that B cells could in theory contribute to breach in tolerance through additional mechanisms to antibody secretion e.g. through antigen presentation. However, the authors have not provided evidence of this either.

Authors’ response: Thank you. We are happy to provide additional clarification, as there is an important piece of evidence and some relevant context that we did not provide in our response.

In our answer, we explained that B cells in autoimmune thyroid disease (AITD) do not contribute to pathogenesis only as plasma cells, but also through their action as professional APCs, their interaction with T-helper cells, their generation of ectopic germinal centres, and their production of cytokines (Hampe, 2012, PMID:23807906; Lund, 2008, PMID:18417336; Getahun et al, 2019, PMID:30883216). In the revision we provided new Xenium and single-cell DNA sequencing data demonstrating that the driver mutations in immune checkpoint genes *TNFRSF14* (HVEM) and *CD274* (PDL1) are found in different donors in germinal centre B cells (including clonal TLSs, which are a key part of AITD pathogenesis and precede the formation of plasma cells) and in plasma cells. This is important as both cell types are causally involved in AITD pathogenesis.

It is also important to provide two additional clarifications to our original response. First, the histology of AITD varies markedly across patients in our cohort, a well-known feature of AITD that is believed to represent different disease subsets and potentially different stages of disease progression (Li et al, 2010, PMID:22555173). Specifically, some patients present with extensive plasma cell infiltration (particularly patient H1 in our cohort) whereas others present with a characteristic pattern of TLS formation (e.g. H3). Second, it is important to note that biopsies from AITD are very rare. Our cohort was a retrospective collection of frozen samples collected over several years by our collaborators. Unfortunately, we cannot isolate plasma cells by cell sorting using cell markers from frozen samples that have not been cryopreserved. Instead, we used a combination of advanced genomic methods to map the immune checkpoint mutations to germinal centre B cells (as shown by Xenium + laser microdissection + low-input DNA sequencing of individual TLSs) or to plasma cells (as shown by a combination of Xenium and new single-nucleus whole-genome sequencing, showing mutations in cells with somatic recombination of their BCR, SHM in their BCR, and class-switch recombination). We hope that this provides useful context.

Importantly, we notice that in the revision we did not include information on whether the 13 cells found to have *TNFRSF14* and/or *CD274* mutations in donors H2 and H8 had undergone class switch recombination. In response to this reviewer's comment, we have assessed how many of the 13 mutant cells have undergone class switch recombination and found this to be the case for all 13 cells. This maps the driver mutations to effector cells (plasma or B memory cells) in patients H2 and H8, as well as H1,

extending this observation to all 3 donors with single-cell DNA sequencing data. Of note, for patient H2, only 4/20 sequenced B cells had undergone class switch recombination (two of these have *TNFRSF14* mutations, of which one also has a *CD274* mutation). By contrast, donor H8 had a substantially higher proportion of B cells with evidence of class switch recombination (30/34), consistent with the greater proportion of plasma cells identified by Xenium spatial transcriptomics (EDF1a). All 11 of these class-switched cells had *CD274* mutations (of which two also had *TNFRSF14* mutations). This information has also been added to Extended Data Fig. 8.

We have now included this important detail in the manuscript. We thank the reviewer for motivating us to confirm this.

The only evidence seems to come from a single donor (H1) in which Xenium spatial transcriptomics suggests that virtually all cells in the tissue are plasma cells. This is surprising, as there are no B cells nor germinal center B cells identified in this tissue that could give rise to these plasma cells (there is no or little identification of T cells or thyroid epithelial cells either). At the very least, for this to be believable, they should show staining (IF or IHC) with key markers (eg. CD3, CD19, CD138, Blimp-1) to verify whether most of this tissue indeed is made up of plasma cells.

Authors' response: We are happy to clarify this, as the evidence requested of both plasma cells and B cells is already provided in several figures in the manuscript. We summarise the evidence here:

1. **IHC and ISH on a diagnostic biopsy from patient H1** (Fig 1a). In Fig 1a, we provided H&E, CD20 IHC (B cells), CD3 IHC (T cells), BCL6 IHC (germinal centres), and in-situ hybridisation for kappa/lambda light-chain mRNA expression for patient H1. The staining clearly shows germinal centre B cells (CD20+ BCL6+ kappa/lambda- aggregates), T cells (CD3+) and plasma cells (CD20- kappa/lambda+ areas by ISH) in this diagnostic biopsy from patient H1. This is a separate biopsy (a diagnostic FFPE block) to the one used for sequencing from patient H1, but clearly shows the presence of B cells and germinal centres in the thyroid of this patient.
2. **Xenium data on two separate histology sections from the biopsy used for sequencing for patient H1** (Fig. 4e and Ext Data Fig 1). In the manuscript we show two histology sections with Xenium spatial transcriptomics data for patient H1, one with the 380-gene immunoncology panel and another with the 5k panel (including in revision). Both were generated from the biopsy used for sequencing, but they are sufficiently separate to show some differences in cell type composition. The Xenium 380 section (shown in Fig. 4e) did not cut across any germinal centres and is indeed dominated by plasma cells, as noted by the reviewer. However, the Xenium 5k section (shown in Ext Data Fig 1) contains clear B cell aggregates, as well as some T cells and residual thyroid epithelium. We hope that this reassures the reviewer. In the first revision, in Ext Data Fig 1 we also provided several plasma cell markers (XBP1 and SLAMF7/CD319) to provide further reassurance that these are indeed plasma cells. An additional image (including the plasma cell marker CD138, requested by the reviewer here) was also provided in our response to reviewer 1 in the first revision.

In summary, in the manuscript we already provide IHC/ISH and spatial transcriptomics evidence of plasma cells, B cells and T cells in patient H1 (Fig 1a, Fig 4e, and Ext Data Fig 1). The reviewer seems to have been confused by the section shown in Fig 4e, which did not cut across any germinal centres. To avoid confusion, we now cite the Xenium 5k supplementary images (Ext Data Fig. 1) in the figure legend of Fig 4e. We also provide the markers requested above by the reviewer (CD3, CD138, CD19, BLIMP-1) in the image below for the Xenium 380 and 5k sections for patient 1. We note, however, that this is largely redundant with the evidence already included in Ext Data Fig. 1b, so to avoid confusion, we have not included these panels in the manuscript. However, we have deposited the Xenium dataset in a repository for access to readers interested in specific markers. We hope this clarifies the confusion.

We appreciate the effort the authors have made to include better controls and inclusion of a large variety of additional tissue samples. Nevertheless, a more transparent approach in comparing mutational burden in 2 genes would be to compare tissue sampled the same way. I.e. Tissue sampled from AITD thyroid by microdissection and specified to contain >25% lymphocytes should ideally be compared to control tissue also captured by microdissection and with a >25% lymphoid content (i.e. ensuring the samples are taken from white pulp rather than red pulp in the spleen, and ideally germinal centers (GC) in tonsil (since human spleen barely contains reactive lymphocytes or GCs). The number of donors sequenced should also be made clear in the text: i.e. 3 donors for colon, 1 for Peyer's patches in ileum and 1 for tonsil were sequenced. At present, the text mentions n=5 (line 276).

Authors' response: Thank you. It is important to clarify that, whereas we generated extensive microdissection data to map the distribution of the clones in the samples (e.g. see Fig 4), the driver discovery datasets used for the pilot and extension cohorts were bulk thyroid biopsies with varying degrees of B cell infiltration (as shown in Fig 1b and 2b). In the revision, we similarly generated deep targeted NanoSeq control data for 5 new control datasets, including lymph nodes, spleens, tonsils, non-autoimmune thyroids and a collection of 183 laser microdissected Peyer patches and other lymphoid aggregates. This was done in addition to a deeply sequenced dataset of flow-sorted B-memory cells included in the original submission, which was chosen as the strictest negative control of largely pure post-germinal centre B cells from older donors (>50 years).

The reviewer makes two comments here, one regarding the cell type composition of the control samples, and another regarding a minor edit to the text to indicate the number of donors. Both are easy to address. We have now made the requested edit to the text, adding the number of donors used for the control datasets. Regarding the cell type composition of the controls, fortunately we have targeted methylation data for many of the samples used in the new control datasets as well as in the bulk AITD samples (Fig 1b and 2b), so we can compare their B-cell content side-by-side. As you can see below, our control datasets generally had comparable (or higher) fractions of B cells as our AITD discovery cohort. We have now added this figure to the manuscript to address this concern (new Extended Data Fig. 4a).

Fig. This figure shows stacked barplots for the estimated cell type composition for our bulk AITD samples (top row) and the control datasets with available targeted methylation data (bottom row). Note that the fraction of B (and T) cells is comparable (or higher) in our B-cell rich controls to the bulk AITD samples. The top panel is composed of figures shown in Fig 1 and Fig 2, and the bottom panel is now shown as Extended Data Fig. 4a.

Reviewer #3:

The authors have done an outstanding job addressing the vast majority of the reviewer concerns. The addition of spatial transcriptomics and additional single-cell analyses substantially strengthens the manuscript.

The one response I had some remaining concerns with relates to the κ/λ staining in Fig. 1a. The explanation provided remains somewhat indirect, as it does not fully clarify whether the κ/λ signal primarily reflects plasma cells versus extracellular Ig deposition, nor why CD20⁺ follicular regions appear devoid of κ/λ staining. This does not detract from the central conclusions of the manuscript; however, a brief clarification in the text or figure legend would improve interpretability for readers. Additional detail in the Methods section describing the staining protocol and antibodies used would also be helpful, as this information is currently lacking.

Overall, this is a highly compelling and important study.

Authors' response: We thank the reviewer for their positive comments.

Regarding the κ/λ staining, there is an important clarification that was described in the figure legend but that we failed to emphasise in our response. The κ/λ staining was done by in-situ hybridisation (ISH) not by immunohistochemistry (IHC). This is important as ISH stains cytoplasmic mRNA and does not stain extracellular immunocomplexes. Lack of κ/λ staining in germinal centres is also typical. We hope that this clarification adds further confidence that the CD20- κ/λ ⁺ areas correspond to plasma cells, which is more definitively proven by the Xenium 380 and Xenium 5k panel results shown for patient H1 in Fig. 4e and Extended Data Fig. 1, respectively.

To avoid confusion, we have made the figure legend clearer and we have improved Extended Data Fig. 1 with additional plasma cell markers in the Xenium 380 and Xenium 5k panels.

Reviewer #5

In this revised manuscript, the authors have addressed all our concerns, as well as the concerns of the other reviewers. I particularly appreciate the authors' efforts in including extensive non-autoimmune B cell negative controls. I congratulate the authors on their important study.

Authors' response: Thank you very much.

I have only a few minor issues remaining:

1. Extended Data Figure 4 and Figure 3d should have a color legend. The significance ****/***/**/* annotations are too small to be visible on their own, and the duplex coverage blue squares need a scale bar as well.
2. Two typos: (i) line 108: "This was was used"; (ii) line 577: "highlighted blue indicate" -> "highlighted in blue"

Authors' response: Thank you for spotting this. We have corrected these two typos and added the colour legends and annotations suggested by the reviewer.

Comments about code:

The R notebook provides is very clear and well documented.

One minor comment: the authors mention they provide the code for dNdSshm, but we couldn't find it. As we understand, the code provided refers to a separate dnds_shm function that is loaded from another file, but that file was not provided.

Authors' response: Thank you. In the final version of the manuscript, we include a link to the final code repository with input files and code (including the code for the dnds_shm function). This is in addition to the html Supplementary Code file (with in-line comments, tables and code generated by an R markdown).